# Surface cooling caused by rare but intense near-inertial wave induced mixing in the tropical Atlantic

Rebecca Hummels [1] ✉, Marcus Dengler [1], Willi Rath [1], Gregory R. Foltz[2], Florian Schütte[1], Tim Fischer [1] & Peter Brandt [1,3]

The direct response of the tropical mixed layer to near-inertial waves (NIWs) has only rarely been observed. Here, we present upper-ocean turbulence data that provide evidence for a strongly elevated vertical diffusive heat flux across the base of the mixed layer in the presence of a NIW, thereby cooling the mixed layer at a rate of 244 W m$^{-2}$ over the 20 h of continuous measurements. We investigate the seasonal cycle of strong NIW events and find that despite their local intermittent nature, they occur preferentially during boreal summer, presumably associated with the passage of atmospheric African Easterly Waves. We illustrate the impact of these rare but intense NIW induced mixing events on the mixed layer heat balance, highlight their contribution to the seasonal evolution of sea surface temperature, and discuss their potential impact on biological productivity in the tropical North Atlantic.

[1] GEOMAR Helmholtz Centre for Ocean Research Kiel, Kiel, Germany. [2] NOAA/AOML, Miami, FL, USA. [3] Kiel University, Kiel, Germany.
✉email: rhummels@geomar.de

Sea surface temperature (SST) is a crucial parameter defining air–sea interactions in the tropical oceans as indicated by the close link between the seasonal cycles of SST and the marine intertropical convergence zone (ITCZ)[1,2]. In the tropical Atlantic, deviations from the mean seasonal cycle of SST result in variations of the position and strength of the rainfall belt impacting, e.g., the onset of the West African Monsoon[3,4] or droughts over Northeast Brazil[5,6]. The prediction of such rainfall variability or other tropical climate extremes such as tropical cyclones is essential for countries surrounding the tropical Atlantic Ocean in order to improve agricultural productivity, allow for more efficient use of water resources, and protect homes and infrastructure against such extremes[7]. Realistic coupled climate model simulations, which are required for prediction of tropical climate and its extremes, are thus of high societal relevance. Improving the observational database is crucial for providing the necessary constraints for model evaluation and for understanding the relevant processes that drive SST variability[8].

Investigations of the processes driving SST variability have been motivated by strong warm biases in the eastern equatorial region and along the Southwest African coast[9] that are still inherent to coupled climate models and hamper climate predictions[10,11]. Models also have severe difficulties simulating realistic biological productivity and biogeochemical cycles, including chlorophyll *a* distribution[12], which are potentially related to biases in SST and ITCZ representation[13,14].

A useful approach to evaluate the processes driving SST variability is the analysis of the individual components of the mixed layer (ML) heat budget. This requires observations of air–sea heat fluxes, upper ocean horizontal velocities and temperatures, and estimates of vertical exchange processes at the ML base. The Prediction and Research Array in the Tropical Atlantic (PIRATA) mooring sites[15] are preferred locations for ML budget studies as most of the required observations are readily available[16–21].

Earlier studies from the tropical Atlantic lacked observations of the vertical diffusive heat flux caused by turbulent mixing across the base of the ML (termed vertical diffusive ML heat flux in the following), but conjectured that it plays an important role because the ML heat budgets require a considerable additional sink of heat (e.g. refs. [7,16,22]). Similar results were obtained for the tropical Pacific from ML budget analysis at the TAO mooring sites

(e.g. ref. [23]). In recent years, upper ocean turbulence data have become increasingly available in the equatorial Atlantic and Pacific, and have supported the hypothesis of elevated vertical diffusive ML heat loss in the central and eastern equatorial oceans[19,20,24,25]. The vertical diffusive ML heat flux is among the largest cooling terms in the ML budgets of the equatorial regions. It is associated with elevated turbulence at the base of the ML resulting from instabilities of the strong mean vertical shear of the mean energetic equatorial currents, particularly the eastward Equatorial Undercurrent and the westward flow above, locally enhanced by other processes such as tropical instability waves[19,24,26]. Similar results were inferred from numerical models[27–31]. Away from the equator large residuals in ML budgets were found as well[18,21,23]. However, sparse turbulence observations prior to this study indicated only weak vertical diffusive ML heat fluxes, which were insufficient to explain those residuals (e.g. ref. [19]). As elevated vertical shear from the mean current system is not pronounced at these latitudes[32], it was hypothesized that infrequent, but very strong vertical mixing events dominate the mean vertical diffusive ML heat flux[15].

There are arguments that near-inertial waves (NIWs) are a suitable candidate for driving this missing cooling. A substantial amount of the total ocean internal wave kinetic energy is observed to correspond to the near-inertial frequency band: Of the $2.0 \pm 0.6$ TW of globally averaged dissipation for the internal wave field, 0.3–1.0 TW are estimated to be associated with wind-forced NIWs, most of it in the upper ocean[33]. NIWs are characterized by circularly polarized velocities that can be resonantly forced in the ML by wind stress fluctuations with near-inertial frequencies[34,35]. They are associated with strong vertical shear of horizontal velocity at the ML base and may lead to intermittent events of energetic upper-ocean mixing[8,34]. By causing strong vertical mixing at the ML base, NIWs not only contribute to cooling of the ML, but also alter the concentrations of salt and nutrients within the ML, the latter further impacting biological productivity and biogeochemical cycles[36,37].

Here, we report on upper-ocean turbulence, velocity, and hydrographic observations taken in the tropical North Atlantic at about 11°N, 21°W (Fig. 1) in September 2015 capturing a strong mixing event caused by a NIW. By combining these data with observations from a nearby PIRATA buoy (11.5°N, 23°W, Fig. 1), the development, strength, and atmospheric forcing of this event are inferred, and its contribution to the ML heat budget is discussed. It is shown that NIWs likely generate a strong mean vertical diffusive ML heat flux in large parts of the tropical North Atlantic. To substantiate this statement, satellite winds are analyzed for their potential to induce NIWs in the surface ocean and thereby to potentially cause elevated shear at the ML base. We further show the impact of NIW-induced mixing on the redistribution of other variables, in the form of an elevated freshwater flux and by the surface chlorophyll pattern in the eastern tropical North Atlantic (ETNA). These results generally agree with the pattern of elevated inertial activity (IA) inherent to the wind forcing.

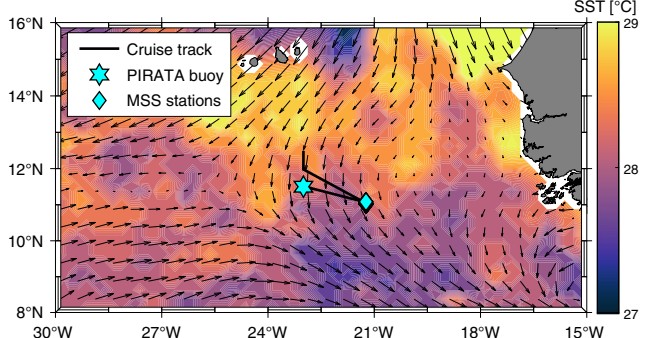

**Fig. 1 Overview map of observational data base.** Location of shipboard observations during September 13–15, 2015 (black line: cruise track; cyan diamonds: microstructure (MSS) stations) and the PIRATA buoy (cyan star). Note that the MSS stations took place almost at the same location and hence appear as a single location. Along the cruise track continuous observations of upper ocean velocity from the acoustic Doppler current profiler (ADCP) are available. Coloring shows a sea surface temperature snapshot of September 14 and arrows show the wind field of September 14, 00:00 UTC.

## Results

**ML heat budget at the PIRATA buoy site.** The seasonal ML heat balance at the 11.5°N, 23°W PIRATA buoy is evaluated using monthly means from the daily ePIRATA estimates[18]. The ePIRATA product contains daily averages of several terms in the heat balance, including net surface longwave radiation, shortwave radiation absorbed in the ML, air–sea turbulent heat fluxes, horizontal advection, and ML heat storage rate. The main advantages of ePIRATA are that instrumental biases have been corrected and that PIRATA buoy observations are complemented

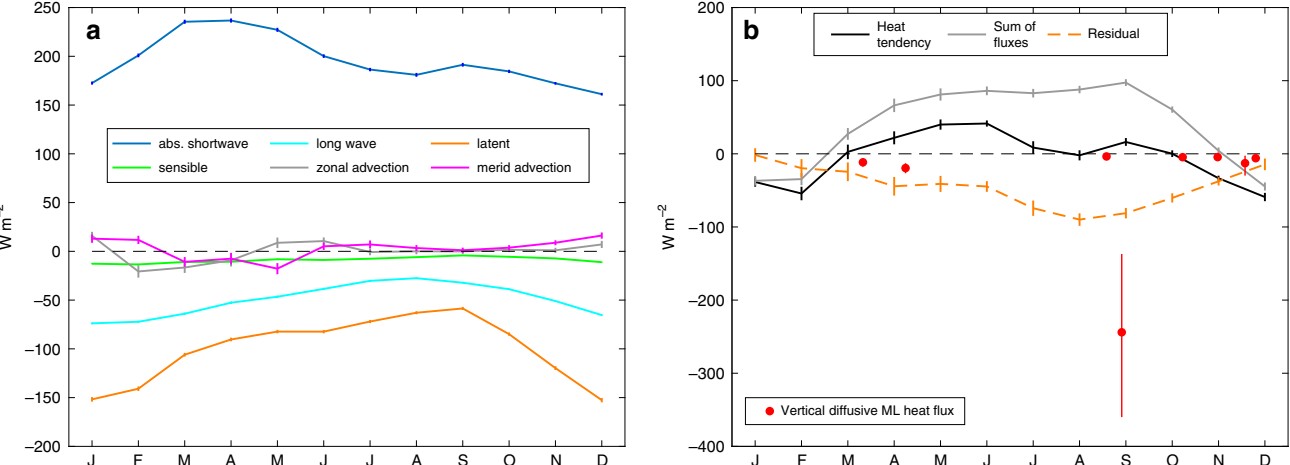

**Fig. 2 Seasonal mixed layer heat budget at 11.5°N, 23°W. a** Seasonal cycle of the individual contributions to the mixed layer (ML) heat budget as indicated in the legend. **b** Heat tendency (black) and the sum of the individual heat budget contributions (gray; sum of fluxes) as shown in **a**. The residual of these two terms is shown by the dashed orange line. Red dots indicate the vertical diffusive ML heat flux estimates from microstructure observations of the eight existing realizations at this location (vertical red lines denote the 95% confidence limits). Error estimates for the individual ePIRATA terms and the residual are calculated as explained in the "Methods" section. Note that the ticks on the x-axis correspond to the middle of the month.

by Argo profiles (see "Methods" section for details). The largest terms from the ePIRATA heat balance are shortwave radiation absorbed in the ML (annual average of 195 W m$^{-2}$), net outgoing longwave radiation (annual average of $-50$ W m$^{-2}$) and the latent heat flux (annual average of $-100$ W m$^{-2}$, Fig. 2a). Monthly averages of horizontal heat advection are smaller than 18 W m$^{-2}$ throughout the year, while the vertical heat exchange at the base of the ML is inferred as a residual term (Fig. 2b) that also includes the errors of the other contributions.

The residual term accounts for an average heat loss of nearly 100 W m$^{-2}$ during boreal summer, representing a substantial contribution to the heat balance. In contrast, vertical diffusive ML heat loss at this location, estimated from microstructure shear sensor (MSS) data, is often much weaker: during seven out of eight sampling periods, totaling 12 days of continuous sampling (see "Methods" section), the heat loss is between 3 and 16 W m$^{-2}$ (Fig. 2b). These low vertical diffusive ML heat fluxes agree with the lack of elevated vertical shear of horizontal currents observed during the cruises at this location and cannot explain the large residual. However, there is one striking exception: the vertical diffusive ML heat flux estimate from MSS data collected in September 2015 shows an elevated heat loss of 244 [349; 134] W m$^{-2}$ averaged over the 20 h of observations (Fig. 2b, numbers in brackets indicate the upper and lower 95% confidence limits determined from statistical error propagation[38]. Note that systematic errors, which may be as large as a factor of 2[39] are not included in this estimate). This event clearly exceeds the vertical diffusive ML heat flux estimates from all other sampling periods and will be analyzed in detail in the following (Figs. 2b, 3, and 4).

**Enhanced vertical mixing during a strong NIW event.** During the cruise M119 of RV Meteor in September 2015, 25 microstructure profiles were taken coincidentally during an energetic NIW event. Concurrently, vertical profiles of horizontal velocity from a vessel-mounted acoustic Doppler current profiler (vmADCP) were available from 17 m to about 800 m depth (Figs. 1, 3).

NIWs are recognizable by their characteristic circularly polarized horizontal velocities and the resulting vertical shear at the ML base[34]. They have a defined frequency between 1 and 1.2$f$, with $f$ being the local Coriolis parameter. At 11.5°N, this corresponds to a period of 2–2.5 days. Observed upper-ocean velocities from September 2015 rotate clockwise in time and have a period of about 2.4 days (Fig. 3a), which corresponds very well to the local inertial frequency. Horizontal velocities associated with this NIW reach up to 0.6 m s$^{-1}$ in the first vmADCP bin centered at 17 m (13–21 m), dropping to nearly zero in the second bin centered at around 25 m (21–29 m; Fig. 3b, c). Downward energy propagation of the NIW is indicated by an anticyclonic rotation of the current vector with depth[40]. Its baroclinic structure results in strongly elevated squared vertical shear at 8-m scales, $Sh^2 = \left(\frac{du}{dz}\right)^2 + \left(\frac{dv}{dz}\right)^2$, reaching values on the order of $10^{-3}$ s$^{-2}$ between 21 m (upper limit of our measurements) and 40 m depth (Fig. 3d). Despite the general absence of strong mean currents at this location (e.g. ref. [32]), the observed levels of $Sh^2$ during this NIW event are comparable to those of the energetic equatorial undercurrent, reaching values of $Sh^2 \sim 5 \times 10^{-4}$ s$^{-2}$ (e.g. ref. [19]).

The mixed layer depth (MLD) was situated between 15 and 20 m depth during the observational period of 20 h (Fig. 3e–h). Here, MLD is calculated as the depth where the density is 0.12 kg m$^{-3}$ larger than the density at 1 m depth. This criterion is chosen as the minimum density difference required to exclude short-lived (diurnal or shorter) transient oscillations of the ML. Below the ML, stratification in the thermocline and halocline (Fig. 3e) reaches values of $N^2 \sim 10^{-3}$ s$^{-2}$ at 8-m scale (Fig. 3f). The ratio of $N^2$ to $Sh^2$, i.e. the Richardson number ($Ri$), $Ri = \frac{N^2}{Sh^2}$, is a measure of the tendency for shear instability and turbulent overturns to develop. It shows small values of <1 below the ML (Fig. 3h). However, the 8m-$Ri$ values are rarely smaller than the actual critical value of $Ri_c = 0.25$. Note that our 8m-$Ri$ calculation is limited by the coarse 8-m vertical resolution of the velocity data and needs to be considered as an upper bound for $Ri$, with 8m-$Ri \leq 1$ (Fig. 3h) suggesting $Ri \leq Ri_c$ at smaller vertical scales. Additionally, the microstructure temperature data revealed frequent overturns with typical scales[41] of 0.3–0.5 m in the strongly stratified layer below the ML[42]. Dissipation rates of turbulent kinetic energy ($\varepsilon$) inferred from these overturning scales according to Ozmidov[43] (Ozmidov length scale $L_{OZ} = \sqrt{\varepsilon N^{-3}}$) agree well with the elevated $\varepsilon$ determined from microstructure measurements (Fig. 3g).

Patches of highly elevated $\varepsilon$ of up to $2 \times 10^{-5}$ m$^2$ s$^{-3}$ are regularly observed below the ML during the entire time series of

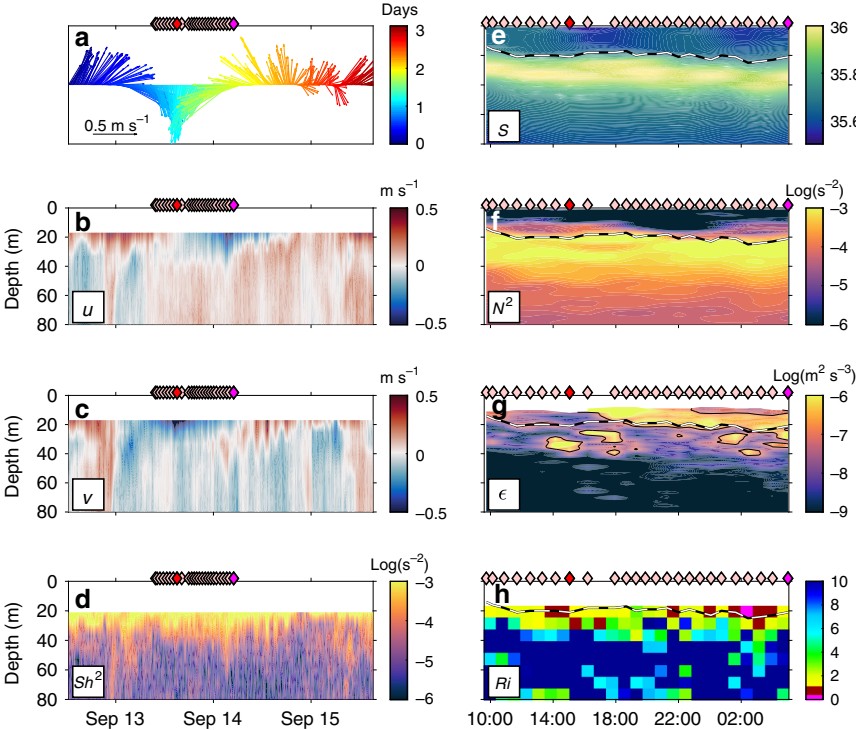

**Fig. 3 Near inertial wave event near 11.5°N, 23°W during September 2015.** Data from the vessel-mounted acoustic Doppler current profiler (vmADCP) from top to bottom: **a** current vectors from the first vmADCP bin (13–21 m depth), **b** zonal velocity ($u$), **c** meridional velocity ($v$), and **d** squared vertical shear of horizontal velocities ($Sh^2$). Microstructure profiler observations acquired between September 13, 10:00 UTC and September 14, 06:00 UTC, from top to bottom: **e** salinity ($S$), **f** squared buoyancy frequency ($N^2$), **g** dissipation rates of turbulent kinetic energy ($\varepsilon$), and **h** 8m-$Ri$ number ($Ri$). On all panels times of microstructure profiles are indicated with light red diamonds, whereas two specific profiles are highlighted in red/magenta and shown in detail in Fig. 4. Color-coding in **a** corresponds to time after first considered vmADCP observation for better visibility of the individual current vectors. Black and white dashed line in **e**–**h** denotes the mixed layer depth.

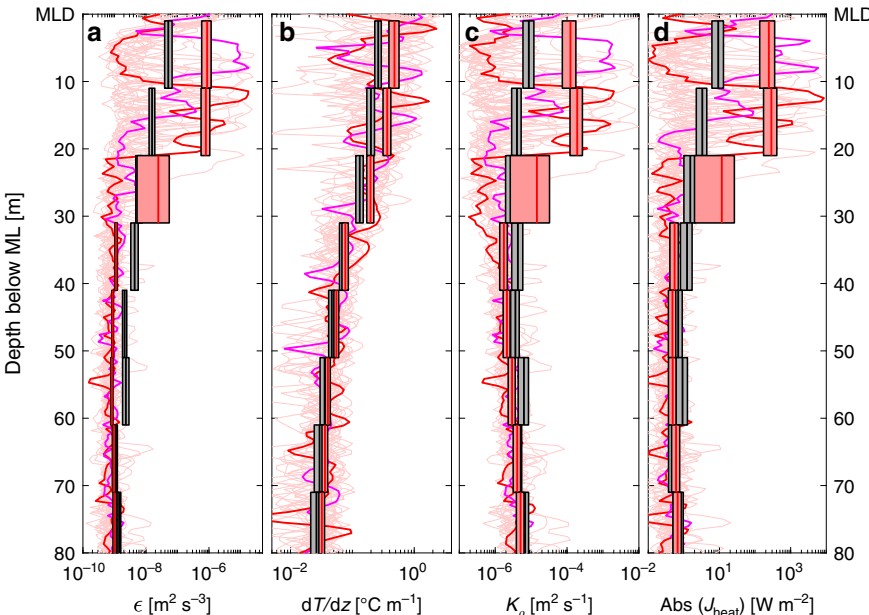

**Fig. 4 Observed mixing parameters near 11.5°N, 23°W.** 10-m average profiles observed with the microstructure profiler (MSS) in the vicinity of the PIRATA buoy of **a** turbulent dissipation rates ($\varepsilon$), **b** vertical temperature gradients ($dT/dz$), **c** eddy diffusivities ($K_\rho$) and **d** vertical diffusive heat fluxes ($J_{heat}$) during September 2015 as thick red bars. All additionally available MSS observations of the other cruises in thick black 10-m average bars. Gray/light red patches around the bars denote the 95% confidence limits. Light red lines show the individual profiles with full vertical resolution from September 2015, where two exemplary profiles are highlighted in red/magenta. The two highlighted profiles are also localized as red/magenta diamonds in Fig. 3.

MSS observations. These turbulence patches are likely sustained by local shear instability[44] due to elevated shear of the baroclinic NIW currents. In some instances, for example at 14:00 UTC, the peak in elevated $\varepsilon$ below the ML coincides with rather low $\varepsilon$ within the ML. This additionally supports the idea that the intense turbulent mixing patches result from local shear that was accumulated during the preceding period of resonant forcing and not due to direct wind forcing, which would also elevate dissipation rates within the ML[45].

Vertical profiles of $\varepsilon$, vertical temperature gradient ($dT/dz$), diapycnal eddy diffusivity ($K_\rho$) and vertical diffusive heat flux ($J_{heat}$) (see "Methods" section for details about the derivation of $K_\rho$ and $J_{heat}$) during the September 2015 NIW event were strongly elevated below the ML compared to all other available MSS observations at this location (Fig. 4a–d). From the vertical profiles, vertical diffusive ML heat and salt fluxes were determined by averaging the fluxes in the depth layer between 1 and 10 m below the ML[19]. The average ML heat loss estimated from all 25 profiles taken in September 2015 during the 20 h of observation was 244 [349; 134] W m$^{-2}$. This heat loss corresponds to a cooling of the ML of more than 0.25 °C day$^{-1}$. The mixing event also strongly impacted the ML freshwater balance. The vertical diffusive ML salt flux was estimated to be $3.5 \times 10^{-6}$ [$4.9 \times 10^{-6}$; $2.6 \times 10^{-6}$] m s$^{-1}$, which is equivalent to an increase of salinity in the ML of 0.015 day$^{-1}$.

In summary, the observations show strongly enhanced vertical diffusive ML heat flux during the presence of a NIW that was encountered during one out of eight sampling periods at the location. Determining the extent to which such short-lived, possibly rare events impact the seasonal or mean ML heat budget requires a quantification of the frequency of their occurrence.

**Modeling the near-inertial ML velocities at the PIRATA buoy site.** We use a linear slab model[35] to investigate local wind forcing of the NIW event during the measurement program. Slab models are capable of representing the wind-driven near-inertial currents in the ML for several inertial cycles[46] (see "Methods" section for details). The model solutions are the sum of an Ekman transport and inertial motions modulated by the MLD. In the Northern Hemisphere the Ekman transport is to the right of the wind and the inertial motions rotate anticyclonically[35]. Here, the slab model is driven by hourly 4-m winds recorded at the PIRATA buoy and a climatological seasonal cycle of the MLD based on the daily ePIRATA MLD available from 2006 to 2017 (Supplementary Fig. 1). A climatological seasonal cycle of MLD is required to provide the slab model with a MLD that is quasi-constant on the inertial time scale. Wind stress was calculated from 10-m winds using the drag coefficient parametrized as in ref. [47] with a levelling for high wind speeds according to Donelan et al.[48]. For the 4-m PIRATA wind measurements, we assumed a constant vertical profile between 4 and 10 m. The linear damping time scale of the slab model ($r^{-1}$) is a tunable parameter and is usually set between 2 and 10 days[49]. An appropriate value is often chosen by seeking the best agreement between model output and observations of inertial currents associated with a particular forcing event[46]. We set it to 5 days and further support this choice with comparisons to the decay scale of the wind power input (WPI) at near inertial frequency (see Supplementary Fig. 2 and Supplementary Note for details). To validate the use of the slab model at this location, the resulting slab ML velocities are compared to hourly velocities at 12 m depth observed at the PIRATA buoy, available from November 2015 to the present. To separate the slowly varying Ekman response from the near-inertial currents, the slab model and the observed velocities are band-pass filtered with a third-order Butterworth filter retaining frequencies

between $0.7f$ and $1.3f$. The filtered velocities show significant correlation over the 2-year time period, with a skill of $R^2 = 0.66$–$0.68$ for both the zonal and the meridional velocity components (significant at the 1% level; a decorrelation scale of 5 days equal to the damping scale is chosen to determine the degrees of freedom for the test). This indicates that a large fraction of the observed ML velocity variability at this location is due to the response to local winds and can be simulated realistically with the slab model.

**Using the PIRATA-wind-forced slab model at the MSS site.** With the PIRATA buoy and the MSS site ~185 km apart, it is necessary to show the applicability of the winds observed at the PIRATA site for modeling the near-inertial ML response at the MSS site. The wind event that triggered the NIW covered ~5° in latitude and longitude (see Supplementary Fig. 3). Hence, the scale of the atmospheric forcing event is about three times larger than the distance between the PIRATA buoy and the location of shipboard observations. Therefore, it is most likely that both locations experienced similar wind forcing in terms of timing, magnitude, and direction, thus resulting in a similar inertial response. In addition, the MLD is similar (17 m ePIRATA vs. 19 m MSS observations on September 13, 2015). This suggests that it is appropriate to compare the vmADCP observations during September 2015 (Fig. 5c) to the output of the slab model forced with wind stress derived from observed PIRATA winds (Fig. 5a, b) using ePIRATA MLD and the damping time scale of $r^{-1} = 5$ days.

The amplitudes and phases of the ML velocities simulated with the slab model agree well with those of the vmADCP. A slight phase shift in the zonal velocity component is evident, less in the meridional component, which could be caused by higher-order effects not accounted for in the slab model or by small spatial variations in or inaccurate translation of the wind field. The MSS observations were acquired during a particularly strong response of ML velocities, characterized by a peak of total kinetic energy from the slab model during the time of observations (Fig. 5d). In addition, the flux of kinetic energy from the wind to inertial motions in the ocean (WPI, also see Supplementary Note for details) shows a peak during this event. This shows that most of the kinetic energy increase takes place in the inertial frequency band. The ML response observed with the vmADCP can be attributed to a certain wind event: shortly before the NIW event the wind turned in a clockwise (anticyclonic) direction with a period close to the local inertial period (Fig. 5a, Supplementary Fig. 3). Although the wind was not particularly strong (about 8 m s$^{-1}$), the sense of rotation and frequency effectively triggered the strong response in ML velocities as also shown in the peak in WPI. At other times (e.g. September 1) the wind is stronger (up to 10 m s$^{-1}$), but as the rotation direction and frequency do not match the inertial motion, WPI stays rather low and the response in the ML is weaker (no peak in kinetic energy).

**Prevalence of strong NIW events in the tropical Atlantic.** The MSS observations showed elevated vertical diffusive ML heat fluxes at the ML base associated with low $Ri$ numbers, while the concurrent velocity observations revealed that the observed elevated shear is related to the presence of a downward propagating wave. The combination of these observations with the results of the slab model strongly suggests that the velocities are associated with a strong NIW event. The question then arises how often the wind causes similar responses in and below the ML. Assuming a vertical mixing parameterization based on the $Ri$-number approach, the role of NIWs can be expressed by the squared vertical shear of horizontal velocities ($Sh^2$) at the base of the ML

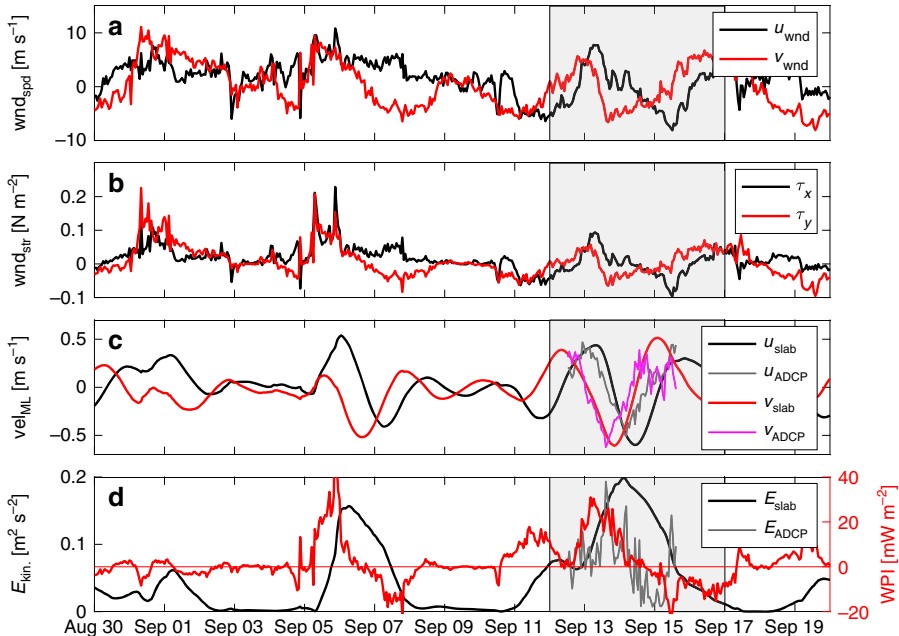

**Fig. 5 Slab ocean response compared to observed velocities. a** Zonal ($u_{wnd}$) and meridional ($v_{wnd}$) 4-m wind speed from the PIRATA buoy at 11.5°N, 23° W, and **b** the resulting zonal ($\tau_x$) and meridional ($\tau_y$) wind stress components. **c** Zonal ($u_{slab}$) and meridional ($v_{slab}$) mixed layer velocities as obtained from the slab model (thick black and red lines, respectively) compared to the zonal ($u_{ADCP}$) and meridional ($v_{ADCP}$) velocity observed at 17 m with the vessel-mounted acoustic Doppler current profiler (vmADCP) during September 2015 (thin gray and magenta lines, respectively). **d** Total kinetic energy from the slab model and observed vmADCP velocities (thick black and gray lines, respectively) and wind power input (WPI, red). The latter had to be calculated with slab model velocities as the high temporal resolution of mixed layer currents from PIRATA is only available from November 2015. Gray shading in all panels indicates the time defined for the NIW event also used in Fig. 7 for the instantaneous heat budget at the PIRATA buoy.

for any given stratification. $Sh^2$ induced by NIWs will depend on the magnitude of NIW-induced currents within the ML, which, in turn, is influenced by the thickness of the ML distributing the wind input. Distribution over a larger (smaller) depth range leads to weaker (stronger) ML velocities and hence $Sh^2$. To separate the roles of these two factors, i.e. the wind input and the MLD, their respective contributions are investigated independently.

The ability of the wind stress to force near-inertial motions in the ML is estimated with a slab model using a constant product of ML depth and density, which is set to unity. This slab model can be identically rewritten in the form of a linear filter, which is done to highlight the fact that the output of the slab model is actually filtered wind stress (see "Methods" section).

We define IA as

$$\text{IA} = \frac{|q|^2}{\text{RMS}(|q|^2_{\text{PIRATA}})} \qquad (1)$$

using the filter output $q = u + iv$ and its root-mean-square (RMS) determined from the whole PIRATA time series. Note that $q$ contains a slowly varying Ekman component that is eliminated by applying a 30-day high-pass filter prior to calculating IA. Due to gaps in the PIRATA wind record, the CCMP[50,51] wind field from 1987 to 2017 is additionally used. This approach is validated by comparing the IA from CCMP with the IA from PIRATA winds (Fig. 6a).

In the context of the linear filter, a near-inertial event (NIE) is defined using a threshold value for IA. As the focus here is on the seasonal variability of the occurrence of NIEs, the relative distribution of NIEs as measured by a threshold criterion should be robust against the choice of the threshold value. The climatological monthly prevalence of NIEs is defined as the fraction of time per month with IA above a given threshold $\alpha$ (Fig. 6a). Figure 6b shows that the climatological prevalence of strong NIEs is robust against the choice of different thresholds,

with a maximum number of NIEs found between July and September. Only the monthly prevalence, defined as the total fraction of time with NIEs per month (detected days with NIEs per month), changes for different $\alpha$, with maxima at 0.33, 0.15, and 0.09 (10, 4.6, and 2.7 days) for $\alpha = 1$, 2, and 3, respectively. In the following we will always refer to the monthly prevalence of NIEs for IA crossing the threshold of $\alpha = 2$ (explicitly shown without normalization in Fig. 6c).

The seasonal cycle of the MLD, which has not been considered in the linear filter, ranges from about 30 m at the beginning of the year to about 15 m during boreal summer (Fig. 6d). Hence, strong near inertial currents are most likely to occur between July and September. At this time of the year, the maximum monthly prevalence of NIEs coincides with the shallowest MLD within the seasonal cycle (Fig. 6c, d).

## Discussion

Velocity observations from September 2015 revealed the presence of strongly elevated vertical shear of horizontal velocity ($Sh^2$) in the vicinity of the PIRATA buoy at 11.5°N, 23°W. The strong $Sh^2$ was found to be associated with a NIW and was two orders of magnitude larger compared to values from previous cruises. The vertical diffusive ML heat loss inferred from concurrently collected turbulence observations yielded 244 [349; 134] W m$^{-2}$, which by far exceeded all previously estimated vertical diffusive ML heat losses at this location. Mixed layer deepening due to local wind stress and buoyancy fluxes cannot explain the elevated vertical diffusive ML heat loss. An evaluation of a one-dimensional ML model, which only accounts for direct wind forcing[45] (see "Methods" section for details), initialized with microstructure temperature profiles and using wind speeds from the PIRATA buoy at 11.5°N, 23°W yields a heat flux across the base of the ML of only 18–22 W m$^{-2}$. This supports the conclusion that the observed intense mixing at the base and below

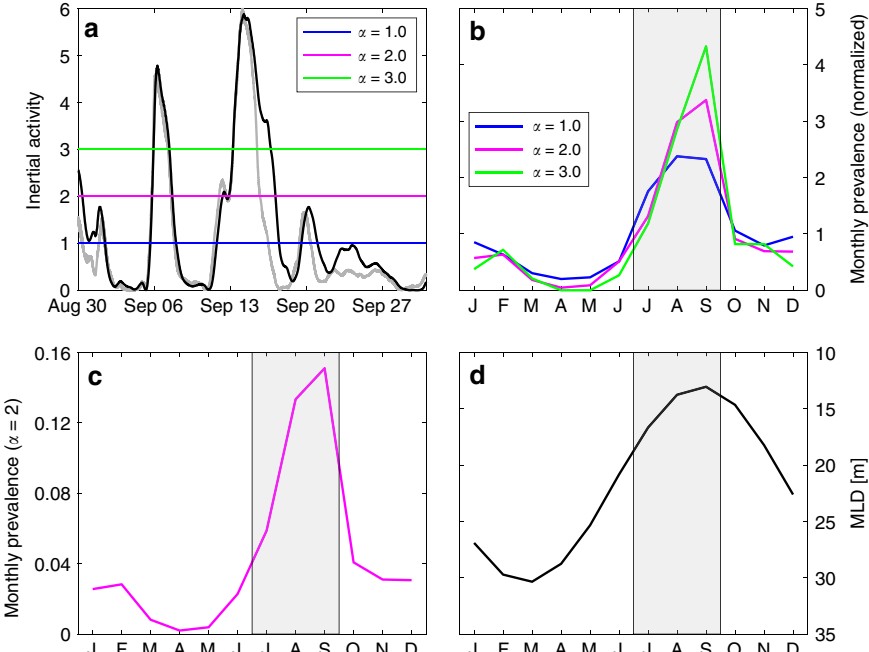

**Fig. 6 Inertial activity evolution and monthly climatology at 11.5°N, 23°W. a** Inertial activity (IA) index (gray solid line based on PIRATA winds at 11.5°N, 23°W, black solid line based on CCMP winds at the grid point 11.125°N, 23.375°W) from August 30 until October 1 including the observed near inertial wave event around September 15. The colored lines mark different thresholds $\alpha$ for IA. Gray shading highlights the boreal summer season July–September (JAS). **b** Seasonal cycle of monthly prevalence for IA (based on CCMP winds at the grid point 11.125°N, 23.375°W) crossing the different thresholds $\alpha$, normalized by their respective annual means (0.14, 0.045, and 0.02) for comparison. **c** Seasonal cycle of monthly prevalence for IA crossing the threshold $\alpha = 2$ (as in **b**), but without normalization. **d** Seasonal cycle of the mixed layer depth (MLD) from ePIRATA at 11.5°N, 23°W.

the ML resulted from local shear instability that was a consequence of resonant wind forcing.

Another independent approach to estimate the magnitude of the vertical diffusive ML heat flux during the NIW event is to quantify the SST decrease during the event and the associated heat balance at the 11.5°N, 23°W PIRATA buoy site. We evaluate the local balance during the event at the PIRATA buoy site because not all required heat balance terms are available from the shipboard observations. The event is defined according to Figs. 5d, 7a to take place from September 12 to 16, 2015. During this time period, SST and ML temperature drop by 0.28 K (Fig. 7b). The heat budget equation can be expressed according to previous studies (e.g. refs. [16,52,53]) as

$$\rho c_p h \left( \frac{\partial T}{\partial t} + \mathbf{v} \cdot \nabla T \right) = Q_0 - R \qquad (2)$$

with $\rho$ and $c_p$ ML density and heat capacity, respectively, $h$ the MLD and $T$ and $\mathbf{v}$ the vertically averaged temperature and horizontal velocity in the ML, respectively. $Q_0$ is the net surface heat flux corrected for the penetrative heat loss through the ML base. All other terms as well as errors are included in the residual term $R$. Considering previous ML heat budget studies of strong equatorial mixing events [19,20,24], it can be conjectured that because the NIW event was associated with elevated turbulent mixing at and below the base of the ML, $R$ is by far dominated by the vertical diffusive ML heat flux as well.

Considering a net surface heat flux of $Q_0 = 80 \, \mathrm{W \, m^{-2}}$ (average surface flux from ePIRATA data during the 5 days, hence a warming term), and the advective heat flux and the heat storage rate (second and first term on the left-hand side of Eq. (2)) from ePIRATA, the residual cooling over the 5-day time period amounts to $150 \, \mathrm{W \, m^{-2}}$. Despite the various uncertainties in this additional estimate, it strongly supports the idea that during the NIW event the vertical diffusive ML heat flux was strongly

elevated. Note that the microstructure observations were taken during the peak of the event (Fig. 5d), and most likely vertical diffusive ML heat fluxes before and after the measurements were less intense. This reconciles the $150 \, \mathrm{W \, m^{-2}}$ over the 5-day period with the MSS observations of $244 \, \mathrm{W \, m^{-2}}$ over 20 h and is consistent with the observed decrease in SST and ML temperature (Fig. 7b). Note that the climatological seasonal cycle during the 5-day period shows a warming of 0.15 K, in contrast to cooling of 0.28 K during the NIW event in 2015 (Fig. 7b). This suggests that the vertical diffusive ML heat flux during the NIW event is able to reverse the climatological warming, leading instead to an intermittent ML cooling.

Furthermore, an increase in ML salinity is observed, which supports the elevated vertical diffusive ML salt flux inferred from the MSS observations. The divergence of the heat flux profile (Fig. 4c) suggests that the heat from the mixed layer is redistributed to between 20 and 40 m below the ML. Indeed, a warming in this depth layer of 0.2–0.3 °C during the 20-h sampling period is indicated by the hydrographic data collected with the MSS profiler (Supplementary Fig. 4). However, a clearer picture is hampered by possible contributions of horizontal advection to the heat (and freshwater) budget below the ML.

Additional support for a relationship between strong ML cooling events and NIEs can be obtained from the ePIRATA heat budget residual ($R$) time series. Converting the residuals to daily temperature changes $\left( \frac{\partial T}{\partial t} = \frac{R}{\rho c_p h} \right)$ shows that their distribution is skewed toward large negative events (skewness $-0.48$). The seasonal frequency distribution of elevated cooling events $\left( \frac{\partial T}{\partial t} < -0.2 \, °C \, day^{-1}, \text{Fig. 7d} \right)$ peaks in July through September, which is in agreement with the seasonal maximum of the monthly prevalence of IA (Fig. 6b, c) and the seasonal minimum of MLD (Fig. 6d). Note that the currently available 12-year ePIRATA heat budget residual time series (2006–2017) exhibits a total of 30 elevated cooling events in the month of August,

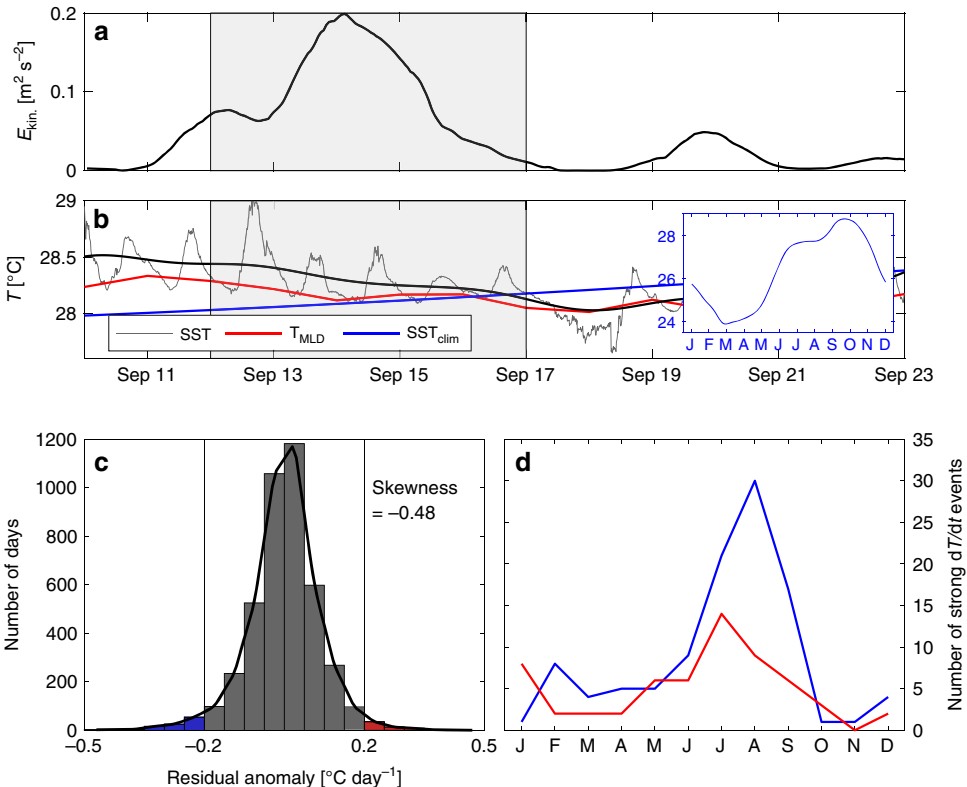

**Fig. 7 Temperature evolution and statistics of daily changes at 11.5°N, 23°W. a** Total kinetic energy derived from the slab model around the near inertial wave (NIW) event (similar to Fig. 5d; gray shading indicates the time defined around the NIW event for the local balance). **b** Sea surface temperature (SST) time series from the PIRATA buoy at 11.5°N, 23°W around the NIW event (gray line) together with the low-pass filtered SST time series using a 3-day Hanning window (black line), the ePIRATA mixed layer (ML) temperature (red line), and the SST climatology (blue line). The SST climatology for the full year is shown in the insertion. **c** Distribution of daily temperature change anomalies ($\frac{\partial T}{\partial t}$) converted from ePIRATA heat budget residuals (June 2006–December 2017). **d** Seasonal cycle of strong ML cooling and warming events ($\frac{\partial T}{\partial t} < -0.2$ °C day$^{-1}$ in blue and $\frac{\partial T}{\partial t} > 0.2$ °C day$^{-1}$ in red) from the ePIRATA daily temperature change anomalies used in **c**.

corresponding to 2.5 events in an average August. The seasonal cycle of strong warming roughly follows that of cooling, but the absolute changes in temperature are much smaller (red line in Fig. 7d).

The simulation of the ML response to the wind forcing at the PIRATA buoy at 11.5°N, 23°W with a linear slab model showed that the ML velocities during this event are well reproduced using a damping time scale of 5 days. A linear filter applied to wind stress data that is identical to a slab model with constant ML depth and density was used to determine the seasonal intermittency of such events at this location. The seasonal distribution of the monthly prevalence of NIEs shows that strong IA is evident throughout the year with a peak in July–September (JAS; Fig. 6b). For the threshold of $\alpha = 2$, a total of only 16.4 days per year are detected with the possibility of strong NIEs to occur, which is only 4.5% of the year. This emphasizes the intermittency of these events and reconciles the MSS observations described here with the previous seven cruises, during which significantly lower vertical diffusive ML heat losses were observed (Fig. 2b). In addition, strong near-inertial ML velocities are expected to be more common when elevated monthly prevalence of NIEs coincides with a shallow MLD. This is the case at 11.5°N, 23°W during JAS, when the heat budget residual, an estimate of turbulent cooling, is at its maximum (Fig. 2b). This strongly suggests the importance of NIW-induced mixing for the seasonal ML heat budget at this location.

The wind event in September 2015, which locally triggered the elevated ML velocities, is characterized by clockwise (anticyclonic) rotating moderate winds with a period close to the local

inertial period. Analysis of IA showed that comparable wind events occur throughout the year, but most frequently during JAS (Fig. 6b). During this season, synoptic-scale disturbances with wavelengths of 2000–4000 km and time scales of 2–10 days form in the easterlies over tropical northern Africa and are known as African Easterly Waves (AEWs)[54–57]. These waves are formed through baroclinic instabilities in the vicinity of the African Easterly Jet centered at around 15°N and travel westward with a mean speed of about 7–9 m s$^{-1}$ [58]. They have been recognized to contribute to tropical cyclogenesis[56,59], and their Pacific equivalents have been shown to force NIWs in the ML[60] with strong resonant forcing of near-inertial currents in the vicinity of the mean track of Pacific Easterly Waves.

For the tropical North Atlantic, within the lower troposphere (925 hPa) AEW activity over the ocean is enhanced between 5°N and 15°N from the African coast to about 40°W (Fig. 13 in[61]). Furthermore, there is a sharp drop-off in the spectral density of the wind variability towards higher frequencies, revealing that there is practically no AEW energy available at periods shorter than 2 days[62]. As a NIW response in the ocean requires atmospheric forcing that includes the frequency of the local Coriolis parameter $f$, an enhanced NIW response is particularly expected at lower latitudes within the ETNA, where there is considerable overlap between the dominant frequencies of the atmospheric forcing and the preferred frequency response of the ML[60]. The inertial periods in the main AEW track (5°N–15°N) are 4.8–5.7/2.4–2.9/1.6–1.9 days at 5°N/10°N/15°N, respectively.

In fact, in the context of this study, we find elevated monthly prevalence of NIEs calculated from CCMP winds during the main

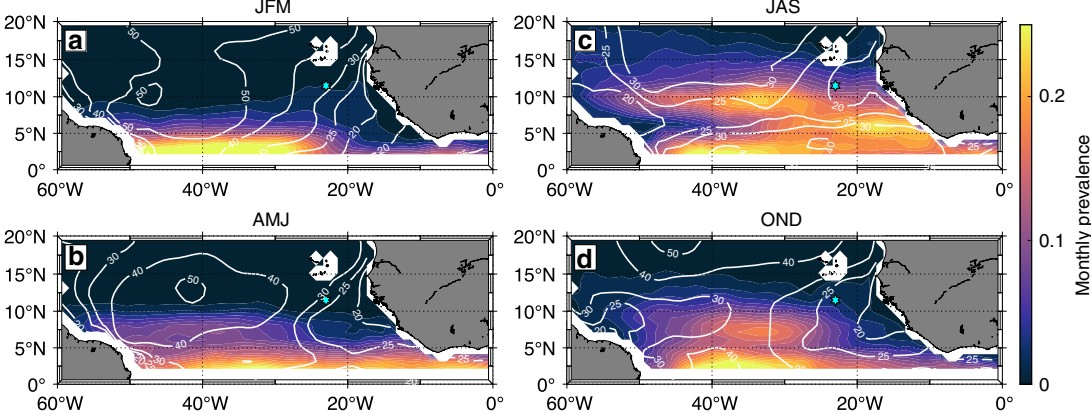

**Fig. 8 Seasonal climatology of inertial activity in the tropical North Atlantic.** The monthly prevalence of the inertial activity (IA) index crossing the threshold for $\alpha = 2$ is averaged over the respective seasons. White contours represent the seasonal MIMOC mixed layer depth climatology[81]. The cyan star marks the position of the PIRATA buoy at 11.5°N, 23°W. **a** JFM: January–March, **b** AMJ: April–June, **c** JAS: July–September, and **d** OND: October–December.

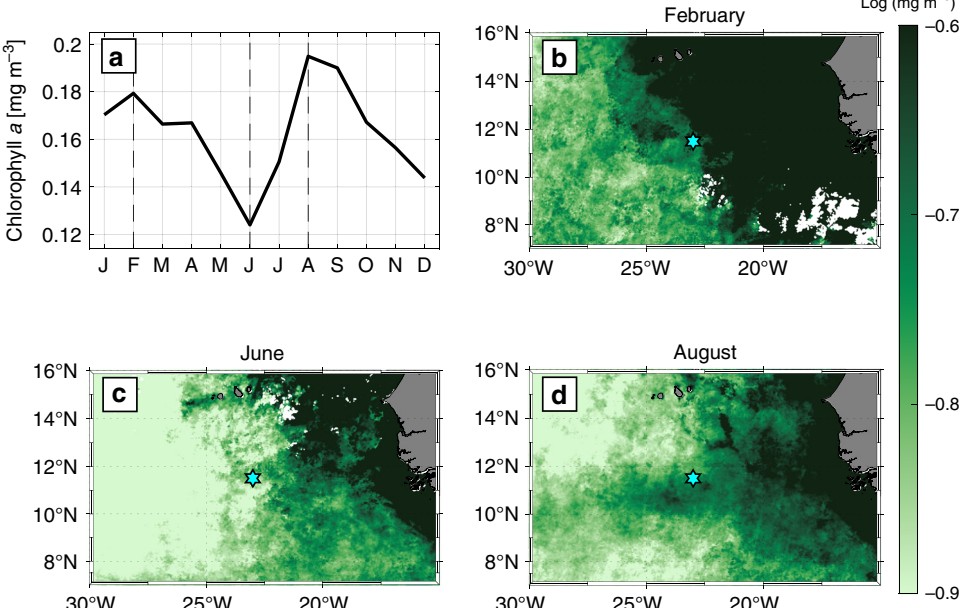

**Fig. 9 Seasonal cycle of chlorophyll *a* in the eastern tropical North Atlantic. a** Seasonal cycle of chlorophyll *a* averaged over a box in the vicinity of the PIRATA buoy (9°N–11.5°N, 23–26°W) for the years 2002–2012. **b–d** Monthly mean (2002–2012) chlorophyll *a* distribution for the respective months February, June, and August. The PIRATA buoy position is indicated as cyan star.

AEW season in JAS between about 5°N and 11°N stretching from about 20°W to 40°W (Fig. 8). During the other seasons the monthly prevalence of NIEs is clearly reduced. As the peak in monthly prevalence of NIEs during JAS in the ETNA coincides with a shallow MLD (Fig. 8c), vertical mixing intensity is most likely enhanced within this area at that time. Likewise, elevated latent heat fluxes associated with AEWs might contribute to ML heat loss. Despite existing knowledge about the presence and seasonal variation of AEWs and their potential to trigger NIWs in the surface ocean, their importance for the seasonal ML heat budget in the ETNA has previously not been pointed out.

In addition to redistributing heat and freshwater between the ML and the stratified ocean below, elevated vertical mixing also enhances the upward flux of nutrients to the euphotic zone, which could, among other factors, potentially impact phytoplankton blooms and the chlorophyll *a* distribution[12]. Indeed, a seasonal cycle in the chlorophyll *a* concentration in the vicinity of

the PIRATA buoy (Fig. 9a) is evident. However, in addition to a concentration maximum during JAS, a second maximum during JFM is also apparent, which does not coincide with elevated IA. Inspection of the spatial patterns during characteristic months within the seasonal cycle (Fig. 9b–d) shows that the elevated chlorophyll *a* concentrations at the beginning of the year are related to the extensive coastal upwelling present during boreal spring (e.g. ref. [38]). Nevertheless, as suggested by the chlorophyll *a* distribution during JAS (Fig. 10), a band-like structure with enhanced concentrations between 6°N and 12°N stretches from the coastal region into the open ocean to 35°W–40°W. This spatial structure at least partially coincides with regions of both elevated monthly prevalence of NIEs and shallow MLDs (Figs. 8, 10). Note though that the spatial overlap is not completely clear for some of the respective years. The averaging interval, the lack of interannual MLD maps and possible spatial differences in the nutrient reservoir below the ML can mask the direct relation of

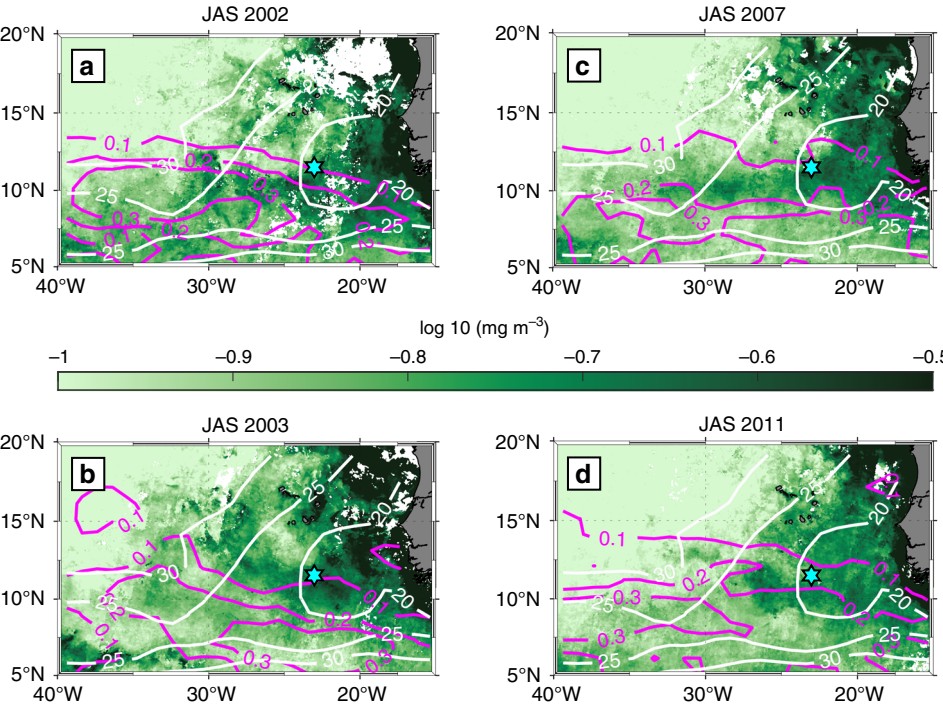

**Fig. 10 Boreal summer chlorophyll *a* distribution for exemplary years.** July–September (JAS) chlorophyll *a* distribution during **a** 2002, **b** 2003, **c** 2007, and **d** 2011. Magenta contours show the seasonal prevalence of near inertial wave events (as in Fig. 8c), but for the respective years; white contours show the corresponding mixed layer depth from the MIMOC climatology as in Fig. 8c (the latter without interannual variations). The cyan star marks the position of the PIRATA buoy at 11.5°N, 23°W.

these short-term events. Previously, phytoplankton blooms within this region, sometimes referred to as Guinea Dome, were thought to be associated with the large-scale cyclonic wind field. The results presented here suggest that in addition to the large-scale cyclonic wind field, wind variability with a frequency close to *f* introduced by AEWs is likely an additional cause for the observed phytoplankton blooms and needs to be considered when analyzing the seasonal cycle of chlorophyll *a* and associated biological productivity.

The presented results support the idea that elevated vertical mixing during NIEs is shear driven and hence *Ri*-number dependent. Diffusive ML heat fluxes calculated with a 1-D K-profile parametrization (KPP) ML model[63] gave results that were generally consistent with those obtained from the MSS measurements. However, the MSS dataset consisting of only 25 profiles over a time period of 20 h is not sufficient to test the model skill of a 1-D ML model like KPP. Hence, further detailed studies based on a comprehensive set of observations are necessary in order to constrain parameterizations for NIW-induced vertical mixing, which was found to be one crucial aspect in improving the quality of SST variability, ITCZ representation and thereby climate and rainfall variability in climate models[8], as well as the representation of biogeochemical variables in coupled biogeochemical ocean circulation models of the tropical Atlantic.

## Methods

**PIRATA and ePIRATA data**. Hourly wind (at the height of 4 m) and ML velocity (at the depth of 12 m) observations as well as 10-min SST observations from the PIRATA buoy at 11.5°N, 23°W are used in this study. In addition, the individual daily ePIRATA ML heat flux contributions at the same location from 2006 to 2017 are used to calculate the monthly terms of the budget shown in Fig. 2b. Daily averaged atmospheric measurements from the buoy include air temperature, relative humidity, wind velocity, and downward surface longwave and shortwave radiation. Oceanic measurements consist of temperature at 5–20 m vertical spacing, salinity at 10–40 m resolution, and horizontal velocity at a depth of 10 m. In ePIRATA, strict quality control is applied to all of these raw buoy measurements. All measurements with known biases are either corrected or eliminated and

replaced with data from other sources such as satellites and reanalyses[18,64]. Ocean temperature and salinity are mapped to a uniform 5 m vertical grid using information from nearby Argo profiles. This represents a significant improvement over simpler linear interpolation techniques that have been used commonly in the past. MLD is calculated using the criterion of a 0.12 kg m$^{-3}$ increase in density relative to the density at 1 m depth. This criterion was found to be a good balance between reducing noise in the daily MLD, which is larger for a smaller criterion, and increasing the agreement between SST and ML temperature, which is greater for a smaller criterion. To calculate horizontal velocity averaged vertically in the ML, the velocity measurements at a depth of 10 m from the mooring are adjusted according to a lookup table for ML velocity (based on the monthly Ocean Reanalysis System 4 (ORAS4) data for 2000–2014[65]) as a function of 10 m velocity, MLD, and time of year (see ref. [18] for details). Horizontal gradients of ML temperature are estimated using centered 1° differences from gridded satellite microwave SST. Advection terms are estimated as the product of ocean density, heat capacity, daily MLD, ML velocity, and horizontal ML temperature gradients. The amount of shortwave radiation that penetrates through the base of the ML is calculated using an algorithm that depends on surface chlorophyll *a* concentration[66,67]. For further details on the calculation of ML heat balance terms for ePIRATA the reader is referred to ref. [18].

The monthly climatological uncertainty for the individual terms of ePIRATA is calculated as a random error assuming a decorrelation time scale of 3 days: The monthly climatological error ($x_{b/monthly}$) for the heat budget term b associated with daily error estimates $x_b$ is given by $x_{b/monthly} = \frac{1}{n}\sqrt{3\sum x_b^2}$, where *n* is the number of available daily error estimates for the term b used for the climatological monthly mean. The 3-day decorrelation is based on the zero-crossing of the daily-lagged autocorrelation of the mixed layer heat budget terms[18]. The climatological monthly error of the residual term is then calculated as the square root of the sum of the squares of the monthly climatological heat storage rate, advection, and surface heat fluxes. Note that this error of the residual does not include bias estimates or systematic error estimates. However, potential biases were estimated to be on the order of maximum 20 W m$^{-2}$, when the ML is thin[68].

**Shipboard observations**. Turbulence and hydrographic data used in this study was obtained from a loosely tethered free-falling microstructure (MSS, Sea & Sun Technology, Germany) profiler with three shear sensors, a fast-responding temperature sensor and conductivity–temperature–depth (CTD) sensors. Additionally, turbulence and hydrographic data from a MicroRider (MR, Rockland Scientific International, Canada) mounted onto a Teledyne Webb Research G1 deep glider (short version) equipped with an unpumped CTD, two shear and two fast thermistor sensors is used. Velocity data was collected by a vmADCP operating at a frequency of 75 kHz and a bin size of 8 m. During post-processing, the single-ping

vmADCP data were corrected for misalignment angle and amplitude factor using water track calibration[69]. Buoyancy frequency $N$ for determining $Ri$-numbers was calculated from the hydrographic data by averaging salinity and temperature using overlapping 16 m triangular bins corresponding to the 8-m depth interval available from the vmADCP velocity observations.

During post-processing of the microstructure data, estimates of $\varepsilon$ were derived by integrating shear spectra determined from one-second time series of shear data (1024 data points) assuming isotropic turbulence while correcting for the shear probe's spatial response[19]. Variance of the unresolved portions of the turbulent shear spectrum was accounted for using the empirical Nasmyth spectrum[70]. While sinking velocity of the MSS profiler was determined by the change of pressure with respect to time, a dynamic glider flight model[71] was used for the microrider to determine the speed of flow past the microstructure sensors.

Turbulent eddy diffusivities for mass were estimated from $\varepsilon$ using the dissipation method of Osborn[72]: $K_\rho \leq \Gamma \varepsilon N^{-2}$. For mixing efficiency $\Gamma$, a constant value of 0.2 was used (for details see ref. [19]). Furthermore, the vertical diffusive heat flux is calculated via $J_{\text{heat}} = -\rho c_p K_\rho \frac{\partial T}{\partial z}$ with $\rho$ and $c_p$ being the density and specific heat capacity of seawater and $\frac{\partial T}{\partial z}$ the vertical temperature gradient derived from an instantaneous temperature profile, in which temperature values are sorted in descending order.

**Satellite data**. The Cross-Calibrated Multi-Platform (CCMP)-gridded surface vector wind product is used, and provides wind estimates at 6 h resolution on a 0.25° horizontal grid[50,51]. The CCMP winds have been shown to better capture synoptic events that are missing in, e.g., reanalysis products[50]. Note that buoy wind data is also assimilated in CCMP.

Chlorophyll a data is obtained from the medium resolution imaging spectrometer (MERIS). In this study, the mapped (4-km) monthly mean data is used.

**Linear filter**. To further evaluate the observations, a linear filter equivalent to the slab ocean model described by Pollard and Millard[35], but with constant ML depth and density is used to derive the quantity termed inertial activity (IA) inherent in the wind forcing at 11.5°N, 23°W and in the whole tropical North Atlantic.

The slab model proposed by Pollard and Millard[35] is based on the local balance of the wind stress, the Coriolis force, and a linear damping term:

$$\frac{\partial}{\partial t}(u, v) + f(-v, u) = \frac{1}{\rho h}(\tau^x, \tau^y) - r(u, v),\tag{3}$$

where $u$ and $v$ are the zonal and meridional velocity components in the ML, $f$ is the Coriolis parameter, $\rho$ and $h$ are ML density and depth, $\tau^x$ and $\tau^y$ the zonal and meridional wind stress component and $r$, a linear damping parameter (see Supplementary Note for details on the choice of this quantity). Using $q = u + iv$ and $F = \tau^x + i\tau^y$, the slab model can be written as follows:

$$\frac{\partial}{\partial t}q + if q = \frac{1}{\rho h}F - rq\tag{4}$$

We discretize this using a leap-frog scheme for the Coriolis term and an Euler-backward scheme for the damping and the wind forcing. With the time since initialization of the model $t = dt \cdot l$ and $dt$ the time step one obtains

$$\frac{q_l - q_{l-2}}{2\,dt} = -if q_{l-1} - rq_{l-2} + \frac{1}{\rho h}F_{l-2}\tag{5}$$

or

$$q_l = -2\,i\,dt\,f\,q_{l-1} + (1 - 2\,dt\,r)q_{l-2} + \frac{2\,dt}{\rho h}F_{l-2}\tag{6}$$

As Plueddemann and Farrar[46] demonstrate, the simple slab model can represent the near-inertial currents in the ML for several inertial cycles, but it does not capture the full kinetic-energy balance and hence cannot be used to estimate the energy available for mixing at the base of the ML. Using a more sophisticated model as Plueddemann and Farrar[46] propose for the whole ETNA requires knowledge of the near-surface stratification at greater spatial and temporal detail than is available. What can be achieved, however, is an estimate of the prevalence of strong inertial events in the region that is based only on the wind-stress signal that avoids making detailed assumptions about the state of the ocean. This was done by using a constant product of ML density and depth ($\rho h = $ const.) in the slab model equations. We then used a threshold criterion to classify the presence or absence of an inertial event for any given point in time and space.

The above formulation of the slab model can be understood as a linear infinite-impulse-response (IIR) filter for the wind-stress $F$. Following Press et al. [73], a linear filter can be expressed as

$$y_l = \sum_{k=0}^{M} c_k x_{l-k} + \sum_{j=1}^{N} d_j y_{l-j}\tag{7}$$

$y_l$ and $x_l$ denote the output and the input at time step $l$. If $N > 0$, the $l$th state of the filter depends on $N$ previous states. Filters with $N > 0$ are, at least in principle, capable of infinitely long impulse response.

Setting $q = y$, $F = x$, we can express the slab model as a linear filter with

$$M = 2;\ c_0 = c_1 = 0;\ c_2 = 2\,dt/(\rho h);\ N = 2;\ d_1 = -2i\,dt\,f;\ d_2 = 1 - 2\,dt\,r$$

As long as $dt\,f \ll 1$ and $0 < dt\,r \ll 1$, the filter is stable. Noting that the filter outputs $q_0$, $q_1$, ... are linear functions of all the wind stress time steps $F_0$, $F_1$, ..., we can redefine $F$ to $F/(\rho h)$ or set $\rho h = 1$ and are left with a linear filter that we use as a metric for the capability of the wind stress to create IA in the ocean.

Eliminating the mass of the water column $\rho h$ yields the following interpretation: The state of the filter $q_l$ determines the ability of the wind stress to cause inertial currents in the ML. The amplitude of the resulting ML velocities will, to leading order, be scaled by the inverse of $\rho h$ and hence be weaker the deeper the prevailing ML.

To eliminate the slowly varying Ekman response contained in $q_l$, we apply a 30-day high-pass filter to the $q_l$ time series. Note that implementing a filter according to the formulation of D'Asaro[49] explicitly excluding the Ekman response and applying this to the PIRATA wind data does not alter the resulting estimates of IA.

The filter integrations were performed using Python v3, xarray[74,75], Dask[76], Numpy[77], and Numba[78] on the NEC Linux Cluster at CAU Kiel. The software was maintained using Anaconda.

**ML mixing scheme**. The one-dimensional ML model used here is conceptually similar to that of Kraus and Turner[45]. In general, we use the implementation as described in ref. [79]. ML deepening due to wind stirring is calculated from the turbulent kinetic energy equation as follows:

$$-\frac{g}{2\rho_0}h\Delta\rho w_{\text{e}} = mU^3,\tag{8}$$

where $g$ is the acceleration of gravity, $\rho_0$ the ML density, $w_{\text{e}}$ the entrainment velocity of the ML, $h$ the MLD, $\Delta\rho$ the density step at the base of the ML, $U$ the wind speed. The wind mixing energy depends on the wind speed and is assumed to be dissipated exponentially with depth:

$$m = \rho_{\text{a}}c_{\text{D}}f_{\text{w}}e^{-\frac{h}{H_0}},\tag{9}$$

where $\rho_{\text{a}}$ is the air density, and $c_{\text{D}}$ the drag coefficient. The efficiency of wind mixing $f_{\text{w}}$ is taken to be $1.5 \times 10^{-3}$ with a dissipation scale $H_0 = 50\,\text{m}$[80]. For the estimates of this study $\rho_{\text{a}}$ is set to $1.2\,\text{kg m}^{-3}$, $c_{\text{D}} = (0.61 + 0.065\,U) \times 10^{-3}$ and $\rho_0 = 1023\,\text{kg m}^{-3}$.

The model was initialized with microstructure temperature profiles and evaluated using wind speeds from the PIRATA buoy at 11.5°N, 23°W. Over the period of 5 days a ML deepening of 1.2–1.5 m was obtained. From the associated ML heat content change we derived a heat flux across the base of the ML of 18–22 $\text{W m}^{-2}$.

## Data availability

PIRATA data are available at https://www.pmel.noaa.gov. The ePIRATA data are available at https://www.aoml.noaa.gov/phod/epirata/. For the shipboard observations, velocity data from the vmADCP are available under https://doi.org/10.1594/PANGAEA.877375, microstructure profiles for the different cruises are available under the following DOIs: M80/2 (December 2009, 9 profiles): https://doi.org/10.1594/PANGAEA.819220; M83 (October 2010, 9 profiles): https://doi.org/10.1594/PANGAEA.819235; M105/M106 (MR, March 2014, 146 profiles): https://doi.org/pangaea.de/10.1594/PANGAEA.920597; M106 (April 2014, 7 profiles): https://doi.org/pangaea.de/10.1594/PANGAEA.920591; M119 (September 2015, 25 profiles): https://doi.org/pangaea.de/10.1594/PANGAEA.920592; M130 (September 2016, 6 profiles): https://doi.org/10.1594/PANGAEA.918280; MSM22 (November 2012, 4 profiles): https://doi.org/10.1594/PANGAEA.846946; MSM23 (December 2012, 6 profiles): https://doi.org/10.1594/PANGAEA.858700. Satellite data are available at https://www.remss.com for CCMP winds and at https://oceancolor.gsfc.nasa.gov for MERIS chlorophyll a data. All data used in this study is publicly available as stated above and is also available from the corresponding author upon reasonable request.

## Code availability

All necessary code for the data analysis and preparation of the figures of this manuscript is freely available at https://doi.org/10.5281/zenodo.3340869.

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

## Acknowledgements

The authors acknowledge financial support from the Deutsche Forschungsgemeinschaft as part of the Sonderforschungsbereich 754 (SFB754) and the project FOR1740, the German Federal Ministry of Education and Research as part of the cooperative project RACE (03F0605B, 03F0824C), the European Union's Horizon 2020 research and innovation program under grant agreement 817578 TRIATLAS project, and NOAA/AOML (G.R.F.). The authors acknowledge the use of data from CCMP Version-2.0 vector wind analyses produced by Remote Sensing Systems, the use of ocean color data (chlorophyll) provided by the Ocean Biology Processing Group from the NASA Goddard Space and Flight center and the PIRATA program for free data access. Many thanks to the captains and crews of RV Maria S. Merian and RV Meteor as well as the technical group of GEOMAR for their help with the fieldwork.

## Author contributions

R.H. led and drafted the manuscript. M.D., T.F., and R.H. processed the microstructure data. W.R. set up and performed the slab model and filter integrations. G.R.F. evoked the motivation for this study providing the ePIRATA heat budget terms. F.S. analyzed the satellite data. P.B. led the RV Meteor cruise M119 in September 2015 and coordinated the measurements. All authors contributed to the interpretation of the results and provided substantial input to the manuscript.

## Competing interests

The authors declare no competing interests.
