## [Peer Review File · Nature Communications]

Peer Review File - Reviewers' comments first round:

Reviewer #1 (Remarks to the Author):

Review of manuscript entitled "Near inertial wave induced mixing in the tropical Atlantic Ocean", by R. Hummels, M. Dengler, W. Rath, G. R. Foltz, F. Schütte, T. Fischer and P. Brandt.

This paper presents observational evidence of a near-inertial wave (NIW) induced strong diapycnal mixing event in the tropical Atlantic. The authors also provide arguments by which events such as this vigorous mixing episode have implications for the ML heat balance and the vertical transport of nutrients by calculating an index for inertial activity (IA) in the tropical Atlantic. The subject of the paper is relevant, and the conclusions claimed by the authors are backed by observational evidence. However, I think the message of the paper need to be conveyed more clearly by tidying the presentation, and particularly by taking special care that all methodological aspects are unambiguously described. To be published, I would recommend major changes to be done. I have some constructive comments/feedbacks which I hope they help to improve the manuscript.

General comments:

- The term 'turbulent heat flux' is used in the manuscript to refer to diapycnal mixing at the base of the mixed layer (L66, L73, L75, L319, L321). However, the term turbulent heat flux is typically used in the bibliography to refer to the sum of air-sea latent and sensible heat fluxes. Thus, I might suggest to find an alternative term (e.g. turbulent heat flux at the ML base; diapycnally-induced turbulent heat flux, ...).

- To highlight more clearly the importance of NIW-induced diapycnal mixing at the ML base for the ML heat budget and the vertical transport of salinity and nutrients in the tropical Atlantic, I would recommend to make the appropriate changes to move the content of lines 93-99 after lines 84 (trying to avoid to split the argumentation). In this way, besides, the main aim of the work and the principal conclusions and the sections where they can be found could be described one after the other (L85-L92, L99-L102). Additionally, in order to place the study in a broad context and further highlight its climatic implications, it could be interesting to briefly mention part of the interesting information and nicely presented in the Introduction of Hummels et al. 2013 and in the discussion of the present manuscript.

- To facilitate the comprehension of the different analyses carried out, instead of simply indicating '(see Supplementary for details)' (L109; L133; L142; L218; L266; L272), it would be recommendable to add the specific section in the Suppl. Inform. where the details are described (e.g. 'See section C in the Suppl. Inform. for details'). In this way, the reader do not need to search throughout the Supplementary for the pertinent information.

- With similar purpose, to include in the Supplementary a brief description of all the data used in the manuscript will be desirable. For example, there is no description of wind, ML velocity observations, MLD or ML heat flux contributions data used. Only its reference is included (Folz et al. 2018). It is convenient to have this reference for readers interested in very specific details. However, more general information regarding e.g. how the MLD is computed (threshold method?), the frequency of the measurements (ML heat flux contributions used in the manuscript are monthly, weekly averages ???) would be useful to have briefly summarized in the Supplementary.

Regarding the use of MLD data, there are several issues that need to be resolved:

- (1) It is indicated that 'a smoothed seasonal cycle of the MLD is used' (L219). However, according to the information displayed in Fig. S1 rather than a smoothed seasonal cycle' it seems that a climatological seasonal cycle (it is exactly the same climatological seasonal cycle every year) is used. If it is exactly the same seasonal cycle, the occurrence and intensity of the NIW event depends completely on the wind forcing.

- (2) On the other hand, later in the manuscript when calculating the IA index (L257-L292) it is indicated that 'To disentangle the roles of these two factors, their respective seasonal cycles are investigated separately' (L261) but it seems that only the magnitude of NIW induced currents are evaluated ('The magnitude of NIW induced currents driven solely by variations in wind forcing is

obtained with a linear filter for wind observations based on the slab model equations (with constant MLD)'). Which is this constant value of the MLD? With which MLD value is the IA shown in Figure 7 calculated? Is it the mean value of the climatological seasonal cycle at the PIRATA buoy? Is it representative of the whole area or alternatively the average value at each location from MIMOC MLD climatology is used?

Regarding the use of the ML heat flux contributions, it is important to clarify somewhere in the manuscript (main text or Supplementary) that the short wave radiation heat flux shown in Fig. 1 and discussed in the manuscript is the solar radiation heat flux absorbed in the ML as indicated in Foltz et al. 2018, That means, it is the short wave radiation heat flux minus what is usually known as penetrative heat flux (Q_{pen} , the amount of short wave radiation that penetrates through the base of the ML) (see e.g: Cronin and McPhaden, 1997; Somavilla et al. 2013).

Without that explanation, one could consider as an alternative explanation to the high residual term during the summer months, the contribution of Q_{pen} not explicitly mentioned in the manuscript. It forces to search for this information in another manuscript (Foltz et al. 2018). That could be easily avoid making a brief mention to this fact in the present manuscript.

References:

- Cronin, M., McPhaden, M., 1997. The upper ocean heat balance in the western equatorial Pacific warm pool during September–December 1992. *J. Geophys. Res.* 102 (c4), 8533–8553
- Somavilla, R., González-Pola, C., Lavín, A., and Rodríguez, C. T and S variability in the south-eastern corner of the Bay of Biscay (NE Atlantic). *Journal of Marine Systems*. Vol. 109–110(Supplement), 2013. doi: 10.1016/j.jmarsys.2012.02.010.

These are just examples to illustrate the general concern of the reviewer about the need that all methodological aspects will be clearly and sufficiently described (see more examples in the specific comments below). So, please consider to include the necessary information about all the data used and make the description of the slab model, the linear filter derived from it, and of the IA index as clear as possible.

Specific comments:

- L31. 'Events such as this vigorous mixing event' seems more appropriate than 'Such vigorous mixing events'
- L47. Add references.
- L58. 'Hence, for the tropical Atlantic PIRATA mooring sites are preferred locations...'
- L72. '..could not be explained by diapycnal mixing related to the Equatorial Counter Current, ...'
- L108. It would be nice to add a first figure showing the location of PIRATA mooring site and additional data used in the tropical Atlantic (as in Fig. 1a in Hummels et al. 2013). It would also be appropriate to mention already that although the different data may seem to be too far from each other, its combined used is justified as discussed in section 3.3.
- L109. (See section A in the Suppl. for details)
- L123. '...0.6 ms⁻¹, (for details ... Hummels et al. 2013, 2014)'
- L127-130. What does it mean 'buoyancy frequency was calculated from least-squared fits to salinity and temperature'?
- L132. No details on the microstructure data is included in the Supplementary apart from general information about when they were collected and where they can be found.
- L152. Add estimation of the magnitude of the turbulent exchange at the base of the ML as done for the horizontal advection term.
- L164. (add reference to Fig. 3).
- L168-L190. Consider to re-order the information here. For non-experts in NIW, it would be useful to start with its definition (L183-L184, L186-L188), show that ADCP data confirm the occurrence of one episode, and then the observed elevated diapycnal mixing associated with this event.
- L177. 'Just below the ML,...' Which is the MLD in Fig. 2? Please indicate in the caption.
- L179. 'Despite the general/typical/... absence of strong ...'
- L209. 'Wind ...and application of derived linear filter' ??
- L211. 'is used (see section A in the Suppl. for details).'

L217-218. I may be wrong, so please change as appropriate but clarify 'We set it to 5 days according to decay scale obtained from wind power input at near inertial frequency analysis (see Fig. S2 and section C in the Suppl. for further information).'

L233. '..., it is appropriate to compare the vmADCP observations during September 2015 (Fig. 4c) to the output ...'

L242-L245. It would be helpful to highlight the period of interest in Fig. 4 with a shadow area.

L257-L259. The sentence is not very clear to me. Please consider revising.

L266. 'derived with the filter based on the slab model (see section A in the Suppl. for details on the model and derived filter; ...)

L271. '; for details on the filter based on the slab model and validation see ...)

L272. 'By design, the linear filter derived from the slab model ...'

L309-L323. If the event lasts 2.5 days, why is it defined for this additional estimate between the 13th and 16th of September 2015 (4 days)? How much is the SST drop and required diapycnal mixing at the ML base to explain it during 2.5 days instead of 4 days?

L315. Which is the frequency of ML heat flux contributions from PIRATA? Are these monthly, weekly...averages? If they are monthly averages, the values shown in Fig. 6 are interpolated values? Please clarify. In any case, since a constant value is shown in Figure 6c, to me there is no need to show this graphically. Just indicating its value could be enough. In Figure 6, as in Figure 4, it would be helpful to highlight the period of interest with a shadow area.

L355. References in order?

L378-384. The coincidence of areas of elevated Chl concentration and elevated monthly prevalence of NIEs is not clear to me from Fig. 7 and 8. If the NIW induced-diapycnal mixing events are so intermittent, wouldn't be worth to check for its correspondence during particular events as that described in the manuscript (e.g. using 8 days mean Chl data instead of monthly mean)? Is Chlorophyll distribution substantially different in the region during JFM, AMJ or OND?

Finally, as the reviewer is not an expert in tropical waves, I would be very grateful if the author could respond to a question that has arisen from the papers read for this review. What is the difference between Tropical Instability Waves (within the Equator) and NIW in tropical oceans? Do not Tropical Instability Waves occur at frequencies very close to the inertial frequency at these locations, as NIW further north or south? It is just the location and the interaction with the Equatorial Counter Current, which justify is different name but they have similar origin?

Reviewer #2 (Remarks to the Author):

General notes:

1. Need a more careful and thoughtful evaluation of the inertial shear event, including, but not limited to more careful treatment of Ri (get vertical scales of Sh and N the same).
2. *You were missing an important term in your ML heat budget!!* I *strongly* suggest you go back to the seminal refs. I mentioned there. The ref you mentioned either was wrong or made some assumptions that may not apply here.
3. You need a more careful and thorough discussion of evidence for cooling due to inertial shear-driven mixing. Where did the heat go that was mixed downwards? Do you see evidence of this? Again, mixing just moves heat around.

Other notes:

1. Why the IIR filter to look at inertial activity? Why not just run the PM70 model, which isn't terribly computationally intensive?
2. What about the role of u^3/kz in driving mixing when the ML is shallow and wind stress is significant? This needs to be compared to inertial kinetic energy or inertial shear.
3. I think Ekman currents are included in your filter---these will need to be extracted somehow if they are included. D'Asaro 1985 shows how to separate the steady and oscillating part of the PM70 solution. If they aren't, then you need to discuss why the filter just includes the inertial solution.
4. It has been known for a long time that inertial shear is important to SST: See Pollard, Rhines and Thompson (PRT 1972) or Crawford and Large 1996. I think there's a story worth publishing here---which I think is the story of the heat flux from microstructure obs (tho a dearth of obs) during the inertial event. How does this compare with climatology of entrainment heat fluxes? Or

with entrainment heat fluxes from model runs of this area? Are the 1-D models getting these right, or are they failing miserably? That's that ultimate question, I think. We know that inertial motions are really important to mixing, and we know they are very intermittent (see D'Asaro 1985 or Ocean Storms publications). Are we capturing their influence? Are there any implications of a ML budget that is often dominated by short-lived mixing events, like bloom formation and longevity? ...this would be good to include in the discussion.

Reviewer #3 (Remarks to the Author):

The authors report observations of strong turbulence mixing induced by storm-driven near inertial waves and the associated SST cooling in tropical Atlantic. A non-dimensional index for inertial activity (IA) is proposed to quantify the seasonal prevalence of NIW events. Near inertial event (NIE) is defined when $IA \geq 2$. Monthly prevalence of NIE reveals seasonal cycle, which is consistent with the unbalanced heat flux on the mooring. This paper is well written. It points out the importance of NIW on surface mixed layer heat budget, and the effect on air-sea flux, chlorophyll and nutrient flux. The seasonal cycle of IA is convincing. However, I don't find novel findings in this paper. NIW induced turbulence mixing and SST cooling in have been observed by many previous studies, some with model simulation comparison. The proposed IA neglects some key processes associated with NIWs and the associated mixing, such as NIW radiation (crudely included in the decaying rate, r), effects of stratification, spatial variation of mixed layer depth, and modulation by eddies, etc... At most, the IA produces a seasonal cycle of NIW effects which is to be expected. Therefore, I suggest the authors provide/clarify novel concepts of this manuscript, which I might have missed.

There are some missing details of analysis of surface mixed layer heat budget. How is the mixed layer defined? How is advective flux computed? At least 3 moorings will be needed for computing advective flux. Is penetrative heating included in the analysis?

Similarly, there are missing details of NIW analysis. What is time scale of satellite wind? Is it sufficient to resolve inertial rotation? The slab model does not include mesoscale effects on modulation NIW response. Also, as the authors point out, shear instability and turbulent mixing depend on stratification below the mixed layer too. The mixed layer is likely modulated by wind event, which is not included in the analysis. These effects should be discussed.

Some specific comments:

1. Line 63: 'Recently ...'. Microstructure measurements of turbulent flux below mixed layer have been taken for decades.
2. REMSS: temporal and spatial resolution.
3. Phase/frequency difference between of shipboard ADCP with slab model result: This might be due to the spatial variation of the wind or the translation of storm.
4. Fig 2b: Strong turbulence occurs at 14:00 below mixed layer. Strangely, the turbulence in the mixed layer is weak. This might suggest remotely generated NIW.
5. Fig. 2 g-h: Is there vertical propagation of NIWs?
6. Line 223-230: Please show the scatter plot of comparison between mooring and slab model results in supplementary material.
7. Line 250-252: The rotation of wind is important. Does the satellite wind resolves inertial rotation?
8. Line 269: What is the temporal and spatial resolution of CCMP?

**Paper NCOMMS-18-31254 „ Near inertial wave induced mixing in the tropical Atlantic Ocean” by Hummels, Dengler, Rath, Foltz, Schütte, Fischer and Brandt
Title changed to: “Surface cooling caused by rare but intense near-inertial wave induced mixing in the tropical Atlantic”**

Response to the Reviewers

We would like to thank all of the reviewers for their detailed and helpful comments to improve the manuscript.

The main changes we applied to our manuscript are a better and more detailed description of the mixed layer heat balance and its individual terms, as this point raised the concerns of all reviewers. The second point, which was not clear enough in the submitted version, was the reasoning for using the inertial activity (IA) index instead of the slab ocean model in order to determine the frequency of occurrence for strong near inertial wave events. We have rephrased this reasoning and the definition of the IA index and hope to have alleviated the reviewers' concerns. In general, we tried to rearrange some paragraphs and figure panels in order to clarify our reasoning and point out the important findings of this study. In addition to these changes with regards to the content, we have adjusted some of the formatting to follow the manuscript requirements of Nature Communications.

We are very grateful for the detailed comments of the reviewers on syntax and formal mistakes and hope that we have adequately addressed these issues within the resubmitted manuscript.

A detailed response to the reviewers' remarks is given in the following. The comments of the reviewers are given in bold letters, followed by our response. Blue color indicates text from the manuscript. Changes within the manuscript that are related directly to the comments below are highlighted with yellow color in the resubmitted manuscript.

Reviewer #1 (Comments to Author):

Major comments:

This paper presents observational evidence of a near-inertial wave (NIW) induced strong diapycnal mixing event in the tropical Atlantic. The authors also provide arguments by which events such as this vigorous mixing episode have implications for the ML heat balance and the vertical transport of nutrients by calculating an index for inertial activity (IA) in the tropical Atlantic. The subject of the paper is relevant, and the conclusions claimed by the authors are backed by observational evidence. However, I think the message of the paper need to be conveyed more clearly by tidying the presentation, and particularly by taking special care that all methodological aspects are unambiguously described. To be published, I would recommend major changes to be done. I have some constructive comments/feedbacks which I hope they help to improve the manuscript.

We would like to thank reviewer #1 for your critical review of our manuscript and your encouragement. After taking into account your remarks and suggestions, we believe

to have significantly improved the revised version of the manuscript and hope to have addressed all of the concerns raised in the review.

General comments:

- The term ‘turbulent heat flux’ is used in the manuscript to refer to diapycnal mixing at the base of the mixed layer (L66, L73, L75, L319, L321). However, the term turbulent heat flux is typically used in the bibliography to refer to the sum of air-sea latent and sensible heat fluxes. Thus, I might suggest to find an alternative term (e.g. turbulent heat flux at the ML base; diapycnally-induced turbulent heat flux, ...).

Thank you for this suggestion. We understand the ambiguity of the term and changed “turbulent heat flux” to “diapycnal heat flux” where appropriate.

- To highlight more clearly the importance of NIW-induced diapycnal mixing at the ML base for the ML heat budget and the vertical transport of salinity and nutrients in the tropical Atlantic, I would recommend to make the appropriate changes to move the content of lines 93-99 after lines 84 (trying to avoid to split the argumentation). In this way, besides, the main aim of the work and the principal conclusions and the sections where they can be found could be described one after the other (L85-L92, L99-L102). Additionally, in order to place the study in a broad context and further highlight its climatic implications, it could be interesting to briefly mention part of the interesting information and nicely presented in the Introduction of Hummels et al. 2013 and in the discussion of the present manuscript.

Indeed, we fully agree, thank you. We rearranged the paragraphs according to your suggestion and also added a paragraph on the climatic implications of an improved understanding of sea surface temperature (SST) variability in the beginning of the introduction.

- To facilitate the comprehension of the different analyses carried out, instead of simply indicating ‘(see Supplementary for details)’ (L109; L133; L142; L218; L266; L272), it would be recommendable to add the specific section in the Suppl. Inform. where the details are described (e.g. ‘See section C in the Suppl. Inform. for details’). In this way, the reader do not need to search throughout the Supplementary for the pertinent information.

Thank you, as suggested we renamed the different sections in the Supplementary Material and reference these sections in the main text accordingly. With this, the text now also fulfills the requirements of the journal.

- With similar purpose, to include in the Supplementary a brief description of all the data used in the manuscript will be desirable. For example, there is no description of wind, ML velocity observations, MLD or ML heat flux contributions data used. Only its reference is included (Folz et al. 2018). It is convenient to have this reference for readers interested in very specific details. However, more general information regarding e.g. how the MLD is computed (threshold method?), the frequency of the measurements (ML heat flux contributions used in the manuscript are monthly, weekly averages ???) would

be useful to have briefly summarized in the Supplementary.

Thank you for this suggestion. We have included more detailed information about the calculation of the individual terms of the ML heat budget in the ePIRATA data set in the Supplementary Material (A). In addition, we briefly explain what is provided from ePIRATA in the beginning of section 2.1.

Regarding the use of MLD data, there are several issues that need to be resolved:

(1) It is indicated that ‘a smoothed seasonal cycle of the MLD is used’ (L219). However, according to the information displayed in Fig. S1 rather than a smoothed seasonal cycle’ it seems that a climatological seasonal cycle (it is exactly the same climatological seasonal cycle every year) is used. If it is exactly the same seasonal cycle, the occurrence and intensity of the NIW event depends completely on the wind forcing.

This is true, we changed the “smoothed seasonal cycle” into “climatological seasonal cycle”.

(2) On the other hand, later in the manuscript when calculating the IA index (L257-L292) it is indicated that ‘To disentangle the roles of these two factors, their respective seasonal cycles are investigated separately’ (L261) but it seems that only the magnitude of NIW induced currents are evaluated (‘The magnitude of NIW induced currents driven solely by variations in wind forcing is obtained with a linear filter for wind observations based on the slab model equations (with constant MLD)’). Which is this constant value of the MLD? With which MLD value is the IA shown in Figure 7 calculated? Is it the mean value of the climatological seasonal cycle at the PIRATA buoy? Is it representative of the whole area or alternatively the average value at each location from MIMOC MLD climatology is used?

The motivation for using the IA index was rephrased. By using the linear filter instead of the slab model here, the seasonal cycle of the ability of the wind to force inertial motions is separated from the seasonal cycle of the MLD, which cannot be disentangled if the slab model itself is evaluated. This motivation together with a rephrased definition of the quantities can now be found in the Supplementary Methods (B) section. In addition, the explanation within the main manuscript was streamlined.

Regarding the use of the ML heat flux contributions, it is important to clarify somewhere in the manuscript (main text or Supplementary) that the short wave radiation heat flux shown in Fig. 1 and discussed in the manuscript is the solar radiation heat flux absorbed in the ML as indicated in Foltz et al. 2018. That means, it is the short wave radiation heat flux minus what is usually known as penetrative heat flux (Q_{pen} , the amount of short wave radiation that penetrates through the base of the ML) (see e.g: Cronin and McPhaden, 1997; Somavilla et al. 2013).

Without that explanation, one could consider as an alternative explanation to the high residual term during the summer months, the contribution of Q_{pen} not explicitly mentioned in the manuscript. It forces to search for this information in another manuscript (Foltz et al. 2018). That could be easily avoid making a

brief mention to this fact in the present manuscript.

Yes, thank you, we failed to state that directly in the manuscript. This is now stated directly in the beginning of section 2 to rule out this explanation for the residual. In addition to this point, in section 3 we have discussed the possible contribution of the entrainment heat flux to the residual as pointed out by Reviewer #2.

References:

Cronin, M., McPhaden, M., 1997. The upper ocean heat balance in the western equatorial pacific warm pool during September–December 1992. J. Geophys. Res. 102

(c4), 8533–8553

Somavilla, R., González-Pola, C., Lavín, A., and Rodriguez, C. T and S variability in the south-eastern corner of the Bay of Biscay (NE Atlantic). Journal of Marine Systems. Vol. 109-110(Supplement), 2013. doi: 10.1016/j.jmarsys.2012.02.010.

These are just examples to illustrate the general concern of the reviewer about the need that all methodological aspects will be clearly and sufficiently described (see more examples in the specific comments below). So, please consider to include the necessary information about all the data used and make the description of the slab model, the linear filter derived from it, and of the IA index as clear as possible.

We have rearranged the data and methods part according to suggestions made here and the additional requirements of the journal given by the editor. As stated above we have also rephrased the explanation of the linear filter used to create the IA index. We hope that we have successfully addressed the concerns about an adequate description of the individual parts are now ruled out. Thank you very much for making this point, we feel that this has strongly improved the clarity of the manuscript.

Specific comments:

L31. ‘Events such as this vigorous mixing event’ seems more appropriate than ‘Such vigorous mixing events’

We still feel that this is the right term to use in the abstract.

L47. Add references.

Added the references [*Gregg et al.*, 1985; *Niiler and Kraus*, 1977] at this place.

L58. ‘Hence, for the tropical Atlantic PIRATA mooring sites are preferred locations...’

Added “for the tropical Atlantic”.

L72. ‘..could not be explained by diapycnal mixing related to the Equatorial Counter Current, ...’

We understand that this sentence was not clear. The Equatorial Counter Current is not observed as far north as 11.5°N. We have now rephrased the sentence, pointing out that prior to this study turbulence observations did not show elevated diapycnal

heat fluxes at the ML base that would be required to explain the residual, while the present study shows that we have missed an important process namely NIW induced mixing. The sentence now reads: However, the sparse turbulence observations prior to this study indicated only weak diapycnal heat fluxes at the ML base, which were not sufficient to explain those residual [e.g. *Hummels et al.*, 2013].

L108. It would be nice to add a first figure showing the location of PIRATA mooring site and additional data used in the tropical Atlantic (as in Fig. 1a in Hummels et al. 2013). It would also be appropriate to mention already that although the different data may seem to be too far from each other, its combined used is justified as discussed in section 3.3.

We added a new figure (Fig. 1) showing a map that includes the cruise track and the location of the PIRATA site. We also mention in the introduction that the wind observations are taken from the nearby PIRATA buoy (Line 97: By combining these data with observations from a nearby PIRATA buoy (11.5°S, 23°W, Fig. 1))

L109. (See section A in the Suppl. for details)

Was adjusted, but is now in the Methods section placed at the end of the manuscript.

L123. '...0.6 ms⁻¹, (for details ... Hummels et al. 2013, 2014)'

Adjusted.

L127-130. What does it mean 'buoyancy frequency was calculated from least-squared fits to salinity and temperature'?

We rephrased this part:

The buoyancy frequency was calculated by averaging over a depth interval of 8 m to match the resolution of the vmADCP observations.

L132. No details on the microstructure data is included in the Supplementary apart from general information about when they were collected and where they can be found.

The overview over the microstructure data used is now included in the data availability section at the end of the manuscript and accordingly this sentence was removed.

L152. Add estimation of the magnitude of the turbulent exchange at the base of the ML as done for the horizontal advection term.

It is already mentioned. We state that at this point the diapycnal heat flux at the ML base is estimated as the residual (dashed orange line in Fig. 2b) and that this reaches a maximum of about 100 Wm⁻² in August. In addition, the magnitude of the diapycnal heat flux at the ML base inferred from microstructure observations is also explicitly stated.

L164. (add reference to Fig. 3).

Added also other Figure references.

L168-L190. Consider to re-order the information here. For non-experts in NIW, it would be useful to start with its definition (L183-L184, L186-L188), show that ADCP data confirm the occurrence of one episode, and then the observed elevated diapycnal mixing associated with this event.

Thank you. We have reordered the information here according to the suggestions, which lead to rephrasing of the entire section 2.2 and a rearrangement of the panels of Fig. 2 (now Fig. 3) accordingly.

L177. ‘Just below the ML,...’ Which is the MLD in Fig. 2? Please indicate in the caption.

Added in the caption.

L179. ‘Despite the general/typical/... absence of strong ...’

Added “general”.

L209. ‘Wind ...and application of derived linear filter’ ??

This section was separated into two parts. The first is entitled **2.3 Modelling the near-inertial ML velocities at the PIRATA site** and the second **2.4 Using the PIRATA-wind-forced slab model at the MSS site**

L211. ‘is used (see section A in the Suppl. for details).’

Added a precise reference to the details.

L217-218. I may be wrong, so please change as appropriate but clarify ‘We set it to 5 days according to decay scale obtained from wind power input at near inertial frequency analysis (see Fig. S2 and section C in the Suppl. for further information).’

Added the corresponding details.

L233. ‘..., it is appropriate to compare the vmADCP observations during September 2015 (Fig. 4c) to the output ...’

Changed.

L242-L245. It would be helpful to highlight the period of interest in Fig. 4 with a shadow area.

Added shading to Fig. 4 (now Fig 5).

L257-L259. The sentence is not very clear to me. Please consider revising.

We hope to have solved this issue by streamlining the explanation and definition of the filter.

L266. ‘derived with the filter based on the slab model (see section A in the Suppl. for details on the model and derived filter; ...)

Added the more precise reference.

L271. ‘; for details on the filter based on the slab model and validation see ...)

Added a more detailed reference.

L272. ‘By design, the linear filter derived from the slab model ...’

This sentence was removed.

L309-L323. If the event lasts 2.5 days, why is it defined for this additional estimate between the 13th and 16th of September 2015 (4 days)? How much is the SST drop and required diapycnal mixing at the ML base to explain it during 2.5 days instead of 4 days?

Averaging only over a short time period will be difficult as the diurnal cycle (as evident in the SST time series) will mask the cooling in the ML and is filtered out more efficiently using the whole event. Instead, according to the concerns of Reviewer #2 we added some other quantities to this figure and discuss the possible contribution of the entrainment heat flux for the budget.

L315. Which is the frequency of ML heat flux contributions from PIRATA? Are these monthly, weekly...averages? If they are monthly averages, the values shown in Fig. 6 are interpolated values? Please clarify.

ePIRATA provides daily averaged of the individual ML heat budget contributions. These values were averaged over the 4 days (now 5 days) of the event. As we adjusted Fig. 6 (now Fig. 7) according to the comments of Reviewer #2, we have removed this panel and only state the values in the text.

In any case, since a constant value is shown in Figure 6c, to me there is no need to show this graphically. Just indicating its value could be enough. In Figure 6, as in Figure 4, it would be helpful to highlight the period of interest with a shadow area.

Added shading and removed this panel to substitute with other quantities as indicated above.

L355. References in order?

The referencing was changed to the Nature Communications Style now.

L378-384. The coincidence of areas of elevated Chl concentration and elevated monthly prevalence of NIEs is not clear to me from Fig. 7 and 8. If the NIW induced-diapycnal mixing events are so intermittent, wouldn't be worth to check for its correspondence during particular events as that described in the manuscript (e.g. using 8 days mean Chl data instead of monthly mean)? Is Chlorophyll distribution substantially different in the region during JFM, AMJ or OND?

In general, it is rather difficult to get a clear picture from the chlorophyll satellite data as it is very patchy in this region due to significant cloud coverage. Even the monthly data can be rather patchy, so shorter time periods will be even worse. However, to clarify though that there is some seasonal variability in the chlorophyll concentration within this region, we added another figure, now Fig. 9. This plot shows a seasonal cycle in the vicinity of the PIRATA buoy location. The seasonal cycle shows two maxima, one as we expect during JAS, but also a secondary maximum at the beginning of the year, which does not coincide with elevated IA at this location. Looking at some of the monthly maps though it seems that the maximum in the beginning of the year is somehow related to the coastal upwelling, which is at its maximum during the beginning of the year. The other maximum found in August is

clearly separated from the coastal region and shows a band-like structure, which resembles the AEW track and the region of elevated IA and shallow MLDs. We added this explanation to the discussion and hope that this is now more convincing.

Finally, as the reviewer is not an expert in tropical waves, I would be very grateful if the author could respond to a question that has arisen from the papers read for this review. What is the difference between Tropical Instability Waves (within the Equator) and NIW in tropical oceans? Do not Tropical Instability Waves occur at frequencies very close to the inertial frequency at these locations, as NIW further north or south? It is just the location and the interaction with the Equatorial Counter Current, which justify is different name but they have similar origin?

Tropical Instability Waves are waves, which appear in the vicinity to the equator, in the Atlantic between 2°N – 5°N and west of 10°W. They have zonal scales of around 1100 km and a westward phase velocity of around 0.5 m/s [Legeckis and Reverdin, 1987]. TIWs are driven by barotropic and baroclinic instability of the energetic equatorial current system [von Schuckmann et al., 2008] and have typically periods between 20 and 50 days (period = 1/ frequency).

Near inertial waves (NIWs) are internal gravity waves with frequencies near the Coriolis frequency / inertial frequency. As explained in the present study NIWs are excited directly by wind variations close to this inertial period, which is about 5 days at 5°N, where TIWs are also found.

Hence, excitation mechanism, length scales and periods of these waves are different.

Reviewer #2 (Comments to Author):

This reviewer sent detailed comments directly indicated in the submitted pdf, which additionally have been considered. We attach a version of the manuscript including these comments together with our answers to the comments.

General notes:

1. Need a more careful and thoughtful evaluation of the inertial shear event, including, but not limited to more careful treatment of Ri (get vertical scales of Sh and N the same).

The vertical scales of Sh and N are the same. We apologize that this was not clear in the original version of the manuscript. We changed the description according to the Reviewer's comments stating that Sh and N were calculated on the same scales now and hope that we have satisfactorily addressed the concerns about this.

2. *You were missing an important term in your ML heat budget!!* I *strongly* suggest you go back to the seminal refs. I mentioned there. The ref you mentioned either was wrong or made some assumptions that may not apply here.

Thank you for this statement. We realized that we did not state some of the details here correctly.

The penetrative shortwave radiation is accounted for in ePIRATA and calculated

using an algorithm that depends on surface chlorophyll-a concentration. What we are showing and using from ePIRATA is therefore the absorbed shortwave radiation, which has been accordingly corrected in the text now.

The entrainment term is included in the residual term, which we did not mention explicitly before. Entrainment is not calculated directly, as it is difficult to estimate since it requires horizontal gradients of mixed layer velocity and mixed layer depth. However, it is very likely of minor importance compared to the diapycnal heat flux as we point out below.

To calculate the entrainment heat flux, referring to mixed layer deepening due to the wind stress only, we would have to evaluate the term: $w_{\text{entrain}}*(T-T_h)$, where T_h is temperature at the base of the ML. According to *Stevenson and Niiler* [1983], $w_{\text{entrain}} = dh/dt + \text{div}(hv)$. The latter term ($\text{div}(hv)$) is difficult to estimate from observations as pointed out above. However, we can estimate the ML deepening using the ML model of *Kraus and Turner* [1967], which is explained in the Supplementary Methods (C). This model was evaluated using temperature from the microstructure profiler and wind speeds from the PIRATA buoy at 11.5°N, 23°W. Over the period of 5 days a ML deepening of 1.2 -1.5 m was obtained. From the associated ML heat content change we derived an entrainment heat flux of 18 – 22 Wm^{-2} .

Compared to our diapycnal mixing term inferred from microstructure observations during the strong NIW event described here it is rather small ($20 / 370 = 5.4\%$) and only accounts for about 5 % of the observed diapycnal mixing.

3. You need a more careful and thorough discussion of evidence for cooling due to inertial shear-driven mixing. Where did the heat go that was mixed downwards? Do you see evidence of this? Again, mixing just moves heat around.

It is not easy to see this is the ePIRATA data, as we have a rather low vertical resolution of observations. The sensors of the PIRATA buoy are located at depth = [1 5 10 13 20 40 60 ..]m for temperature and [1 10 20 40 ..]m for salinity. ePIRATA increases the resolution to 5m adding information from Argo profiles in the region (see Supplementary Methods A for details).

However, we did add some additional panels to Fig. 6 (now Fig. 7), which show a cooling of ML temperature and an increase in salinity during the event. We also show the evolution of temperature and salinity between 25 and 30 m below the ML, where according to the profile of the diapycnal heat flux, heat should accumulate and salinity should decrease. There is some indication of a warming, however it is a rather abrupt feature. A slight decrease in salinity is also evident, however the vertical resolution of the observations is not optimal.

Other notes:

1. Why the IIR filter to look at inertial activity? Why not just run the PM70 model, which isn't terribly computationally intensive?

In Supplementary Methods B.2 we now explicitly state why we use the linear filter. The text reads:

As Plueddemann and Farrar [2006] demonstrate, the simple slab-ocean model can represent the near-inertial currents in the ML for several inertial cycles, but it does not capture the full kinetic-energy balance and hence cannot be used to estimate the

energy available for mixing at the base of the ML. Using a more sophisticated model as *Plueddemann and Farrar* [2006] propose for the whole eastern tropical North Atlantic (ETNA) requires knowledge of the near-surface stratification at greater spatial and temporal detail than is available. What can be achieved, however, is an estimate of the prevalence of strong inertial events in the region that is based only on the wind-stress signal and avoids assumptions about the state of the ocean. To this end, we rewrite the slab model equations as a linear filter that solely depends on the wind stress, and use a threshold criterion to classify the presence or absence of an inertial event for any given point in time and space.

2. What about the role of u^3/kz in driving mixing when the ML is shallow and wind stress is significant? This needs to be compared to inertial kinetic energy or inertial shear.

As described above we made a rough estimate of the entrainment term using the ML mixing model according to *Kraus and Turner* [1967], which is also based on the indicated u^3/kz scaling to get a measure for this term (see Supplementary Methods (C) for details).

As pointed out above over the period of 5 days a ML deepening of 1.2 -1.5 m was obtained. From the associated ML heat content change we derive an entrainment heat flux of 18 – 22 Wm^{-2} . If we now include the entrainment heat flux estimated with this model over the 5-day time period (20 Wm^{-2}), we still end up with a residual of 130 Wm^{-2} , which has to be explained now exclusively by the diapycnal heat flux at the ML base. Note that this is probably the lower limit for the diapycnal heat flux, as the entrainment term is likely smaller as it can be offset by a compensating convergence, which we are not able to estimate with the available observations.

3. I think Ekman currents are included in your filter---these will need to be extracted somehow if they are included. D'Asaro 1985 shows how to separate the steady and oscillating part of the PM70 solution. If they aren't, then you need to discuss why the filter just includes the inertial solution.

Thank you for this comment. We did not state this clearly in the text. The steady-state Ekman component is filtered out using a 30-day high-pass filter. This is now stated, where IA is defined and additionally explained in the Supplementary Methods (B.2).

4. It has been known for a long time that inertial shear is important to SST: See Pollard, Rhines and Thompson (PRT 1972) or Crawford and Large 1996. I think there's a story worth publishing here---which I think is the story of the heat flux from microstructure obs (tho a dearth of obs) during the inertial event. How does this compare with climatology of entrainment heat fluxes? Or with entrainment heat fluxes from model runs of this area?

We cannot use the *Kraus and Turner* [1967] mixing scheme to estimate the seasonal cycle of the entrainment heat flux based on the ML deepening as we do not have sufficient information about the seasonal variability of the density step the base of the ML, which is required to use this model. However, the entrainment heat flux should be largest, when the ML is shallowest within the seasonal cycle, as the wind only has to act on a smaller mass. The ML is shallowest during July to September, when we obtained an entrainment heat flux of 18 – 22 Wm^{-2} as pointed out above. Hence, it is

likely that during other seasons the entrainment heat flux is even smaller.

Are the 1-D models getting these right, or are they failing miserably? That's that ultimate question, I think. We know that inertial motions are really important to mixing, and we know they are very intermittent (see D'Asaro 1985 or Ocean Storms publications). Are we capturing their influence? Are there any implications of a ML budget that is often dominated by short-lived mixing events, like bloom formation and longevity? ...this would be good to include in the discussion.

ML mixing schemes without a Ri-number dependent term as *Kraus and Turner* [1967] are not able to reproduce the magnitude of mixing correctly as explained above. We have also tried to reproduce our event using another 1-D mixing model with a Ri-number dependent term, in this case the KPP mixing scheme as proposed by *Large et al.* [1994]. The initial values for the model were based on the observations of mooring surface fluxes and our profiles of $v(z)$, $T(z)$, and $S(z)$ from the shipboard campaign. The diapycnal heat flux around 5-10 m below the ML (around 17-20 m) from this 1-D Model were a factor of about 3-4 too high (reaching around 1000 Wm^{-2} at about 22 m) in the highly sheared layer compared to the diapycnal heat flux inferred from microstructure observations.

Zaron and Moum [2009] showed similar results with regards to KPP overestimating the diapycnal heat flux in their case in the equatorial Pacific region, where strong vertical mean shear is present and diapycnal heat fluxes are generally high. In addition to the factor of 4 overestimation of the heat flux, they found that the parametrized heat flux divergence had the incorrect sign when using KPP. In their study, they proposed a new parametrization for turbulent diffusivities, which still depends on the gradient Richardson number, but also accounts for the shear length scale and the background kinetic energy. In addition, it uses dimensional reasoning and includes a measure for the forcing of the turbulence.

Thus, despite the fact that the observed shear and shear from OGCMs are probably underestimated because of the limited vertical resolution, KPP still overestimates turbulent diffusivities and diapycnal heat fluxes in strongly sheared environments. This emphasizes that this parametrization scheme needs to be refined to adequately reproduce deep diurnal cycle mixing at the equator or NIW induced mixing as presented in this study.

In another study *Jochum et al.* [2013] aimed at including NIWs and their induced mixing at the ML base in a coupled climate model by increasing the temporal coupling between atmosphere and ocean in order to be able to allow for the existence of NIWs. Although this enabled them to include NIWs, the shear at the ML base had to be artificially tuned in order to make KPP able to produce adequate diapycnal heat fluxes at the ML base.

This again emphasizes that despite the fact that diapycnal mixing induced by NIWs is probably driven by shear instabilities as proposed within this study, available parametrization schemes still do not seem to be able to adequately reproduce the right magnitude of the diapycnal heat flux.

We have included essential parts of this discussion in the discussion section of the

manuscript.

Reviewer #3 (Remarks to the Author):

The authors report observations of strong turbulence mixing induced by storm-driven near inertial waves and the associated SST cooling in tropical Atlantic. A non-dimensional index for inertial activity (IA) is proposed to quantify the seasonal prevalence of NIW events. Near inertial event (NIE) is defined when $IA \geq 2$. Monthly prevalence of NIE reveals seasonal cycle, which is consistent with the unbalanced heat flux on the mooring. This paper is well written. It points out the importance of NIW on surface mixed layer heat budget, and the effect on air-sea flux, chlorophyll and nutrient flux. The seasonal cycle of IA is convincing.

Thank you for this assessment of the presented results.

However, I don't find novel findings in this paper. NIW induced turbulence mixing and SST cooling in have been observed by many previous studies, some with model simulation comparison. The proposed IA neglects some key processes associated with NIWs and the associated mixing, such as NIW radiation (crudely included in the decaying rate, r), effects of stratification, spatial variation of mixed layer depth, and modulation by eddies, etc... At most, the IA produces a seasonal cycle of NIW effects which is to be expected. Therefore, I suggest the authors provide/clarify novel concepts of this manuscript, which I might have missed.

The novel aspects of our study are:

- the near-surface turbulence observations away from ocean boundaries that show intense mixing and extremely elevated diapycnal heat fluxes during a NIW event. This NIW event was not excited by a storm or hurricane, but by moderate winds changing direction thereby producing a resonant response.
- We show resonant forcing of NIEs by African Easterly Waves in the tropical North Atlantic in a spatially limited region with a distinct seasonal cycle.
- We highlight parameters that are uniquely affected by the NIW events namely SST, SSS and chlorophyll *a*. Previous studies show that mixed-layer heat balances in the region of interest require an additional cooling source. Likewise, phytoplankton blooms which are often reported from this region (often named the Guinea Dome region) were previously thought to be driven by the large-scale wind stress curl. The results of this study show the importance of burst-like short term wind variability introduced by AEWs causing the observed phytoplankton blooms.

We have tried to highlight these aspects more clearly in the revised version.

Certainly, other aspects such as NIW radiation, effects of stratification, and modulation by mesoscale eddies are not considered here (spatial variation of mixed layer depth is considered, see e.g. Fig. 8). As far as mesoscale eddies are concerned former studies revealed that around 15% of the eastern tropical North Atlantic is covered with anticyclonic eddies [*Schütte et al.*, 2016]. Hence, in most other cases (in 85%) mesoscale features are irrelevant. Also, note that during our

shipboard campaign no anticyclonic eddies were present.

There are some missing details of analysis of surface mixed layer heat budget. How is the mixed layer defined? How is advective flux computed? At least 3 moorings will be needed for computing advective flux. Is penetrative heating included in the analysis?

We have added more details about the surface mixed layer heat budget analysis in the beginning of section 2 and the Supplementary Methods A in order to better explain how the individual terms are calculated. Additionally, the equation is given in the discussion.

In addition, we discuss the residual in more detail in section 3 according to the concerns of Reviewer #2 and hope that all details are clarified now.

The mixed layer depth is calculated in ePIRATA using the criterion of a 0.12 kg m^{-3} increase in density from a depth of 1 m, which is now also included in the Supplementary Methods (A).

The penetrative flux is included in the analysis. We did not to clarify this before, but have adjusted the text accordingly as pointed out above.

Horizontal advection is calculated with horizontal velocity averaged vertically in the mixed layer, where the velocity measurements at a depth of 10 m from the mooring are adjusted based on a lookup table for mixed layer velocity as a function of 10 m velocity, mixed layer depth, and time of year. Horizontal gradients of mixed layer temperature are estimated using centered 1° differences from gridded satellite microwave SST. Advection terms are estimated as the product of ocean density, heat capacity, daily mixed layer depth, mixed layer velocity, and horizontal SST gradients. This is now included in the Supplementary Methods A.

Indeed, if we would calculate the entrainment heat flux term as $w_{\text{entrain}}*(T-T_h)$, with $w_{\text{entrain}}=dh/dt+\text{div}(hv)$ as in *Stevenson and Niiler* [1983], we would need 3 moorings. As discussed now in the manuscript for the instantaneous heat balance we do not explicitly calculate this term.

Similarly, there are missing details of NIW analysis. What is time scale of satellite wind? Is it sufficient to resolve inertial rotation? The slab model does not include mesoscale effects on modulation NIW response. Also, as the authors point out, shear instability and turbulent mixing depend on stratification below the mixed layer too. The mixed layer is likely modulated by wind event, which is not included in the analysis. These effects should be discussed.

Several issues are pointed out here, which we will try to answer point by point.

The time resolution of the CCMP wind product is 6 hours, which is explicitly stated now in the Methods section and is sufficient to resolve the inertial rotation as also evident in the Supplement Figure 3. As discussed in the paragraph about the slab ocean model, the crucial point of the wind event during 12-16 September 2015 is the rotation close to the inertial period. Stronger winds have been observed before, but have not caused this strong response, as they do not rotate at an adequate frequency (Fig. 5).

We are aware of the fact that the slab model does not include mesoscale effects on the modulation of the NIW response. During the observed event, we do not find mesoscale features to be present as pointed out above.

The analysis of the IA index and its seasonal variation does not claim to directly parametrize NIW induced mixing. We only use it to investigate the ability of the wind to induce such events and the seasonal distribution of the frequency of occurrence of these events. The seasonal cycle of the residual and the monthly prevalence suggest a close relation of these quantities, however, as we do not use a parametrization of the diapycnal heat flux at the ML base we are not able to directly make the connection.

The modulation of the ML in terms of ML deepening solely due to the wind induced mixing within the ML (which we refer to as entrainment heat flux) is now also discussed.

Some specific comments:

1. Line 63: 'Recently ...'. Microstructure measurements of turbulent flux below mixed layer have been taken for decades.

We have rephrased this paragraph.

2. REMSS: temporal and spatial resolution.

We used in the manuscript the following remote sensing satellite data: On the one hand Cross-Calibrated Multi-Platform (CCMP) gridded surface vector wind product (www.remss.com), which provides wind estimates at 6-hour resolution on a 0.25 degree horizontal grid [*Atlas et al.*, 2011; *Wentz et al.*, 2015]. On the other hand,

Chlorophyll data obtained from the Medium Resolution Imaging Spectrometer (MERIS) with a resolution of 4 km and as monthly mean. The data is available at <https://oceancolor.gsfc.nasa.gov>. Both datasets are now described in the **satellite data** section under the paragraph **methods** at the end of the manuscript.

3. Phase/frequency difference between of shipboard ADCP with slab model result: This might be due to the spatial variation of the wind or the translation of storm.

This might be the case. We included this as a possibility for the slight shift in velocities, thank you.

4. Fig 2b: Strong turbulence occurs at 14:00 below mixed layer. Strangely, the turbulence in the mixed layer is weak. This might suggest remotely generated NIW.

We do not believe that this is a remotely generated NIW. The relation to the wind event is quite clear and the slab ocean model is able to reproduce the event using the local wind forcing. Indeed, mixing within the ML at 14:00 is rather weak, while below the ML a peak is evident. We have added the following lines to the manuscript in section 2.2 explaining these observations: *This additionally supports the idea that the intense turbulent mixing patches result from resonant forcing. Elevated shear of the baroclinic NIW currents drive Kelvin-Helmholtz instabilities at the base of the ML and in the upper thermocline that sustains the locally elevated turbulence. On the contrary, the direct wind stress forcing and associated vertical shear within the ML would result in enhanced turbulence within the ML, which is not observed (Fig. 3h).*

5. Fig. 2 g-h: Is there vertical propagation of NIWs?

There seems to be a weak, out-of-phase signal below the ML. However, to assess the vertical propagation would require a longer time series than is available (3 days only).

6. Line 223-230: Please show the scatter plot of comparison between mooring and slab model results in supplementary material.

We made a scatterplot of the inertial currents observed at the PIRATA buoy and the inertial currents inferred with the slab model, which is shown below, confirming significant correlation over the two-year time period. However, we decided to not include these plots in the Supplementary Material.

7. Line 250-252: The rotation of wind is important. Does the satellite wind resolves inertial rotation?

Yes, this can be seen in the Supplement Figure 3.

8. Line 269: What is the temporal and spatial resolution of CCMP?

6-hour temporal resolution on a 0.25 degree horizontal grid. This is now stated in the methods section.

References

Atlas, R., R. N. Hoffman, J. Ardizzone, S. M. Leidner, J. C. Jusem, D. K. Smith, and D. Gombos (2011), A Cross-calibrated, Multiplatform Ocean Surface Wind Velocity Product for Meteorological and Oceanographic Applications, *Bulletin of the American Meteorological Society*, 92(2), 157-174, doi: 10.1175/2010bams2946.1.

Gregg, M. C., H. Peters, J. C. Wesson, N. S. Oakey, and T. J. Shay (1985), Intensive measurements of turbulence and shear in the equatorial undercurrent, *Nature*, 318(6042), 140-144, doi: 10.1038/318140a0.

Hummels, R., M. Dengler, and B. Bourles (2013), Seasonal and regional variability of upper ocean diapycnal heat flux in the Atlantic cold tongue, *Progress in Oceanography*, 111, 52-74, doi: 10.1016/j.pocean.2012.11.001.

Jochum, M., B. P. Briegleb, G. Danabasoglu, W. G. Large, N. J. Norton, S. R. Jayne, M. H. Alford, and F. O. Bryan (2013), The Impact of Oceanic Near-Inertial Waves on Climate, *Journal of Climate*, 26(9), 2833-2844, doi: 10.1175/jcli-d-12-00181.1.

Kraus, E. B., and J. S. Turner (1967), A one-dimensional model of the seasonal thermocline II. The general theory and its consequences, *Tellus*, 19(1), 98-106, doi: 10.1111/j.2153-3490.1967.tb01462.x.

Large, W. G., J. C. McWilliams, and S. C. Doney (1994), Oceanic vertical mixing: A review and a model with a nonlocal boundary layer parameterization, *Reviews of Geophysics*, 32(4), 363-403, doi: 10.1029/94RG01872.

Legeckis, R., and G. Reverdin (1987), Long waves in the equatorial Atlantic Ocean during 1983, *Journal of Geophysical Research: Oceans*, 92(C3), 2835-2842, doi: 10.1029/JC092iC03p02835.

Niiler, P. P., and E. B. Kraus (1977), One dimensional models of the upper ocean, in *Modelling and Prediction of the Upper Layers of the Ocean*, edited by E. B. Kraus, pp. 143-172, Pergamon Press, New York.

Plueddemann, A. J., and J. T. Farrar (2006), Observations and models of the energy flux from the wind to mixed-layer inertial currents, *Deep Sea Research Part II: Topical Studies in Oceanography*, 53(1), 5-30, doi: <https://doi.org/10.1016/j.dsr2.2005.10.017>.

Schütte, F., P. Brandt, and J. Karstensen (2016), Occurrence and characteristics of mesoscale eddies in the tropical northeastern Atlantic Ocean, *Ocean Sci.*, 12(3), 663-685, doi: 10.5194/os-12-663-2016.

Stevenson, J. W., and P. P. Niiler (1983), Upper Ocean Heat Budget During the Hawaii-to-Tahiti Shuttle Experiment, *Journal of Physical Oceanography*, 13(10), 1894-1907, doi: 10.1175/1520-0485(1983)013<1894:Uohbdt>2.0.Co;2.

von Schuckmann, K., P. Brandt, and C. Eden (2008), Generation of tropical instability waves in the Atlantic Ocean, *Journal of Geophysical Research: Oceans*, 113(C8), doi: doi:10.1029/2007JC004712.

Wentz, F., J. Scott, R. Hoffman, M. Leidner, R. Atlas, and J. Ardizzone (2015), Remote Sensing Systems Cross-Calibrated Multi-Platform (CCMP) 6-hourly ocean vector wind analysis product on 0.25 deg grid, Version 2.0, Remote Sensing Systems, Santa Rosa, CA Google Scholar.

Zaron, E. D., and J. N. Moum (2009), A New Look at Richardson Number Mixing Schemes for Equatorial Ocean Modeling, *Journal of Physical Oceanography*, 39(10), 2652-2664, doi: 10.1175/2009jpo4133.1.

Reviewers' comments second round:

Reviewer #1 (Remarks to the Author):

As indicated in the previous review of this manuscript, this paper presents observational evidence of a near-inertial wave (NIW) induced strong diapycnal mixing event in the tropical Atlantic. The authors also provide arguments by which events such as this vigorous mixing episode have implications for the ML heat balance and the vertical transport of nutrients by calculating an index for inertial activity (IA) in the tropical Atlantic. The subject of the paper is relevant, and the conclusions claimed by the authors are backed by observational evidence.

The new version of the manuscript is much improved, and the revised manuscript reads well and clear. The authors have satisfactorily addressed almost all my previous concerns and comments. Although few more clarifications/changes are needed, I might recommend acceptance with minor revision.

The minor problems that still remain are:

1. Use of MLD data (or more exactly the absence of its use) when calculating the IA index.

Still in line 258 in the new version it is indicated 'To separate the roles of these two factors, their respective seasonal cycles are investigated separately' (these two factors are wind forcing and MLD). I still find this confusing because, as well explained now in the the Supplementary Methods (B) section), IA index estimates solely depends on the wind stress (Line 71: 'To this end, we rewrite the slab model equations as a linear filter that solely depends on the wind stress, and use a threshold criterion to classify the presence or absence of an inertial event for any given point in time and space.'

Later in line 282 it is said 'The seasonal cycle of the MLD ranges from about 30 m at the beginning of the year to about 15 m during boreal summer (Fig 6d). Hence, strong ML responses are most likely to occur between July and September.' Thus, since the inclusion of the MLD data (seasonal cycle) in the calculation of IA index will apparently only enhance and never dampen the IA seasonal cycle (MLD is at its shallowest value when IA index is maximum), I do not find necessary that the authors include MLD variability in IA index calculations.

Still, I think that changes can be made in the main text (e.g L. 258 and L. 282) to clarify these points (e.g. IA index depends solely on the wind stress and MLD variability is not included; and this fact does not change the results because MLD seasonal cycle would only amplify the estimated IA seasonal cycle).

2. Coincidence of areas of elevated Chl concentration and elevated monthly prevalence of NIEs.

I agree with the authors that 'as suggested by the chlorophyll a distributions during JAS (Fig. 10), a band like structure with enhanced concentrations between 6°N and 12°N stretches from the coastal region into the open ocean to 35°W-40°W.' (L. 382). However, I only partially see that 'this structure spatially coincides with regions of both elevated monthly prevalence of NIEs and shallow MLDs (Fig. 8, 10).' I see the highest prevalence of IA north of 5°N during JAS in Figure 8 in a diagonal band from 5°N, 10°W to 10°N, 40°W. This band coincides in Figure 10 with an area of lower chlorophyll concentration (bluish colors).

Here MLD variability can play a more important factor, not only in the IA index estimation but also in the near-surface signal of burst of phytoplankton biomass. I might be wrong but I expect the chlorophyll vertical distribution in this area to show a Deep Chlorophyll Maximum (DCM) associated with Typical Stable Water Structure (TSWS) (Cullen, 2015). If that is the case, the DCM will be located below the MLD. Thus, burst in phytoplankton biomass associated with NIW-induced diapycnal mixing will be not equally observable from satellite if they occur at 25, 30 or 40 m depth (the depth range integrated by satellite measurements is 1/5 of the euphotic zone calculated as the depth at which 1% of the PAR at the surface penetrates. To my knowledge not more than 20 m in many places).

3. Specific comments:

L. 31. It could be interesting to add the equivalent cooling in °K/day as done in L. 183.

L. 155. MLD, which deepens during the observational program from 15 m to about 20 m (Fig. 3e-h). Is this value correct? According to Fig. 3 it seems lower and closer to the estimation based on the entrainment rate (1.5 – 2 m).

L. 165. 'Additionally, the microstructure temperature data revealed frequent overturns with typical scales of about 0.3-0.5 m in the strongly stratified layer below the ML.' Is it shown in the figures? Could be appropriate to add '(Results not shown)'? Maybe I missed something.

L. 370. ETNA instead of (ETNA)

Reviewer #2 (Remarks to the Author):

This paper presents potential evidence of greatly elevated turbulent heat fluxes out of the mixed layer in the North Tropical Atlantic, and argues that these greatly elevated heat fluxes are a consequence of very strong vertical shear (and reduced 8-m Ri) that is a consequence of resonant wind forcing of mixed layer near inertial motions. The authors provide motivation for the presence of such strong, but rare, mixing events by noting that the ML heat budget in the region has a very large residual term, that is combined error and vertical turbulent heat fluxes at the MLD (entrainment heat fluxes). They compute an inertial activity index based on the Pollard-Millard model that shows a seasonality to these infrequent events, and connect this to regional phytoplankton blooms (through enhanced mixing).

Although I still believe there is the potential for a meaningful contribution here, presently there are a number of inconsistencies in the analysis and significant further refinement is needed.

1. In presenting such large directly-estimated turbulent heat fluxes using MSS, the authors need to present their approach very carefully instead of referring the reader to another ref. The details of the microstructure are *very* important here, as most readers (at least with a background in these measurements) will be skeptical of those values. In particular, the authors need to show high KT in regions of high dT/dz . Why weren't detailed microstructure profiles shown? This is where this work can significantly contribute to ML model work by showing *how* these fluxes were generated. It appears to be shear instability, but *where* this instability occurred in the water column is important. The vertical resolution in fig 4 is very coarse and loses much of this depth-dependence. There is some indication that much of this elevated mixing took place within the thermocline (eps plot in fig 3)--potentially explaining why there was a limited ML temp. response. One suggestion is to include individual microstructure profiles and associated Ri , KT and heat flux profiles.

2. There is sufficient analysis here showing that inertial motions were strong and led to the high shear (and potentially elevated epsilon). As near-inertial motions are fairly ubiquitous and well-understood, I think maybe there is *too* much presented on making this argument and it can be condensed and presented in a more succinct manner. Fig. 5 is good and sufficient for showing the presence of strong wind-driven inertial motions.

3. Errors in residual heat budget term: the authors suggest this is all due to missing mixing, but never address potential summed errors/biases in the other terms. There are lots of refs. out there on this (Weller? Cronin?). I'm guessing they are documented in the ePIRATA database, too. It would be good to put error bounds on this residual and say how much could be due to just compounded errors of the other terms.

4. If mixing during strong inertial events is significant, why do we not see huge SST drops in PIRATA buoy data associated with times of elevated IA? This an obvious comparison. This didn't show up so much in the SST of the PIRATA data for the Sept. 2015 event. Why not? The authors need to do a much more careful analysis closing this gap---comparing *heat content* changes in the ML and thermocline with the directly-estimated vertical turbulent fluxes and the vertical divergence of these fluxes. There is some attempt to do this (e.g. fig 7), but it is incomplete. Showing temp change is qualitative--looking at heat content is quantitative.

5. It would be useful (though not necessary) to show how presently-used 1-D mixed layer models (KPP, PWP, etc.) fail to reproduce the observed mixing. Using a 50 yr old+ model as a basis for motivation is not appropriate. The authors said they ran KPP, why is this not shown? If they can show why models with some sort of Ri scheme included (most of them now), then that could contribute to model improvement. Additionally, there appears to be significant confusion over the relationship between entrainment (bulk representation of turbulent mixing at MLD) and the directly-estimated turbulent heat fluxes. In strong mixing events, they should be close, not vastly different. See Niiler Kraus 77, Stevenson and Niiler 1983, Large et al. 94, Wijesekera et al 2005.

6. The filter used to describe IA has an inverse-dependence on MLD, so one would expect higher IA when the MLD is shallow. It is important to either remove this dependence or more clearly state this dependence. When the MLD is shallower, both increased near-inertial shear and stronger direct wind mixing (from u^*^3/kz) will result. These are well-known and rather obvious consequences, so suggest that authors do not dwell on this but instead focus on important details of the mixing event.

Reviewer #3 (Remarks to the Author):

The authors have greatly improved the manuscript. I have only a few minor comments. I suspect that the strong mixing below mixed layer around Sept 13 14:00 (Fig. 3h) is due to breaking of NIW generated one NIW period ahead in Sept 13 0000-0600, which is trapped in thermocline and break. Previous studies have shown NIW trapped in thermocline roughly after \sim one NIW period (e.g. Sanford et al. 2011, JPO v41). Fig. 5h also shows the stronger WPI on Sept 13 0000. Therefore, I recommend the authors rewrite the paragraph in lines 168-176.

Minor suggestions:

1. Add observed HKE in Fig. 5h. I suspect that HKE is the largest around 00-06 on Sept 13.
2. Line 234: the phase difference might be due to inaccurate wind translation speed.
3. Line 262: Please add $q = u + iv$.

Paper NCOMMS-18-31254A „ Surface cooling cause by rare but intense near-inertial wave induced mixing in the tropical Atlantic” by Hummels, Dengler, Rath, Foltz, Schütte, Fischer and Brandt

Response to the Reviewers

We thank all reviewers for their useful comments and suggestions. With the revised manuscript we hope to have addressed remaining issues. The main changes are the addition of an advanced SST variability analysis in order to link periods of strong ML cooling events to the duration and seasonality of elevated near-inertial wave events. Other changes focus on the clarifying issues related to the IA index and a more precise definition of the terminology used to describe “entrainment” and “diapycnal mixing”.

Reviewer #1 (Remarks to the Author):

As indicated in the previous review of this manuscript, this paper presents observational evidence of a near-inertial wave (NIW) induced strong diapycnal mixing event in the tropical Atlantic. The authors also provide arguments by which events such as this vigorous mixing episode have implications for the ML heat balance and the vertical transport of nutrients by calculating an index for inertial activity (IA) in the tropical Atlantic. The subject of the paper is relevant, and the conclusions claimed by the authors are backed by observational evidence.

The new version of the manuscript is much improved, and the revised manuscript reads well and clear. The authors have satisfactorily addressed almost all my previous concerns and comments. Although few more clarifications/changes are needed, I might recommend acceptance with minor revision.

Thank you very much, we are glad that we managed to address most concerns in the previous resubmitted version.

The minor problems that still remain are:

1. Use of MLD data (or more exactly the absence of its use) when calculating the IA index.

Still in line 258 in the new version it is indicated ‘To separate the roles of these two factors, their respective seasonal cycles are investigated separately’ (these two factors are wind forcing and MLD). I still find this confusing because, as well explained now in the the Supplementary Methods (B) section), IA index estimates solely depends on the wind stress (Line 71: ‘To this end, we rewrite the slab model equations as a linear filter that solely depends on the wind stress, and use a threshold criterion to classify the presence or absence of an inertial event for any given point in time and space.’

Later in line 282 it is said ‘The seasonal cycle of the MLD ranges from about 30 m at the beginning of the year to about 15 m during boreal summer (Fig 6d). Hence, strong ML responses are most likely to occur between July and

September.’ Thus, since the inclusion of the MLD data (seasonal cycle) in the calculation of IA index will apparently only enhance and never dampen the IA seasonal cycle (MLD is at its shallowest value when IA index is maximum), I do not find necessary that the authors include MLD variability in IA index calculations.

Still, I think that changes can be made in the main text (e.g L. 258 and L. 282) to clarify these points (e.g. IA index depends solely on the wind stress and MLD variability is not included; and this fact does not change the results because MLD seasonal cycle would only amplify the estimated IA seasonal cycle).

We apologize for the remaining issues on the explanation of the IA index in the previous version and are grateful to the reviewer for pointing these out. With the IA index we are quantifying the ability of the wind to force near-inertial variability in the near-surface ocean - without considering MLD variability. To unambiguously clarify this, we now state up-front in the text when introducing the filter (IA index): “The ability of the wind stress to force near-inertial motion in the ML is estimated with a linear filter for the wind stress that is based on the slab model equations but does not take into account MLD variability (see Supplement Methods (B.2))” (lines 266-268). We fully agree with your suggestion that including MLD in the IA index is not necessary. Instead, when discussing the impact of an elevated IA index, we point out that periods of elevated IA coincide with periods of shallow mixed-layers and that both, the wind forcing and MLD, favor strong near-inertial currents during a particular season.

2. Coincidence of areas of elevated Chl concentration and elevated monthly prevalence of NIEs.

I agree with the authors that ‘as suggested by the chlorophyll a distributions during JAS (Fig. 10), a band like structure with enhanced concentrations between 6°N and 12°N stretches from the coastal region into the open ocean to 35°W-40°W.’ (L. 382). However, I only partially see that ‘this structure spatially coincides with regions of both elevated monthly prevalence of NIEs and shallow MLDs (Fig. 8, 10).’ I see the highest prevalence of IA north of 5°N during JAS in Figure 8 in a diagonal band from 5°N, 10W°W to 10°N, 40W°. This band coincides in Figure 10 with an area of lower chlorophyll concentration (bluish colors).

Here MLD variability can play a more important factor, not only in the IA index estimation but also in the near-surface signal of burst of phytoplankton biomass. I might be wrong but I expect the chlorophyll vertical distribution in this area to show a Deep Chlorophyll Maximum (DCM) associated with Typical Stable Water Structure (TSWS) (Cullen, 2015). If that is the case, the DCM will be located below the MLD. Thus, burst in phytoplankton biomass associated with NIW-induced diapycnal mixing will be not equally observable from satellite if they occur at 25, 30 or 40 m depth (the depth range integrated by satellite measurements is 1/5 of the euphotic zone calculated as the depth at which 1% of the PAR at the surface penetrates. To my knowledge not more than 20 m in many places).

We agree that the band-like structure of IA and the elevated chlorophyll-a distributions do not exactly coincide in Fig. 10. We also expect that an increase in the nutrient availability will be dependent on the combination of elevated IA and shallow ML. Note that for Fig. 10 we have averaged over the JAS season of the respective years due to the limited coverage of the chlorophyll-a data, and the MLD distribution

is a climatological mean as MIMOC does not give any interannual variability of MLD. As additionally mentioned by the reviewer, the ability of the satellite to “see” the chlorophyll-a burst is also limited to the upper layers and is dependent on the availability of nutrients in the deeper layers, which will also be influenced by other factors, e.g. advective processes. Hence, there are a lot of uncertainties in relating these processes.

However, the point we wanted to make here is that it is very likely that other quantities besides heat are influenced by strong NIE events and that it is important to investigate this further in order to understand the seasonal distribution of chlorophyll-a and related phytoplankton blooms.

We have added this to the discussion section in order to make clear that we suggest this relation, but have no direct proof of it and should not directly expect a 1:1 overlap between elevated IA and chlorophyll-a.

3. Specific comments:

L. 31. It could be interesting to add the equivalent cooling in °K/day as done in L. 183.

Thank you for this suggestion, but unfortunately we are already at the word limit for the abstract.

L. 155. MLD, which deepens during the observational program from 15 m to about 20 m (Fig. 3e-h). Is this value correct? According to Fig. 3 it seems lower and closer to the estimation based on the entrainment rate (1.5 – 2 m).

Indeed, the previously reported MLD change during the 20-hour profiling station were based on the first and last few profiles, but those differences cannot be understood as gradual MLD increase. To avoid confusion, we replaced the words in question and now state that “MLD varies between 15 and 20m”.

L. 165. ‘Additionally, the microstructure temperature data revealed frequent overturns with typical scales of about 0.3-0.5 m in the strongly stratified layer below the ML.’ Is it shown in the figures? Could be appropriate to add ‘(Results not shown)’? Maybe I missed something.

Thank you, we added “results not shown”.

L. 370. ETNA instead of (ETNA)

Changed.

Reviewer #2 (Remarks to the Author):

This paper presents potential evidence of greatly elevated turbulent heat fluxes out of the mixed layer in the North Tropical Atlantic, and argues that these greatly elevated heat fluxes are a consequence of very strong vertical shear (and reduced 8-m Ri) that is a consequence of resonant wind forcing of mixed layer near inertial motions. The authors provide motivation for the presence of such strong, but rare, mixing events by noting that the ML heat budget in the region has a very large residual term, that is combined error and vertical turbulent heat fluxes at the MLD (entrainment heat fluxes). They compute an inertial activity index based on the Pollard-Millard model that shows a seasonality to these infrequent events, and connect this to regional phytoplankton blooms (through enhanced mixing).

Although I still believe there is the potential for a meaningful contribution here, presently there are a number of inconsistencies in the analysis and significant further refinement is needed.

We would like to thank Reviewer #2 for his/her critical review of our manuscript as well as the corrections and suggestions to improve the manuscript. We believe to have significantly improved the revised version of the manuscript upon his/her remarks and suggestions. Please find our response to the main comments below, while our response to annotations which were added to a pdf-version of the main document are added to a separate document uploaded as an extra file (point_by_point_to_reviewer2) are uploaded as an extra file.

1. In presenting such large directly-estimated turbulent heat fluxes using MSS, the authors need to present their approach very carefully instead of referring the reader to another ref. The details of the microstructure are *very* important here, as most readers (at least with a background in these measurements) will be skeptical of those values. In particular, the authors need to show high KT in regions of high dT/dz . Why weren't detailed microstructure profiles shown? This is where this work can significantly contribute to ML model work by showing *how* these fluxes were generated. It appears to be shear instability, but *where* this instability occurred in the water column is important. The vertical resolution in fig 4 is very coarse and loses much of this depth-dependence. There is some indication that much of this elevated mixing took place within the thermocline (eps plot in fig 3)--potentially explaining why there was a limited ML temp. response. One suggestion is to include individual microstructure profiles and associated Ri , KT and heat flux profiles.

Indeed, we agree that the manuscript benefits from a better description of our methods and from presenting details of the microstructure data. In the revised version, we have rephrased parts of the methods section concerning the shipboard observations and now provide details about the microstructure sampling and post-processing.

Additionally, we are now showing all individual microstructure and hydrographic profiles in Fig. 4, while highlighting two individual profiles showing particularly elevated signals. We now also include a panel showing profiles of dT/dz , which we had not included in Fig. 4 previously. Marked profiles in the contour presentation of the quantities in Fig. 3 are highlighted in Fig. 4.

2. There is sufficient analysis here showing that inertial motions were strong and led to the high shear (and potentially elevated epsilon). As near-inertial motions are fairly ubiquitous and well-understood, I think maybe there is *too* much presented on making this argument and it can be condensed and presented in a more succinct manner. Fig. 5 is good and sufficient for showing the presence of strong wind-driven inertial motions.

We generally tried to further condense the paper and streamline the argumentation. However, we want to keep Fig. 6 as it shows the seasonal cycle of monthly prevalence of strong NIEs, which we also need for our new analysis concerning the

seasonal cycle of strong ML cooling inferred from the ePIRATA heat budget residual time series (see point 4), which is included in Fig. 7 now.

3. Errors in residual heat budget term: the authors suggest this is all due to missing mixing, but never address potential summed errors/biases in the other terms. There are lots of refs. out there on this (Weller? Cronin?). I'm guessing they are documented in the ePIRATA database, too. It would be good to put error bounds on this residual and say how much could be due to just compounded errors of the other terms.

Thank you for this suggestion. We calculated the monthly climatological error for the individual terms of ePIRATA using the daily error estimates of the individual terms given from ePIRATA and assuming a decorrelation time scale of 3 days: Accordingly the monthly climatological error ($x_{b/monthly}$) for a heat budget term b with daily error estimates x_b is given by $x_{b/monthly} = \frac{1}{n} \sqrt{3 \sum x_b^2}$, where n is the number of available daily error estimates for the term b used for the climatological monthly mean. The climatological monthly error of the residual term is then calculated as the sqrt of the sum of the squares of the monthly clim. heat storage rate, advection, and surface heat fluxes. The resulting errors are included in Fig. 2 and the explanation added to the Methods section.

4. If mixing during strong inertial events is significant, why do we not see huge SST drops in PIRATA buoy data associated with times of elevated IA? This an obvious comparison. This didn't show up so much in the SST of the PIRATA data for the Sept. 2015 event. Why not? The authors need to do a much more careful analysis closing this gap---comparing *heat content* changes in the ML and thermocline with the directly-estimated vertical turbulent fluxes and the vertical divergence of these fluxes. There is some attempt to do this (e.g. fig 7), but it is incomplete. Showing temp change is qualitative--looking at heat content is quantitative.

The SST drop in PIRATA is not directly equivalent to the diapycnal heat flux of 370 Wm^{-2} as other terms in the ML heat balance provide significant warming. We are not able to provide a local balance for the ship board observations as we detail below. Therefore, we used the ePIRATA data to analyze the local balance over 5 days. This allows us to estimate a reliable SST cooling, as we can filter out the diurnal variations in SST and the dataset provides an estimate of horizontal heat advection. We also added a discussion (line 327-332) that connects the inferred diapycnal heat flux from microstructure with the results from the ePIRATA budget that was lacking in our previous version. We argue as follows: The detailed analysis of the event from the slab ocean (Fig. 5) shows that our microstructure observations were obtained during the peak of the event. Hence, it is rather likely that diapycnal mixing was less intense during the other 4 of the 5 days over which the local balance at PIRATA was calculated. Still, the required cooling derived from the observed drop in SST over the 5 days amounts to 130 Wm^{-2} on average. Hence, this does show that diapycnal mixing must be strongly elevated during this 5-day period and supports the finding from direct microstructure measurements at the location of the shipboard measurements.

Finally, we added an elaborate analysis of the heat budget residual from the ePIRATA data set further supporting a tight relationship of ML cooling events and NIEs: Using the ePIRATA heat budget residual (R) and converting those to temperature changes ($dT/dt = R / (\rho c_p h)$) we first see that the histogram is slightly skewed towards negative events (skewness = -0.48, Fig. 7c). Furthermore, we can now look at the seasonal cycle of strong ML cooling events ($dT/dt < -0.2^\circ\text{C day}^{-1}$, Fig. 7d). The seasonal cycle of these strong cooling events looks very similar to the seasonal cycle of monthly prevalence of IA as shown in Fig. 6 supporting the idea that NIEs cause strong ML cooling during July-September.

**Comment inserted in the previous version of the manuscript at line 333:
It is good to look at this, but why can't you look at the 10-hour time series of the MSS profiles, use better vertical resolution, and actually calculate the vertical divergence to get warming/cooling rates? ..and then compare these to observed rates?**

We thank the reviewer for this comment. Certainly, the 20-hour time series of hydrographic profiles warrants a calculation of heat content changes in the mixed layer and the diapycnal heat flux convergence layer 10-30m below the mixed layer. Unfortunately, as explained below, the results do not lead to strong conclusions. The MSS data was collected between 10am (UTC) Sep. 13 and 6am Sep. 14, 2015. ML temperatures were 28.1°C when we started sampling at 10am and 28.2°C when we finished at 6am the next day (see Fig. R1).

Fig. R1: Time series of mixed-layer temperature (upper panel), heat content (middle panel) and required heat flux (lower panel) calculated from the hydrographic data collected during MSS profiling

Instead of an anticipated cooling, the observed temperature change over the whole sampling period is equivalent to an average mixed layer heat gain of about 100 Wm^{-2} during the 20-hour period. However, as shown in Fig. R1, a warming of the mixed layer from 9pm to midnight and again from 2am to 4am was observed. This warming cannot be explained by air-sea heat fluxes, because it occurred during night time. Instead, we hypothesize that the warming originates from lateral heat advection. When looking at the SST distribution in Fig. 1, a strong SST gradient is evident in the

vicinity of the measurement location showing warmer waters in the east of the position. Westward flow due to the NIW prevailed from about noon Sep. 13 to noon Sep. 14 (see Fig. 3) likely advected warmer waters (at speeds of up to 0.5 ms^{-1}) across our measurement position which strongly contributed to the local heat budget. As we lack detailed information of mixed-layer velocities and horizontal temperature gradients in the ML, errors in estimating advective heat flux would obscure any meaningful estimate of ML heat loss.

The same arguments hold for the diapycnal heat flux convergence layer 10-30m below the ML. Here, elevated variability of isopycnal displacement, likely do to elevated internal wave variability, additionally hinders an accurate evaluation of the mixed layer heat balance.

For our study, we thus decided to use ePIRATA data that allows evaluating the budget over a longer time period and includes estimates of the advective heat flux.

5. It would be useful (though not necessary) to show how presently-used 1-D mixed layer models (KPP, PWP, etc.) fail to reproduce the observed mixing. Using a 50 yr old+ model as a basis for motivation is not appropriate. The authors said they ran KPP, why is this not shown? If they can show why models with some sort of Ri scheme included (most of them now), then that could contribute to model improvement. Additionally, there appears to be significant confusion over the relationship between entrainment (bulk representation of turbulent mixing at MLD) and the directly-estimated turbulent heat fluxes. In strong mixing events, they should be close, not vastly different. See Niiler Kraus 77, Stevenson and Niiler 1983, Large et al. 94, Wijesekera et al 2005.

We agree that there are many different definitions of entrainment, which can cause some confusion. We used the 50-yr old model mainly because it was suggested by a reviewer in the previous round, as it is based on the pointed out u^3/kz scaling and it shows that ML deepening based only on wind forcing (without accounting for shear driven mixing) is not of similar magnitude as the diapycnal heat flux obtained from microstructure observations. We refer to entrainment as the ML deepening according to the wind forcing and not as the bulk representation of the turbulent heat flux. This is in agreement with former ML heat balance studies as e.g. Foltz et al. 2003, 2013, 2018, where the entrainment term is considered to be of minor importance. There seems to be a different terminology used in the definition of entrainment and turbulent mixing though in general. In the definition of the PWP model or Niiler and Kraus 77 we agree that entrainment as a bulk representation of turbulent mixing and the directly estimated turbulent heat fluxes should be close. In the definition for entrainment we use here based on the ML heat balance studies, though, this is not the case.

Concerning the KPP results, we ran the model, but it did not perform satisfactorily as we point out in the manuscript. Similar unsatisfactory results have been shown in previous studies trying to directly simulate observed mixing rates during energetic events (Zaron and Moum, 2009). In our experience, KPP has clear deficiencies when used with direct observations, being designed for large scale ocean models providing integral mixing properties. This is also shown in another study in preparation by Foltz et al. 2019.

However, as we feel that our data set consisting of 25 microstructure profiles during a 20-hour time period is too limited to properly evaluate a 1-D ML model like KPP, we did not elaborate on this in more detail. We hope for further studies that will be based on a more comprehensive data set including a much larger number of turbulence profiles as well as a better resolution of vertical shear of horizontal velocities to evaluate this in more detail.

We included some of these arguments in the discussion part of the manuscript.

6. The filter used to describe IA has an inverse-dependence on MLD, so one would expect higher IA when the MLD is shallow. It is important to either remove this dependence or more clearly state this dependence. When the MLD is shallower, both increased near-inertial shear and stronger direct wind mixing (from u^3/kz) will result. These are well-known and rather obvious consequences, so suggest that authors do not dwell on this but instead focus on important details of the mixing event.

The filter does not include a variable MLD ($c_2 = 2 dt/(\rho H)$ is set constant in the filter, see Section B.2 in the supplementary information). We emphasize this more strongly in the main text and in the supplementary materials now. Sorry for the confusion.

Reviewer #3 (Remarks to the Author):

The authors have greatly improved the manuscript. I have only a few minor comments.

I suspect that the strong mixing below mixed layer around Sept 13 14:00 (Fig. 3h) is due to breaking of NIW generated one NIW period ahead in Sept 13 0000-0600, which is trapped in thermocline and break. Previous studies have shown NIW trapped in thermocline roughly after ~ one NIW period (e.g. Sanford et al. 2011, JPO v41). Fig. 5h also shows the stronger WPI on Sept 13 0000.

Therefore, I recommend the authors rewrite the paragraph in lines 168-176.

Thank you for pointing this out. We have added a statement to the given paragraph that the winds rotating at near-inertial frequency were present about one day prior to the observations in Fig. 3, which generally fits the findings in the Sanford et al. (2011) paper that it takes a while until the currents to start to rotate after the wind forcing acts. However, in our case it does not take a complete inertial period, which at this location is 2-2.4 days.

Minor suggestions:

1. Add observed HKE in Fig. 5h. I suspect that HKE is the largest around 00-06 on Sept 13.

There is no Fig. 5h, probably Fig. 5d is meant. We suspect that HKE refers to the horizontal kinetic energy from observations and is accordingly added to the panel 5d. Note for better visibility of the different curves we switched to hourly averages of the observed velocities here instead of the previous 10-minute averages. However, it seems that the peak in HKE is more around 12:00 on Sept 13 and does not really correspond to the response time mentioned in Sanford et al. (2011). However, in

Sanford et al. (2011) the ML response to the passage of a hurricane is investigated, which in any case was a much stronger event than the one we describe here and maybe some of the response in our case is masked by other ML processes.

2. Line 234: the phase difference might be due to inaccurate wind translation speed.

Thank you, we followed your suggestion and added inaccurate translation of the wind field as a possible explanation of the phase difference to line 240.

3. Line 262: Please add $q = u + iv$.

Added.

Reviewers' comments third round:

Reviewer #2 (Remarks to the Author):

This latest version is significantly clearer, more succinct and yet more thorough and careful with the analysis and discussion. Significant improvements include the addition of details of the individual microstructure profiles and associated heat flux profiles and further analysis of the expected associated changes in heat/temperature tendency based on these fluxes.

A few remaining concerns and issues that need to be addressed before this work should be published.

1. First, again, the definition of "diapycnal fluxes" should be clarified. I am happy to write to Gregg or Moum to help to clarify if needed.

Please see Osborn80 equations 8, 9, and 10. All gradients and equations are ref. to the vertical and the TKE equation balance that results in these relationships relies on the balance of (shear) production, dissipation and PE creation via buoyancy fluxes. The PE buoyancy flux is in the *vertical*, of course, as it works against gravity. Yes, I do see where Osborn80 notes cross-isopycnal, but this isn't correct if he's considering 3-dimensions. Also see Gregg87 about diapycnal/isopycnal coordinates where he writes the familiar Osborn80 relationship in terms of the vertical coordinate (equations 34-38), though incorrectly calls them diascalar or diapycnal.

I think page 88 of Osborn80 clarifies it a bit. The reason he (they) says cross-isopycnal and folks use diapycnal/diascalar is that this estimate does not include the influence of along-isopycnal mixing with sloping isopycnals. Thus, I think more correctly, $K_{\rho}N^2$ is the *vertical component* of the cross-isopycnal or diapycnal turbulent buoyancy flux, which, when isopycnal slopes are small is very close to the true diapycnal diffusivity and flux. Given the uncertainty of ϵ s and K_{ρ} of a factor of two (Osborn80, Peters88), these distinctions are likely unimportant to the results, but are more important to saying exactly what you mean and being truthful to the science. Additionally, *vertical* fluxes are more applicable to mixed layer heat balances, etc, as the solar penetrative radiation is modeled in the vertical (among other reasons).

2. The ML temperature evolution equation (near line 306) needs some work. First, a simple correction, all terms need to have the same units (W/m^2). You forgot to multiply some terms by ρ cp. Secondly, your definition of h here is very different than in Stevenson and Niiler 83 (which your eqn. is based upon) who integrated in depth to an *isotherm*. This is a very important distinction as the entrainment velocity (w_e) is often *defined* as the turbulent "pseudo-flow" across an isopycnal (or isotherm in this case) (also see St. Laurent 2000, Alford05). The problem arises from the fact that your definition of h is based on a vertical density gradient of sorts (0.12 dens diff. from 1 m value), and *not* on an isotherm or isopycnal. So the evolution of your " h " in time is dependent upon the vertical divergence of the vertical buoyancy flux. Thus, there are vertical turbulent fluxes influencing ML-averaged temperature BOTH in your entrainment term (w_e) AND in your $Q(-h)$ term. I have been working on deriving the Stevenson Niiler 83 formulation to get to the root of the problem and suggest you do the same. Also see Emery76 and Cronin McPhadden 97 as refs---but note that the latter could have similar issues to SN83. I recognize that this is a rather complex problem, but it is certainly worth looking at it more closely and noting these oversights in the text (unless equation is exact).

3. When you calculated vertical turbulent fluxes in/out of the ML, you for some reason averaged over depth bins well below the ML (5 to 15 m). Why did you not just average the fluxes *AT* h or at least in a depth-window centered on h ? This may explain why expected turbulent fluxes based on the residual is much less than the estimated fluxes from observations. Also note that your flux estimates have a built-in factor of 2 uncertainty (Osborn80, Peters88). No getting around that.

4. You need to clearly state why you use IIR filter for IA vs. just the PM model. It still isn't clear in the manuscript.

Paper NCOMMS-18-31254B „ Surface cooling cause by rare but intense near-inertial wave induced mixing in the tropical Atlantic” by Hummels, Dengler, Rath, Foltz, Schütte, Fischer and Brandt

Response to the Reviewers

Our answers to the reviewers are marked in green.

Reviewer #1 (Remarks to the Author):

As indicated in the previous review of this manuscript, this paper presents observational evidence of a near-inertial wave (NIW) induced strong diapycnal mixing event in the tropical Atlantic. This is the third version of the manuscript I previously reviewed. Although I found the previous (second) version of the manuscript much improved with respect to the first version, frankly speaking – using the expression used by the authors- I only partly see a similar improvement of the last submitted version. I might say that I am overall happy with the responses to all reviewers.

However, my main concern is that I do not see that what is mentioned in the responses is completely done in the manuscript. Some of these issues touch key messages of the manuscript and need to be resolved before publication. Still I think that the subject of the paper is relevant, and I might recommend acceptance with revision.

Thank you very much for this comment. We are very sorry that some of the points mentioned in the responses did not appear treated adequately in the manuscript. We hope that we resolved this issue in the revised version. We are very grateful for the additional points raised in this review and have included your comments and suggestions in the manuscript.

Examples of the problems that still remain are:

1. Evidence of NIW mixing events impact on nutrient availability and biological activity ?????

In my previous review, I commented to the authors:

‘I agree with the authors that ‘as suggested by the chlorophyll a distributions during JAS (Fig. 10), a band like structure with enhanced concentrations between 6°N and 12°N stretches from the coastal region into the open ocean to 35°W-40°W.’ (L. 382). However, I only partially see that ‘this structure spatially coincides with regions of both elevated monthly prevalence of NIEs and shallow MLDs (Fig. 8, 10).’ I see the highest prevalence of IA north of 5°N during JAS in Figure 8 in a diagonal band from 5°N, 10W°W to 10°N, 40W°. This band coincides in Figure 10 with an area of lower chlorophyll concentration (bluish colors). Here MLD variability can play a more important factor, not only in the IA index estimation but also in the near-surface signal of burst of phytoplankton biomass. I might be wrong but I expect the chlorophyll vertical distribution in this area to show a Deep Chlorophyll Maximum (DCM) associated with Typical Stable Water Structure (TSWS) (Cullen, 2015). If that is the case, the DCM will be located below the MLD. Thus, burst in phytoplankton biomass associated with NIW-induced diapycnal mixing will be not equally

observable from satellite if they occur at 25, 30 or 40 m depth (the depth range integrated by satellite measurements is 1/5 of the euphotic zone calculated as the depth at which 1% of the PAR at the surface penetrates. To my knowledge not more than 20 m in many places).’

In their response, they mentioned: ‘We agree that the band-like structure of IA and the elevated chlorophyll-a distributions do not exactly coincide in Fig. 10. We also expect that an increase in the nutrient availability will be dependent on the combination of elevated IA and shallow ML. Note that for Fig. 10 we have averaged over the JAS season of the respective years due to the limited coverage of the chlorophyll-a data, and the MLD distribution is a climatological mean as MIMOC does not give any interannual variability of MLD. As additionally mentioned by the reviewer, the ability of the satellite to “see” the chlorophyll-a burst is also limited to the upper layers and is dependent on the availability of nutrients in the deeper layers, which will also be influenced by other factors, e.g. advective processes. Hence, there are a lot of uncertainties in relating these processes.

However, the point we wanted to make here is that it is very likely that other quantities besides heat are influenced by strong NIE events and that it is important to investigate this further in order to understand the seasonal distribution of chlorophyll-a and related phytoplankton blooms. We have added this to the discussion section in order to make clear that we suggest this relation, but have no direct proof of it and should not directly expect a 1:1 overlap between elevated IA and chlorophyll-a.’

Notice the authors that in their response I have highlighted the key message that they tell they want to suggest. However, in the manuscript the hedge used in their response is completely absent. First in the abstract (L36), it is said: ‘We illustrate the impact of these rare but intense NIW induced mixing events on the ML heat balance and nutrient availability, clearly showing their contribution to the seasonal evolution of sea surface temperature and biological productivity in the tropical North Atlantic.’ I find this sentence a clear assertion that is not backed by observational evidence, both for nutrient availability (that is not shown at all), for sea surface temperature, which seasonal cycle is neither shown in the manuscript (see below comment to L 304), or biological productivity.

Later in the discussion in relation with results presented in Fig. 8, 9 and 10, it is said (L412-416): ‘The results presented here provide evidence that in addition to the large-scale cyclonic wind field, wind variability with frequency close to f introduced by AEWs is most likely an additional cause for the observed phytoplankton blooms and needs to be considered when analyzing the seasonal cycle of chlorophyll and associated biological productivity.’

As explained in my previous review (see comments above) from the results and 10, I would not conclude the same as the authors. However, that is not a problem. I understand that the reviewer and the author may retain somewhat different points of view on several aspects of the observations and physical phenomena under discussion. The reviewer does not believe that such differences should prevent publication of the paper. The problem here is that the authors seem to agree in their response that **they do not have direct proof of their conclusions** and say that what they just want to suggest that is ‘it is **very likely** that other quantities besides heat are influenced by strong NIE

events'. However, in the manuscript they reiteratedly indicate that **they provide evidence** that NIE events **are most likely** an additional cause for the observed phytoplankton blooms.'

The reviewer thinks that the authors must be as cautious as they are in their responses to the reviewers as in the manuscript. In the view of that, changes similar to those suggested below are required:

Thank you very much for coming back to this point. We apologize for not having made the changes consequently throughout the manuscript. We agree with the suggestions of the reviewer and changed the formulations in the manuscript in the lines given in the following:

L418-422: 'The results presented here ~~provide evidence~~ suggest that in addition to the large-scale cyclonic wind field, wind variability with frequency close to f introduced by AEWs is ~~most~~-likely an additional cause for the observed phytoplankton blooms and needs to be considered when analyzing the seasonal cycle of chlorophyll and associated biological productivity.'

We fully agree with the suggested changes and rephrased the text accordingly.

L36: 'We illustrate the impact of these rare but intense NIW induced mixing events on the ML heat balance and nutrient availability, ~~clearly showing~~ discussing their *potential* contribution to the seasonal evolution of sea surface temperature and biological productivity in the tropical North Atlantic.' or any other suitable alternative found by the authors.

Thank you. In agreement with your comment, we modified the sentence mentioned above and now state, "*We illustrate the impact of these rare but intense NIW induced mixing events on the ML heat balance, highlight their contribution to the seasonal evolution of sea surface temperature, and discuss their potential impact on biological productivity in the tropical North Atlantic.*"

2. Clarification of the use of MLD data.

As for the previous comment, this comment originates from previous ones very similar in my previous reviews. As mentioned in the second review, Still in line 258 (now in line 264) it is indicated 'To separate the roles of these two factors, their respective seasonal cycles are investigated separately' (these two factors are wind forcing and MLD). I still find this confusing because as well explained now in the following line in the manuscript 'The ability of the wind stress to force near-inertial motion in the ML is estimated with a linear filter for the wind stress that is based on the slab model equations but does not take into account MLD variability'.

To avoid unnecessary misunderstanding in the explanation, if the authors want to maintain this sentence I might suggest to slightly modifying it (e.g. 'To separate the roles of these two factors, their respective contributions are investigated independently. Regarding the ability of the wind stress').

Thank you. We have changed the sentence accordingly and now write: *To separate the roles of these two factors i.e. that wind input and the MLD, their respective contributions are investigated independently. The ability of the wind stress....*

3. Use of the terms turbulent heat flux and diapycnal mixing.

As mentioned by reviewer 2, I'm also happy to correspond with reviewer 2 to work this out. To this respect, I would like to indicate that in my first review I pointed the need to find an alternative term to 'turbulent heat flux' used in the manuscript to refer to 'diapycnal mixing' at the base of the mixed layer and typically used in the bibliography to refer to the sum of air-sea latent and sensible heat fluxes. However, I did not specifically indicated that this term should be 'diapycnal mixing'. I simply suggested to find an alternative term and gave examples (e.g. turbulent heat flux at the ML base; diapycnally-induced turbulent heat flux,). Thus, I will be happy with any other term that reviewer 2 and the authors find convenient.

We fully agree with Reviewer #1 that the phrase “turbulent heat flux” was confusing considering the air-sea exchange in terms of the latent and sensible heat flux. According to the comments of Reviewer #2 we have now decided to call our term “vertical diffusive ML heat flux” throughout the manuscript and hope that everybody can agree to that. The first time we refer to this term we expand the explanation: *vertical diffusive heat flux due to turbulent mixing across the base of the mixed layer termed vertical diffusive ML heat flux in the following.*

4. Entrainment definition.

L 122. Why in your definition the entrainment flux refers to ML deepening caused ONLY by the wind stress? What about ML deepening caused by negative heat or buoyancy fluxes as indicated by reviewer 2? I also understand that the entrainment flux refers to the flux (heat, nutrient, ...) caused by ML deepening, independently of its origin (wind stress vs. buoyancy fluxes). Is it possible that a negative air-sea heat flux event deepened the ML and increased entrainment in September 2015? The one-dimensional ML model used in the paper evaluates ML deepening due only to wind stirring. If net air-sea heat fluxes are negative (heat loss by the ocean) during the event, it would be necessary to evaluate ML deepening due to buoyancy fluxes (following e.g. Marshall, 1999). If net air-sea heat fluxes are positive, please indicate to discard this possibility (L 122 e.g: In our definition the entrainment flux refers to ML deepening caused only by the wind stress since net air-sea heat fluxes are positive (heat gain by the ocean) during the period of investigation ...)

Thank you for this comment. The average air-sea heat flux during the evaluated 5-day event (September 12 -17, 2015) is 80 Wm^{-2} considering the incoming short-wave radiation absorbed in the ML, the outgoing longwave radiation as well as the latent and sensible heat fluxes. Hence, the net air-sea heat flux is positive during the event as well as for the monthly means from March to October and we therefore discard this possibility and also state so in the manuscript as suggested by the reviewer, which at this place (L122) now reads:
Here, the entrainment flux refers to ML deepening caused only by the wind stress since the net air-sea heat flux is positive (heat gain by the ocean) during most of the year (March – October) and in particular during the period of the NIW event analyzed here.

Additionally, we added the value of $+80 \text{ Wm}^{-2}$ over the 5-day event in the discussion section to indicate that the net air-sea heat flux does not contribute to ML deepening during the evaluated time period.

L 78: *As elevated vertical shear from the mean current system is not pronounced at these latitudes, it was hypothesized that infrequent, but very strong diapycnal mixing events dominate the mean diapycnal heat flux at the ML base: It is necessary to provide a reference here since this is the main hypothesis of the manuscript.*

Thank you for the comment. We are referencing Brandt et al. 2015 for the lack of elevated vertical shear from the mean current system is not pronounced at these latitudes. Additionally, we cite Bourlés et al. 2019 for the second part of the sentence that it was hypothesized that rare, but strong mixing events dominate the mean vertical diffusive heat flux at the ML base.

L 156, *The depth layer of elevated Sh^2 follows the mixed layer depth (MLD), ...'* Does it mean 'it is located below the MLD'? '**follows its variability**'? **Fig. 3e does not show that the layer of elevated Sh^2 slightly deepens. Please clarify.**

Thank you. We have rephrased this paragraph, pointing out the fact that shear close to the MLD is elevated, it now reads:

The vertical distribution of horizontal velocity results in strongly elevated squared vertical shear, $Sh^2 = \left(\frac{du}{dz}\right)^2 + \left(\frac{dv}{dz}\right)^2$, reaching values on the order of 10^{-3} s^{-2} between 21 m (upper limit of our measurements) and 40 m depth (Fig. 3d). Despite the general absence of strong mean currents at this location e.g. ³¹, the observed levels of Sh^2 during this NIW event are comparable to those of the energetic Equatorial Undercurrent e.g. ¹⁸. The mixed layer depth (MLD) was situated between 15 m and 20 m depth during the observational period of 20 hours (Fig. 3e-h).

L 161, *The ratio of stratification, N^2 , and Sh^2 , i.e. the Richardson number (Ri), that is a measure for the tendency of shear instability, frequently shows small values less than 1 directly below the ML (Fig. 3g). The current color scale in Fig. 3g only shows two measurements directly below the ML between approximately 25 and 30 m depth and 16 and 17 hours (brown grid cells in Fig. 3g) and another one between 20 and 25 m at approximately 20:00 with Ri values less than 1. That does not seem frequent. On the other hand, magenta grid cells (indicating also $Ri < 1$) are more common, but they are located within the ML. Please modify the color scale in Fig. 3g to show more clearly what the sentence indicates or modify the sentence accordingly.*

Thank you for this comment. We have decided to rephrase the sentence pointing out the fact that we find small Ri in regions of high Sh^2 and relatively weak stratification just below the ML. It now reads: *The ratio of stratification, N^2 , and Sh^2 , i.e. the Richardson number (Ri), defined as $Ri = \frac{N^2}{Sh^2}$, that is a measure of the tendency for shear instability, shows small values less than 1 also just below the ML (Fig. 3h).* Please note that the Ri distribution (Fig. 3h) has slightly changed as Ri was recalculated upon a comment by Reviewer 2.

L185 and Figure 4, From the vertical profiles, diapycnal heat and salt fluxes into the ML were determined by averaging the fluxes in the depth layer between 5 and 15 m below the ML. How 10-m average values of turbulent dissipation rates, eddy diffusivities, and diapycnal heat fluxes between 15 and 25 meters (red bars highlighted by green arrows below) can have the values shown in Fig. 4 a, c and, d respectively if most of the individual profiles show values ranging outside (highlighted by green brackets) the mean and standard deviation indicated by these bars? In view of the profiles, one would expect the mean values closer to the grey bars, and so to previous estimations. Is there something here that I missed? That is a very important information since the high diapycnal fluxes are calculated from these estimates.

Please note that Fig. 4 a, c, d use a logarithmic scale on the x-axis. The averaging is done arithmetically, which is appropriate averaging formalism for turbulent quantities (e.g. Davis, R. E., 1996, Sampling turbulent dissipation, Journal of Physical Oceanography, 24, 341-358). Hence, the average quantities on a logarithmic scale are dominated by the largest values. This is typical for average turbulent quantities being dominated by rare, but very strong events.

L 232. If the vertical resolution is 5 m at PIRATA, how can be the MLD 17 m deep? Maybe wrongly, I assumed the 5 m resolution at e.g. 0, 5, 10, 15, 20 In that case, the MLD can be estimated at 15 or 20 m. Perhaps 17 m is an average value? If so, please indicate.

It is correct that the vertical resolution of T and S profiles at PIRATA is 5 m, but linear interpolation is used between depths and therefore MLD can take any value. The MLD calculation is also detailed in Foltz et al. 2018.

L 294. 'Velocity observations from September 2015 revealed the presence of a strong NIW associated with elevated vertical shear of horizontal velocities (Sh^2) in the vicinity of the PIRATA buoy at 11.5°N, 23°W.' Is Sh^2 significantly lower below the ML during the rest of the cruise? One can only say that the NIW event was associated with elevated vertical shear of horizontal velocities (Sh^2) if before and after the event Sh^2 was significantly lower. Otherwise, the intensity of Sh^2 cannot be associated with the occurrence of the NIW event. MSS observations are available for a shorter period but vmADCP data should provide a longer record.

Thank you for pointing this out. Concerning the September 2015 R/V Meteor cruise M119, we show in Fig. R1 zonal and meridional velocity as well as Sh^2 from the vessel-mounted ADCP data covering the period from September 11 - 21. While the data is plotted against time, we have added a second x-axis in the lower panel to also show the latitudinal position of the ship according to the time of the cruise (note that due to the cruise track the latitude axis is not linear). It is obvious that during the time of the event and near the PIRATA buoy at 11.5°N, Sh^2 is elevated below the ML. It is still elevated until about September 18.

To interpret Fig R1, one has to consider not only time, but also location. During the cruise, measurements near the PIRATA buoy were carried out until about September 15. R/V Meteor then moved to the south. South of 10.5°N, an eastward current was present during the cruise (northern branch of the North Equatorial Countercurrent, nNECC). In this region, elevated Sh^2 is observed below the nNECC, which was

rather strong during this cruise. Further south, elevated Sh^2 clearly descends from the ML into deeper layers together with the NECC and is not related to a NIW event. For the rest of the section until September 21, Sh^2 below the ML is clearly reduced.

Fig. R1: Upper panel shows cruise track of R/V Meteor cruise M119. Lower panel shows horizontal velocities as measured by the vessel-mounted ADCP between 16°N and 1°N.

As another example, a similar plot is shown for R/V Meteor cruise M130 (Fig. R2), which conducted a measurement program in the same region from August 28 to October 3, 2016. On this cruise, no NIW event was evident during the cruise near the PIRATA buoy. Note that also the NECC was not as pronounced at this time of the year resulting in rather low Sh^2 during the entire cruise track along 23°W.

Fig. R2: Upper panel shows cruise track of R/V Meteor cruise M130. Lower panel shows horizontal velocities as measured by the vessel-mounted ADCP between 16°N and 1°N.

In comparison with the other cruises to the location of the PIRATA buoy analyzed within this study, Sh^2 was one to two orders of magnitude larger in September 2015 than Sh^2 determined from vessel-mounted ADCP data collected during all other cruises to the region. We have added the latter sentence to the beginning of the discussion section.

L 304, During this time period SST and ML temperature drop by 0.28 K (Fig. 7b): Which is the climatological SST change between these days? To provide further confidence to the estimates presented in this paragraph, I think the manuscript will be benefit from showing here (as an insert in Fig. 7b), the climatological seasonal cycle SST at the study area. From satellite-derived SST

and using Fourier decomposition to obtain the climatological seasonal cycle, it seems to be as shown below:

It seems to be in agreement with climatological seasonal cycle of heat tendency shown in Fig. 2b (as should be) and reinforces the argument that NIW may be the potential main cooling term explaining the absence of a slow warming in SST during JAS. From this climatological seasonal cycle the climatological SST change between 12 and 15th September is $+0.0509^{\circ}\text{C}$ (highlighting the relative importance of the NIW event). Finally, showing the climatological seasonal cycle somewhere in the manuscript will provide arguments to state that the authors 'illustrate the impact of these rare but intense NIW induced mixing events on the ML heat balance and nutrient availability, clearly showing their contribution to the seasonal evolution of sea surface temperature' as indicated now in the abstract.

Thank you very much for this comment, this is great and really reinforces the argument that NIW induced mixing is potentially the main cooling term explaining the absence of a slow warming in SST as expected from the climatology as you say. We have used the PIRATA SST from 2006-2017 to construct a climatological seasonal cycle, which looks much like the one shown by the reviewer from satellite observations and show it inserted in Fig. 7b as suggested. In addition, we included the climatological seasonal cycle for the same time period as the SST in September 2015 and the ML temperature in Fig. 7b (from September 10 - 23). For the 5-day period from September 12 - 16 defined as the NIW event, we obtain a warming of $+0.15^{\circ}\text{C}$, which emphasizes the importance of the NIW event for cooling SST during this period.

Fig. R3: Same as Fig. 7 in the manuscript.

Hence, now we include at this place in the discussion the following lines:

Note that the climatological seasonal cycle during the 5-day period shows a warming of 0.15 K, in contrast to cooling of 0.28 K during the NIW event in 2015 (Fig. 7b). This additionally suggests that the vertical diffusive ML heat flux during the NIW event is able to reverse the climatological warming, leading instead to an intermittent ML cooling

We also reconciled the sentence of the abstract pointed out in this comment with the first comment concerning the chlorophyll distribution, which now reads as suggested: *We illustrate the impact of these rare but intense NIW induced mixing events on the ML heat balance, showing their contribution to the seasonal evolution of sea surface temperature, and discuss their potential impact on biological productivity in the tropical North Atlantic.*

L 320. ‘Over the period of 5 days, the model predicts a ML deepening of 1.2 -1.5 m.’ Is this deepening the difference between the MLD at the beginning and the end of the simulation period, as done in previous versions for the observations? For the entrainment, it is necessary to accumulate the effect of all the potential mixing events (e.g.: if during the 5 days the MLD was at 17, 22, 18, 21 and 19, the entrainment flux would be the resulting from passing the MLD from 17to 22 m; from 22 to 18 m; from 18 to 21 m; ... and not directly from 17 to 19).

Thank you for pointing this out. In fact, the ML deepens continuously during the simulation period. Its deepening thus represents the accumulative effect. We have rephrased the sentence to clarify this, which now reads:
Over the period of 5 days, the model predicts a **continuous ML deepening totaling 1.2 -1.5 m.**

L 347, *On the contrary, the seasonal cycle of strong warming events is clearly reduced and does show only very few strong warming events in August and September (red line in Fig. 7d).* Since heat exchange anomalies during the summer in the ML would be distributed in a shallower layer, they would necessarily result in higher warmer or cooler anomalies as shown in Fig. 7 d with the peak of strong anomalies during the summer. Thus, I suggest to remove the second part of the sentence (as indicated above) since the peak of warm anomalies is also observed in summer. However, the peak is clearly smaller and this fact together with the distribution of daily temperature anomalies in Fig. 7c provide very interesting for the manuscript.

Thank you for this comment; we have removed the second part of the sentence from the manuscript.

Reviewer #2 (Remarks to the Author):

This latest version is significantly clearer, more succinct and yet more thorough and careful with the analysis and discussion. Significant improvements include the addition of details of the individual microstructure profiles and associated heat flux profiles and further analysis of the expected associated changes in heat/temperature tendency based on these fluxes.

A few remaining concerns and issues that need to be addressed before this work should be published.

Thank you for this evaluation. We hope that with the revised version we could tackle the remaining concerns and issues.

1. First, again, the definition of "diapycnal fluxes" should be clarified. I am happy to write to Gregg or Moum to help to clarify if needed.

Please see Osborn80 equations 8, 9, and 10. All gradients and equations are ref. to the vertical and the TKE equation balance that results in these relationships relies on the balance of (shear) production, dissipation and PE creation via buoyancy fluxes. The PE buoyancy flux is in the *vertical*, of course, as it works against gravity. Yes, I do see where Osborn80 notes cross-isopycnal, but this isn't correct if he's considering 3-dimensions. Also see Gregg87 about diapycnal/isopycnal coordinates where he writes the familiar Osborn80 relationship in terms of the vertical coordinate (equations 34-38), though incorrectly calls them diascalar or diapycnal.

I think page 88 of Osborn80 clarifies it a bit. The reason he (they) says cross-isopycnal and folks use diapycnal/diascalar is that this estimate does not include the influence of along-isopycnal mixing with sloping isopycnals. Thus, I think more correctly, $K_{\rho N}^2$ is the *vertical component* of the cross-isopycnal or diapycnal turbulent buoyancy flux, which, when isopycnal slopes are small is very close to the true diapycnal diffusivity and flux. Given the uncertainty of ϵ and K_{ρ} of a factor of two (Osborn80, Peters88), these distinctions are likely unimportant to the results, but are more important to saying exactly what you mean and being truthful to the science. Additionally, *vertical* fluxes are more applicable to mixed layer heat balances, etc, as the solar penetrative radiation is modeled in the vertical (among other reasons).

We thank the reviewer for the discussion on the terminology appropriate for the heat fluxes due to turbulent mixing across the base of the mixed layer. We now think that an adequate term should be "*vertical diffusive ML heat flux due to turbulent mixing across the base of the mixed layer*" (line 65-67) which we then abbreviate as "*vertical diffusive ML heat flux*" in the following.

Indeed, vertical fluxes are more applicable to the mixed layer balance presented in the manuscript. We hope that you can agree to using the above terminology.

2. The ML temperature evolution equation (near line 306) needs some work. First, a simple correction, all terms need to have the same units (W/m^2). You forgot to multiply some terms by ρc_p .

Thank you very much. This was corrected.

Secondly, your definition of h here is very different than in Stevenson and Niiler 83 (which your eqn. is based upon) who integrated in depth to an *isotherm*. This is a very important distinction as the entrainment velocity (w_e) is often *defined* as the turbulent "pseudo-flow" across an isopycnal (or isotherm in this case) (also see St. Laurent 2000, Alford05). The problem arises from the fact that your definition of h is based on a vertical density gradient of sorts (0.12 dens diff. from 1 m value), and *not* on an isotherm or isopycnal. So the evolution of your " h " in time is dependent upon the vertical divergence of the vertical buoyancy flux. Thus, there are vertical turbulent fluxes influencing ML-averaged temperature BOTH in your entrainment term (w_e) AND in your $Q(-h)$ term. I have been working on deriving the Stevenson Niiler 83 formulation to get to the root of the problem and suggest you do the same. Also see Emery76 and Cronin McPhadden 97 as refs---but note that the latter could have similar issues to SN83. I recognize that this is a rather complex problem, but it is certainly worth looking at it more closely and noting these oversights in the text (unless equation is exact).

Thank you for bringing our attention to these issues. The derivation of this equation is presented in the appendix of Moisan and Niiler (JPO, 1998). They chose an isotherm for the vertical integration, but the derivation is the same if MLD is chosen instead. This equation does not include any vertical diffusive heat flux, which instead shows up as a residual. The entrainment term is due only to additional water being added to the mixed layer from below, either through thickening of the ML or horizontal divergence of mass within the ML. In reality there is turbulence when the ML thickens, but this shows up in the ML heat balance equation only in the residual. There may be an indirect impact of turbulence on the ΔT term in entrainment, but it would not be straightforward to quantify this effect. Based on these considerations, we conclude that the ML heat balance equation that we use is exact, though separating into entrainment and diffusive heat flux is not easily done.

3. When you calculated vertical turbulent fluxes in/out of the ML, you for some reason averaged over depth bins well below the ML (5 to 15 m). Why did you not just average the fluxes $\Delta T \cdot h$ or at least in a depth-window centered on h ? This may explain why expected turbulent fluxes based on the residual is much less than the estimated fluxes from observations. Also note that your flux estimates have a built-in factor of 2 uncertainty (Osborn80, Peters88). No getting around that.

We understand the reviewers concern about determining vertical diffusive heat flux in a depth intervall somewhat below the mixed layer as a proxy for the heat flux out of the mixed layer, inparticular, when the individual heat flux estimates show decreasing values right below the mixed layer.

There are two main reason why we avoid calculating heat fluxes in the mixed layer itself: ship induced turbulence and uncertainties of the mixing efficiency Γ in regions with low stratification. Ship induced turbulence typically reaches to water depth of 10 m to 15 m below the surface. These depth are close to the base of the mixed layer which has minimum values of 14 m in our study. Secondly, for regions of low

N^2 , the canonical mixing efficiency of 0.2 is not valid, and we would need to make assumptions on how Γ behaves in our regime. In low stratified regimes mixing efficiency has a functional dependence on buoyancy Reynolds number $\varepsilon/\nu N^2$ (ν -kinematic viscosity), gradient Richardson number and Reynolds number (e.g. Gregg et al., Mixing Efficiency in the Ocean, Annual Review of Marine Science, 2018) that can not be accurately determined from our data set. Thus, for our estimate, we have to rely on regions where the canonical value is applicable.

However, after some reconsideration, we found that due to the elevated stratification just below the mixed layer, the canonical value of 0.2 holds in the depth range between 1m and 10m below the mixed layer where buoyancy Reynolds number are $10 < \varepsilon/\nu N^2 < 1000$. We thus decided to use this depth interval in the revised version of the manuscript and hope to accommodate the reviewers concern.

Indeed, vertical diffusive heat fluxes decrease when using the 1-10m depth interval below the ML. Additionally, vertical diffusive heat flux convergence now occurs in the depth layer between 20-40m below the ML (see Fig. R4 below or Fig. 4 in the manuscript).

Fig. R4: same as Fig. 4 of the manuscript.

Concerning the second point, we agree that there are systematic errors in the estimate of epsilon due to the uncertainty in the sensitivities of the shear sensors as mentioned in Peters et al. 1988. However, we decided to retain our statistical error estimate, which can be derived from the observational dataset. We added a comment about possible additional systematic errors, which are not accounted for in this statistical error estimate and now state:

(Fig. 2b, numbers in brackets indicate the upper and lower 95% confidence limits determined from statistical error propagation³⁷. Note that systematical errors, such as uncertainties in the shear sensor sensitivity are not included in this estimate).

When checking our uncertainty estimates, we found a bug in our calculation of the statistical error, which after correction increased the error bars in Fig. 4 (see below). Corresponding adjustments were made throughout the text.

4. You need to clearly state why you use IIR filter for IA vs. just the PM model. It still isn't clear in the manuscript.

Thank you for pointing this out. We have added another sentence before the definition of IA in section 2.5 stating: We reformulate the model equations into an equivalent linear filter to emphasize the fact that it is the wind stress alone that we base our metric on.

Reviewer #2 (Remarks in the marked copy of the manuscript):

Line 31, Abstract, 370 W m^{-2} : This is a rate. Can you please add a timeframe associated with this rate? e.g. 370 W m^2 over X hours, days, etc. OR describe this is a *peak observed* flux.

Thank you. According to one of the previous comments, we have added the 20-h continuous measurements.

Line 33, Abstract, "During the 32 course of a year, elevated IA prevails during only 16 days on average": This is *very* subjective and based on whatever threshold that you use to define "elevated". Can you put this in more objective quantitative terms? E.g. "the 5 most energetic events only span X% of the time" ...or something along those lines.

Thank you, we agree that the 16 days are very subjective and based on our choice of $\alpha = 2$. Hence, we have removed this subjective number.

Line 54: Citations?

We are now citing a paper by Bill Johns et al.: Johns, W. E., P. Brandt, and P. Chang, Tropical Atlantic variability and coupled model climate biases: results from the Tropical Atlantic Climate Experiment (TACE), *Climate Dynamics*, 43, 2887, doi:10.1007/s00382-014-2392-1, 2014.

Line 60: chlorophyll what? distribution? concentration?

Thank you. The term "distribution" was added.

Line 103: Trying to reconcile these events being "rare" but causing strong *mean* vertical mixing at the mixed layer base? Maybe you clarify this later.

We have removed "rare" from the sentence above, as this can not yet be stated from the 2015 observations.

Line 116: you didn't hyphenate in previous "shortwave"

Thank you, "short-wave" was changed to "shortwave".

Line 116: annual average?

Yes, “annual” is now included in the text.

Line 121-123, “In our definition the entrainment flux refers to ML deepening caused by the wind stress and is mostly considered to be of minor importance for the ML heat balance”: Stevenson and Niiler 83 (which your eqn. is based upon) integrated in depth to an *isotherm*. This is a very important distinction as the entrainment velocity (w_e) is often *defined* as the turbulent "pseudo-flow" across an isopycnal (or isotherm in this case) (also see St. Laurent 2000, Alford05). ...so ML temp evolution equation needs some work.

Please see our reply to point 2. In the response letter above on the issues about the ML temperature equation. We also altered the text and now state, “In our definition the entrainment flux refers to ML deepening caused only by the wind stress since the net air-sea heat flux is positive (heat gain by the ocean) during most of the year (March – October) and in particular during the period of the NIW event analyzed here. Mostly, the entrainment flux is considered to be of minor importance for the ML heat balance ^{15, 16, 17} .

Line 132-133, “numbers in brackets indicate the upper and lower 95% confidence limits determined from error propagation”: Both Osborn80 and Peters88 indicate an uncertainty in K_{rho} and ϵ measurements of a factor of two , so your heat fluxes error bounds should reflect this. Also, please state the time-averaging used to get 371. Avg. over 20 hours of sampling?

Please see our detailed response (point 3. above) about our uncertainty estimate above explaining the factor of 2 and thank you for pointing out the suspiciously low uncertainties. We found an error in our confidence interval calculation which is now corrected. We have also added the time period for our estimate as suggested.

Line 150-151, “At deeper depth, a reversal of the NIW related flow indicates a baroclinic nature of the NIW”: It would be helpful to note the rotation direction *in depth* which will tell you about the phase propagation and thus will support other evidence that this was surface-generated. I think Sanford76? has something about this.

Thank you for the comment. We replaced the sentence above and instead state “A downward energy propagation of the NIW is indicated by an anticyclonic rotation of the current vector with depth (Leaman and Sanford 1975)”. The reference is Leaman and Sanford, Vertical energy propagation of internal waves: a vector spectral analysis of velocity profiles, J. Geophys. Res., 80, 1975-1978, 1975.

Line 159-160, “Below the ML, stratification in the thermocline and halocline (Fig. 3e) is at maximum (Fig. 3f, $N^2 \sim 10^{-3} s^{-2}$) separating the ML from the ocean interior.”: at maximum $\sim 10^{-3} s^{-2}$ (Fig 3f) separating the... But I don't really understand this sentence. Do you mean strat. reaches a local maximum in depth?.or peaks at ? Thank you, indeed, the sentence was not very clear. We replaced the sentence by “Below the ML, stratification in the thermocline and halocline (Fig. 3e) reaches values of $N^2 \sim 10^{-3} s^{-2}$ (Fig. 3f)”.

Line 168, “Additionally, the microstructure temperature data revealed frequent overturns with typical scales of 0.3-0.5 m in the strongly stratified layer below the ML (results not shown)”: **Do these overturn sizes correspond with eps. values as per Osmidov scaling ($\sqrt{\epsilon/N^3}$)? Overturn size can be determined by comparing sorted and unsorted profiles of potential density. This would be a fantastic check on your epsilon values.**

Thank you for this comment. The overturn sizes mentioned in the manuscript (0.3-0.5 m) were determined as you mention above (Thorpe scale, Thorpe, (1977)). According to Ozmidov scaling ($L_o = \sqrt{\epsilon N^{-3}}$), overturning scales of 0.3-0.5 m in the thermocline ($N^2 \approx 10^{-3} \text{ s}^{-2}$) correspond to dissipation rates of $3 - 8 \times 10^{-6} \text{ W kg}^{-1}$, which matches the observed dissipation rates in the turbulent patches very well (see figure 4). To express this consistency we added “Dissipation rates of turbulent kinetic energy (ϵ) inferred from these overturning scales according to Ozmidov agree very well to the elevated ϵ inferred from microstructure measurements (Fig. 3h).”

Line 175-176, “Elevated shear of the baroclinic NIW currents drive Kelvin-Helmholtz instabilities at the base of the ML and in the upper thermocline”: **Unless you can see this directly in your observations you should try to find a citation to support such a claim. I agree with you (tho maybe not purely Kelvin Helmholtz) but need a citation.**

Thank you! We changed the term “Kelvin-Helmholtz (KH)” to “shear instability” to account for possible differences to a standard KH model and included a reference to Smyth, W. D., and J. N. Moum (2012), Ocean mixing by Kelvin-Helmholtz instability, *Oceanography*, 25, 140–149, doi:10.5670/oceanog.2012.49.

Line 180, “..., but winds rotating at near-inertial 180 frequency were present about one day prior to the observations presented in Fig. 3.”: **clockwise in time at the near-inertial**

Thank you, we changed the text accordingly.

Line 185-187, “From the vertical profiles, diapycnal heat and salt fluxes into the ML were determined 185 by averaging the fluxes in the depth layer between 5 and 15 m below the ML as in 18. The 186 average ML heat loss estimated from all 25 profiles taken in September 2015 was 371 [482; 187 295] Wm^{-2} .” **Why did you average over such a large vertical window and start the integration 5 m below your definition of the ML. The fluxes into the ML should be the average of the ones along $z=-h$ (maybe a 5 m window centered on $z=-h$). By defining a ML that extends well within the thermocline and averaging over depths well below your defined ML you are obscuring details that could be important ---and it may well explain why we don't see the ML or SST temp response we expect to see from your flux numbers.**

Please see our detailed answer to this remark in our response to comment 3. above.

Line 186-187: “The average ML heat loss estimated from all 25 profiles taken in September 2015 was 371 [482; 295] Wm^{-2} ”. **Suggest noting that it is over 20 hours.**

Thank you, added “during the 20 hours of observation”.

Line 192-193, “In summary, the observations show strongly enhanced diapycnal mixing during the presence of a NIW that was encountered during one out of eight sampling periods at the location”: What percent of total profiles collected at this location? As it is unknown how long each sampling period is, this would be a more quantitative and descriptive metric.

Here, we disagree with your suggestion. We think that the percent of total profiles would be misleading as a large number of profiles taken within short period of time do not represent independent measurements. Our choice of sampling periods reflects better the actual degrees of freedom in our datasets.

Line 266, “The ability of the wind stress to force near-inertial motion in the ML ...”:
Motions?

Yes, thank you changed accordingly.

Line 266 ff: Can you please again state here why this filter is advantageous over just running the PM model with a constant H? How is the filter different than the model?

We have added the sentence “We reformulate the model equations into an equivalent linear filter to emphasize the fact that it is the wind stress alone that we base our metric on.” to clarify the purpose of the filter.

Line 307, heat budget equation: See previous note about concerns about this equation.

Thank you. We have corrected the equation. Please see detailed answer in the comment 2. above.

Line 329-331: This reconciles the 130 Wm^{-2} over the 5-day period with the MSS observations of 370 Wm^{-2} over 20 hours and in addition, it directly relates the elevated vertical diffusive ML heat loss to the observed decrease in SST and ML temperature (Fig. 7b): OR is because you looked at vertical turbulent fluxes too far below the MLD.

As noted above, we adjusted the vertical interval of our vertical diffusive heat flux and salt flux calculation and now report the heat and salt flux between 1m and 10m below the mixed layer.

Line 334-336, “The divergence of the heat flux profile (Fig. 4c) suggest that the heat from the mixed layer is redistributed to between 10 m and 30 m below the ML.” : But this warming is not observed in the MSS data, suggesting that advection may be cooling this layer”? Why use PIRATA data and not use MSS obs as the fluxes directly related to the MSS profiles? Mixing events can be very patchy and intermittent, so the mixing may also be elevated at the mooring site, I don't expect the details (timing, depth range, etc) of the mixing there would be similar to the MSS time series results.

Our revised heat flux calculation now indicated the heat flux divergence in the depth layer between 20m and 40m below the mixed layer. As noted in our previous response letter, elevated variability of isopycnal displacement prevents a reliable estimate of a warming trend in the heat flux convergence layer 20-40m below the ML. We use ePIRATA data because they cover coherently a longer period in time. Nevertheless, the temperature profiles from the MSS do provide a hint of warming in the depth layer between 20 and 40m.

Fig. R5: Mean temperature (upper panel) and heat content (lower panel) of the depth layer 20m to 40m below the mixed-layer. Blue dashed line indicated a trend from a least square fit. Statistical confidence of the fit is only 50%.

Mean temperature and heat content in the layer 20-40m below the mixed layer strongly varies over the 20 hours of MSS profiling (Fig. R5). However, a warming trend of 0.28°C and a heat content increase of 2.2×10^7 J result from a least square fit to the data. Both fits are not statistically significant, but the independently determined trends would require a heat flux of about 300 W m^{-2} .

The warming in the 20m-40m depth layer below the ML can also be illustrated by subtracting a mean vertical temperature profile calculated from the first 12 MSS casts from the mean vertical profile calculated from the last 12 MSS profiles (Fig. R6). From this analysis, we obtain a mean temperature increase of 0.195°C in the depth layer between 20m and 40m now during half the sampling period, which is equivalent to a vertical heat flux divergence of 430 W m^{-2} . However, as suggested by the strong warming in the upper 10m below the thermocline and the cooling between 10m and 20m below the thermocline, horizontal advection of heat likely contributes to the results presented above. We refer to these results in the manuscript by stating: “The divergence of the heat flux profile (Fig. 4c) suggest that the heat from the mixed layer is redistributed to between 20 m and 40 m below the ML. Indeed, despite elevated variability of vertical displacement in the upper thermocline, a warming in this depth layer of $0.2\text{-}0.3^\circ\text{C}$ during the 20h sampling period is indicated by the hydrographic data collected with the MSS (not shown). However, a clear picture is hampered by possible contributions of horizontal advection to the heat (and freshwater) budget below the ML.”

Fig. R6: Vertical profile of the temperature difference below the mixed layer calculated from subtracting the average temperature profile of the first 12 MSS profiles from the average temperature profile of the second 12 MSS profiles.

Line 340-343: I would guess that the frequency of tropical storms follows the IA? ...and thus other cooling effects (large outgoing latent, sensible) fluxes would occur over short periods of time. So this skewness may not be due ONLY to IA but just to the nature of surface heat fluxes in storms. If you plot the same for dT/dt from latent heat or sensible heat fluxes do you get the same skewness? It may be impossible to separate these, but you should mention that these could also play a role in the observed dT/dt distribution.

We investigated the skewness and seasonality of elevated latent heat fluxes (variability of long wave radiation and sensible heat flux are generally too small to significantly contribute to ML temperature variability) using the ePIRATA data set. We find elevated cooling events due to latent heat fluxes (Fig. R7.) having the same order of magnitude as the residual (Fig. 7 in the manuscript).

Fig. R7: Mixed layer heat loss due to latent heat flux in $^{\circ}\text{C}$ per day calculated from the ePIRATA data set. Rare events of elevated latent heat fluxes ($> 0.2^{\circ}\text{C}$) are present in the data set.

However, the seasonality of the elevated latent heat fluxes shows that these events preferentially occur in between September and January with a maximum in December (Fig. R8). Thus, these events do not coincide with elevated residual heat flux events that are likely associated with vertical diffusive heat fluxes due to NIW events that peak in July through September. We decided not to alter the text in the manuscript.

Fig. R8: Seasonal distribution of elevated latent heat flux events leading to a cooling of the mixed layer by more than 0.2°C per day.

Lines 422-424: *The attempt to reproduce the observed diapycnal heat 422 fluxes with a 1-D ML Model KPP, 52 failed and provided diapycnal heat fluxes that were 423 elevated by a factor 3-4 compared to the observations: Probably not best to use absolute and subjective terms when discussing model performance (fail, succeed, verify, validate, disprove, etc)., instead suggest using more objective terms such as "low-skill" . Or leave out the personal interpretation and just say that KPP resulted in vertical heat fluxes that were 3-4 times those observed.*

Thank you. Changed the statement accordingly.

Line 425: *However, the MSS data set consisting of only 25 profiles over a time period of 20 hours is not sufficient to verify or falsify a 1-D ML model like KPP: test model skill*

Thank you, changed.

Line 442-446: *The monthly climatological error for the individual terms of ePIRATA are calculated assuming a decorrelation time scale of 3 days: The monthly climatological error (xb/monthly) for the heat budget term b associated with daily error estimates xb is given by $xb/monthly = 1/n \sqrt{3 \sum xb^2}$, where n is the number of available daily error estimates for the term b used for the climatological monthly mean. Sorry, but I don't understand what you are doing here. Do you have a ref. for this calc? Do the daily value error estimates change in time? Why a de-correlation timescale and how did you arrive at it? How much of the error is potentially bias and the rest uncertainty that can be reduced by averaging? I doubt this will change your results (as the residual is huge), but this needs clarification or at least a reference.*

The uncertainty for the monthly estimates of the ML heat budget terms is calculated for the random errors and a decorrelation time scale of 3 days. We added the wording: *The 3-day decorrelation is based on the zero-crossing of the daily lagged autocorrelation of the mixed layer heat budget terms [Ref. Foltz et al., 2018, J. Clim.].*

Line 460-462: *Buoyancy frequency N for determining Ri -numbers was determined from the hydrographic data by averaging N over the corresponding 8m-depth interval available from the vmADCP velocity observations. This isn't quite correct and will result in problems with Ri --tho likely won't change your big-picture results. Density should be first treated like velocity measurements (half-overlapping triangular bins) and interp'd to the same depth bins. Then density should be *first differenced* over the same vertical scale as that used to get shear to get the appropriate scale of N^2 . Computing N^2 over some small vertical scale (what was it?) and then smoothing in the vertical will actually give different results.*

Thank you for this remark. Indeed, we agree that the calculation was inconsistent with the velocity observations. In the revised version, N^2 was calculated as you mentioned. We use averages of temperature and salinity determined from 16m-overlapping triangular bins (barlett window) and then derive N^2 from the 8m salinity, temperature and pressure grid. The resulting Ri numbers (shown in Fig. 4 h) are only slightly different from the previous numbers, but the calculation is now consistent. We now state: *Buoyancy frequency N for determining Ri -numbers was calculated from the hydrographic data by averaging salinity and temperature using overlapping 16m triangular bins corresponding to the 8m-depth interval available from the vmADCP velocity observations.*

Lines 467-469: *While sinking velocity of the MSS profiler was determined by the change of pressure with respect to time, a dynamic glider flight model⁵⁸ was used to determine the speed of flow past the microstructure sensors. You mean for the microrider? Not the sensors on MSS, correct?*

Yes, thank you, we included the word "microrider".

Line 473: *Furthermore, the vertical diffusive heat flux is calculated via $J_h = -\rho c_p K_\rho \partial T / \partial z$ with ρ and c_p being the density and specific heat capacity of seawater and $\partial T / \partial z$ the instantaneous vertical temperature gradient: Note in Winters, D'Asaro 1996 they note that in order to represent the action of turbulent mixing via a turbulent diffusivity \times a gradient, the gradient should be some time-mean gradient, not necessarily instantaneous as the instantaneous will have overturns associated with it. A sorted profile would work.*

Thank you for your comment. We did sort the temperature profiles prior to calculating temperature gradients. Sorry for being not exact in our description. We changed the text accordingly, and now added to the end of the sentence: *derived from an instantaneous temperature profile, in which temperature values are sorted in descending order.*

Line 877, caption Fig. 1: the legend says "MSS profile" suggesting there is only one profile. Suggest changing to either "station" or "profiles".

Thank you for pointing this out. We changed profile to *station* in the legend.

Line 875, figure 2: If these are average monthly quantities then it seems like they should be plotted mid-month, not at the beginning of the month, no? Unless then Jan average value, for example, is calculated between Dec 15 and Jan 15?

The values are monthly values as you presume and placed at the ticks as we thought it is more convenient to have them at the ticks on the grid lines. We have added an explanation this to avoid confusion to the end of the caption saying: *Note that the ticks on the x-axis correspond to the mid of the month.*

Line 887, caption Fig. 3, a current vectors from the first vmADCP bin (13-21m depth), ...: Is color really needed in (a) if there's already a time-axis? --tho I suppose overlapping vectors are clearer if they are different colors.

Exactly we chose the color code to better distinguish between individual vectors, hence we keep want to keep the colors.

Line 887, caption Fig. 4, suggest reversing the x-axis on this to make it clearer. (larger absolute values to right). Also, if I were investigating Delta T of the ML, I'd look at the vertical turbulent fluxes *just* under the MLD (at the boundary). Not sure why you are averaging well below the MLD. This change doesn't look like it would influence your results very much, however --but it may reconcile differences between observed changes in ML temp and these large downward heat fluxes. One important result of these events is that they may modify thermocline vertical heat distribution (bringing colder water up) which would eventually influence ML temps. This is indicated by the very strong mixing you see in the stratified thermocline. Eyeballing the vertical divergence of J_h shows heat from above "piling up" between 10 and 30 m below the MLD.

Thank you for this comment. We agree to all your remarks here. We have changed our average vertical diffusive heat flux to represent the depth intervall from the mixed-layer base to 10m below the mixed layer. The strong vertical diffusive heat flux convergence now occurs between 20m and 40m depth below the mixed layer as explained in the comments above. Here, the temperature data from the MSS do indicate a warming (Fig. R5 and R6 of this response letter).

Line 964, caption Fig. 7, suggest changing y-axis label on panel (d) to "number of strong Delta T events" as it includes warming events too!

Thank you. Changed the ylabel.

Line 1015, caption Fig. 10, This fig is fine as-is, but for this comparison it might simplify things to include a version of IA that *does* include a seasonally-varying MLD. I'm guessing the correlation with chl. distribution would be strong.

You are probably right. However, we would like to stay with the so far used parameters and want to avoid introducing new definitions of the prevalence close to

the end of the manuscript. It would also require introducing new thresholds, which could complicate the follow-up discussion.

Reviewer #2 (Remarks in the marked copy of the Supplementary):

Line 25: Daily-averaged I hope.

Yes, thank you we added the average.

Line 29: There are likely unknown biases, particularly in the radiative terms.

Added known.

Line 46: citation? In your response to my comments about various schemes for this you incorrectly stated that Ohlmann03 does not include a chl a dependence. It was, in fact, expressly published to show the impacts of a model with a chl a dependence!

Thank you. We have added the appropriate citations for the method used here,

Line 81: Tau vs. T? T was used previously for temperature. Also in your figs you used Tau.

Thank you for this comment. We fully agree that this could have been confusing with the symbol used for temperature. It was meant to be a capital tau, but surely this looks the same than the T for temperature. In our figures we use the real and imaginary part τ_{ax} and τ_{ay} , while here we are referring to the complex forcing term of the wind stress. We therefore decided to use F, for forcing.

Line 143: what about spin-up time? It takes time for wind-stress to communicate the wind KE to the ML. You may want to soften this statement.

Thank you, we softened the statement accordingly.

Reviewers' comments fourth round:

Reviewer #1 (Remarks to the Author):

This paper presents observational evidence of a near-inertial wave (NIW) induced strong diapycnal mixing event in the tropical Atlantic. The authors also provide arguments by which events such as this vigorous mixing episode have implications for the ML heat balance, highlighting their contribution to the seasonal evolution of sea surface temperature, and discussing their potential impact on biological productivity in the tropical North Atlantic. This is the fourth version of the manuscript I previously reviewed, and I happy to say that the authors have addressed in this last version all my previous concerns and comments, which I honestly appreciate. The new version of the manuscript is significantly improved, and the revised manuscript reads well and clear. Since besides the subject of the paper is relevant, and the conclusions claimed by the authors are backed by observational evidence, I recommend it for publication without further changes.

Raquel Somavilla

Reviewer #2 (Remarks to the Author):

There have been significant improvements to the manuscript in this last revision, specifically in terms handling/processing of the MSS turbulence measurements, discussion of water column temperature changes due to this mixing and regarding other technical details such as the calculation of R_i . I continue to think this is important research that should be published. However, I still have some concerns that were brought up during the previous review but were not satisfactorily addressed.

1. I still have issue with the attempt to somehow separate the effects of "entrainment" and "diffusive fluxes" on time changes of ML temperature. Please see my attempt to shed light on some of these things in the attached write-up, where I provide a suggestion of potentially how to do this via MSS profiles. As per the Moisan and Niller 98 equation, entrainment is the movement of an isotherm w.r.t. a background temp gradient (due to turbulent mixing). In addition to this influence on T_a there is the vertical turbulent flux through that isotherm (or interface). It is not clear how you separate these two terms in the N-K model. If the N-K somehow only includes the influence of only one of these terms, then this should be clarified. Also, because you use a density gradient to define h (and not an isothermal or isopycnal surface), there will likely be a complex relationship between "entrainment" and vertical turbulent fluxes at $z=-h$.

2. You still have not fully addressed why you use the linear filter to investigate the tendency of the wind to cause inertial motions. You mentioned this was to remove the influence of the mixed layer depth, but why not just keep this constant in the PM model? If there is no real difference, then simply state so. If there is, the readers need to know what it is.

Paper NCOMMS-18-31254C „Surface cooling caused by rare but intense near-inertial wave induced mixing in the tropical Atlantic” by Hummels, Dengler, Rath, Foltz, Schütte, Fischer and Brandt

Response to Reviewer #2

Remarks to the authors:

There have been significant improvements to the manuscript in this last revision, specifically in terms handling/processing of the MSS turbulence measurements, discussion of water column temperature changes due to this mixing and regarding other technical details such as the calculation of R_i . I continue to think this is important research that should be published. However, I still have some concerns that were brought up during the previous review but were not satisfactorily addressed.

We would like to thank the reviewer for his/her critical review of our manuscript as well as the corrections and suggestions to improve the manuscript. We believe to have significantly improved the revised version of the manuscript based on his/her remarks and suggestions, and through this hope to have satisfactorily addressed the remaining concerns.

1. I still have issue with the attempt to somehow separate the effects of "entrainment" and "diffusive fluxes" on time changes of ML temperature. Please see my attempt to shed light on some of these things in the attached write-up, where I provide a suggestion of potentially how to do this via MSS profiles. As per the Moisan and Niiler 98 equation, entrainment is the movement of an isotherm w.r.t. a background temp gradient (due to turbulent mixing). In addition to this influence on T_a there is the vertical turbulent flux through that isotherm (or interface). It is not clear how you separate these two terms in the N-K model. If the N-K somehow only includes the influence of only one of these terms, then this should be clarified. Also, because you use a density gradient to define h (and not an isothermal or isopycnal surface), there will likely be a complex relationship between "entrainment" and vertical turbulent fluxes at $z=-h$.

We thank the reviewer for pointing this out. Indeed, our estimate of entrainment in the former version of the manuscript was inconclusive and most likely invalid. We thank the reviewer for his/her thoughts on the entrainment flux in the mixed layer heat balance equations by expanding the Moisan and Niiler (1998) derivation. Indeed, it nicely shows that a vertical diffusive ML heat flux is required for entrainment to act in the case when the depth of the ML is defined by an isotherm. Furthermore, it highlights the importance of the exact form of the vertical profile of the vertical diffusive heat flux in the upper ocean for the exchange of heat at the base of the mixed layer. We agree with the reviewer that the definition of the ML depth (i.e. an isotherm or a density gradient criteria) alters the entrainment term in the Moisan and Niiler (1998) approach.

However, we think that any attempt to quantify the heat flux contribution by the entrainment term in theory, and if possible using our data goes far beyond the scope of the manuscript. As detailed in our responses to your annotations below, we refrained from any attempts to separate entrainment and vertical diffusive ML heat flux in the revised version of the manuscript. Also, we reinterpret the N-K model results as merely delivering a heat flux across the base of the mixed layer.

2. You still have not fully addressed why you use the linear filter to investigate the tendency of the wind to cause inertial motions. You mentioned this was to remove the influence of the mixed layer depth, but why not just keep this constant in the PM model? If there is no real difference, then simply state so. If there is, the readers need to know what it is.

Thank you for pointing out that our explanation of the linear filter is still not fully understandable. As detailed below in the responses to your comments in the manuscript, we have changed the text in several places to clarify this issue. The linear filter is an identical reformulation of the slab model using a constant product of ML depth and density. The linear filter does not attempt to simulate a realistic ocean, as the term “slab” or “PM model” might suggest. We prefer to use the term “linear filter” to distinguish between our effort to model the real ocean (using the PM model, section 2.3 and 2.4) or to merely filter the wind stress time series for identifying near-inertial events (section 2.5).

Annotations in the manuscript:

Line 96, “near-by”, **One word I think.**

Yes, thank you, corrected.

Line 120, “The horizontal advection terms are smaller than 18 Wm^{-2} during the entire year”: **Do you mean for the annual average? Or for *any* time during the year? Note that the advective fluxes depend on the lengthscale (e.g. dT/dx) so where there are sharp gradients these can be important on short timescales.**

The statement refers to Fig. 2, where we show monthly averages of horizontal advection from the ePIRATA dataset. To express this more precisely, we now write, “The monthly averages of horizontal heat advection are smaller than 18 W m^{-2} throughout the entire year ...”. We fully agree with your remark that advective fluxes depend on length and time scales. In the sentence before the statement in question, we refer the reader to Supplementary Methods (A) for details on how the fluxes in ePIRATA were derived. In Supplementary Methods (A), we state, “Horizontal gradients of ML temperature are estimated using centered 1° differences from gridded satellite microwave SST. Horizontal advection terms are estimated as the product of ocean density, heat capacity, daily MLD, ML velocity, and horizontal ML temperature gradients”.

Line 122, “Note that this residual term includes the vertical diffusive ML heat flux as well as the vertical entrainment flux.” **You still need to clarify the difference between the two with math.**

Indeed, this sentence is misleading and was deleted in the revised version. However, as stated in the response to the reviewer’s first point above, we feel it is not appropriate to include a discussion on the separation of entrainment and vertical turbulent heat flux in this manuscript.

Line 123, “Here, the entrainment flux refers to ML deepening caused only by the wind stress since the net air-sea heat flux is positive (heat gain by the ocean) during most of the year (March – October). **So, you're suggesting that it is *turbulently* deepening? By wind-driven turbulence e.g. $u^3/\kappa z$ for similarity scaling, acting on the density gradient at the base of the mixed layer? Or by wind-driven mean velocities (e.g. near-inertial motions) causing local shear instability and mixing? If so, how is this different than a vertical turbulent density flux? e.g. $\rho'w'$ acting to increase the depth of a uniformly-mixed layer via vertical density (buoyancy) flux convergences at the base of**

the mixed layer. *You can't have deepening without vertical turbulent buoyancy fluxes* unless you have convection in the ML (surface cooling or evap).

Thank you for pointing out our erroneous statement here. It was a crude and false attempt to quantify entrainment. The sentence was deleted from the manuscript. We fully agree with you that the ML can't deepen without vertical turbulent mixing occurring.

Line 138, "Note that systematical errors, such as uncertainties in the shear sensor sensitivity, are not included in this estimate". **Why are you not referencing the factor of two uncertainty listed in Peters88, etc? You say that these are not included, but do not say how big these are? For full transparency the readers need to know what this is.**

The sentence (line 133-134) now reads, "Note that systematical errors, which may be as large as a factor of 2 (Peters et al., 1988) are not included in this estimate".

Line 145, "Concurrently, horizontal velocity profiles from a vessel-mounted acoustic Doppler current profiler (vmADCP) were available from 17 m to about 800 m depth (Fig. 1, 3)." **A bit clearer if you say profiles of horizontal velocity. so "horizontal" doesn't seem to be referring to "profiles".**

Thank you, we now state "Concurrently, vertical profiles of horizontal velocity from ..."

Line 156, "A downward energy propagation ...". **Suggest just "Downward energy...**

Thank you again; we removed the "A" at the beginning of the sentence in line 150 of the revised version.

Line 158, "strongly elevated squared vertical shear". **at 8-m scales? Or 16 meter?**

We added "at 8-m scales" after shear in line 152 of the revised version.

Line 164, "MLD is calculated using a density increase of 0.12 kg m^{-3} relative to the density at 1 m depth.". **Which is based on...? A typical value used is 0.01, so this may reach some distance into the thermocline. Maybe say this is chosen as the minimum density difference to exclude short-lived (day or less) transient oscillations of the ML.**

In fact, we used the density difference criterion of 0.12 kg m^{-3} to be consistent with the ePIRATA dataset. The criterion gives very similar results as the 0.2°C difference criterion, which is often used for mixed layer balances in the tropics. The mean MLD difference between the two criteria is 0.23 m and max, min differences are 1.9 m and -1.7 m respectively. We added the statement suggested by the reviewer to the manuscript and state in lines 159-161: "This criterion is chosen as the minimum density difference required to exclude short-lived (diurnal or shorter) transient oscillations of the ML".

Line 176, "overturning scales according to Ozmidov". **Why don't you put the scaling here?**

Thank you, we added $L_{oz} = \epsilon^{1/2} N^{-3/2}$ to the text in line 171.

Line 176, "to". **with**

Corrected.

Line 200, "during one out of eight sampling periods at the location". **OK, I can concede that maybe a better indication of independent realizations are the different occupations or cruises with MSS profiles, but it would also be good to be more quantitative. You mentioned a heat flux of 8 W/m^2 for previous cruises. What is the* range of values* of previous cruises compared to this cruise? this would be a good metric of the difference that would help make this statement more robust and meaningful.**

Thank you. We agree with your suggestion and have added “have magnitudes between 3 W m^{-2} and 16 W m^{-2} (Fig. 2b)” to line 127, where the 8 W m^{-2} average was mentioned before.

Line 277, “We reformulate the model equations into an equivalent linear filter to emphasize the fact that it is the wind stress alone that we base our metric on.” **You didn't appropriately address my question/comment from last round. How is this filter *different than just running the PM model with a fixed mixed layer depth?* If it is equivalent, say so. If it is different, say how it is different. This is an important part of this paper, you need to better clarify why you took this specific approach.**

As stated in the response to the second point raised by the reviewer above, there is no difference between the slab model (Pollard and Millard, 1970) and the linear filter. The way we use the linear filter here is equivalent to running the PM model with a setting of $\rho h = 1$ (i.e. setting the product of mixed layer density and depth to unity). We altered the text in question to make this more clear and now state, “The ability of the wind stress to force near-inertial motions in the ML is estimated with a slab model using a constant product of ML depth and density, which is set to unity. This slab model can be identically rewritten in the form of a linear filter, which is done to highlight the fact that the output of the slab model is actually filtered wind stress (see Supplementary Methods (B.2)).”

We also changed the text in the Supplementary Methods section B.2 where we now write (lines 74-78), “What can be achieved, however, is an estimate of the prevalence of strong inertial events in the region that is based only on the wind-stress signal that avoids making assumptions about the state of the ocean. This was done by using a constant product of ML density and depth ($\rho h = \text{const.}$) in the slab model equations. We then used a threshold criterion to classify the presence or absence of an inertial event for any given point in time and space.

The above formulation of the slab model can be understood as a linear infinite-impulse-response (IIR) filter for the wind-stress F . Following ⁸, a linear filter can be expressed as ...”

Line 315, heat balance equation. **Please better clarify the difference between entrainment and "diffusive" fluxes here. This is critical for your subsequent discussion.**

Thank you for the comment. We think that any attempt to quantify the heat flux contribution by the entrainment term in theory and in using our data, if at all possible, goes far beyond the scope of the manuscript. Therefore, we refrained from any attempts to separate entrainment and vertical diffusive ML heat flux in the revised version of the manuscript.

Lines 326-338, “The model was evaluated using an initial temperature profile derived from the microstructure measurements and wind speeds from the PIRATA buoy at 11.5°N , 23°W . Over the period of 5 days, the model predicts a continuous ML deepening totaling in 1.2 -1.5 m. From the associated ML heat content change we derived an entrainment heat flux of $18 - 22 \text{ Wm}^{-2}$. Using this estimate for the entrainment flux and considering Q_0 (80 Wm^{-2}) and hence a warming term during the 5 days), the advective heat flux and the heat storage rate (second and first term on the left-hand side) from ePIRATA, the residual cooling over the 5-day time period amounts to 130 Wm^{-2} ”. **I have concerns with the argument you present here as I'm not sure how you separate "entrainment" from vertical turbulent fluxes at the mixed layer base. Please clarify.**

We fully agree with your concern. We deleted this section from the manuscript. However, as former reviewer # 2 asked us to provide an estimate of the impact of wind mixing (quote of the reviewer remark) “What about the role of u^3/kz in driving mixing when the ML is shallow

and wind stress is significant?”, we now state in line 303-310 “Stirring due to local wind stress and buoyancy fluxes cannot explain the elevated vertical diffusive ML heat loss. An evaluation of a one-dimensional ML model (⁴⁴, see Supplementary Methods (C) for details) initialized with microstructure temperature profiles and using wind speeds and surface fluxes from the PIRATA buoy at 11.5°N, 23°W yields a heat flux across the base of the ML of only 18 – 22 W m⁻². This supports the perception that the observed intense mixing at the base and below the ML resulted from local shear instability that was a consequence of resonant wind forcing.”

Line 348-351, “Indeed, despite elevated variability of vertical displacement in the upper thermocline, a warming in this depth layer of 0.2-0.3°C during the 20h sampling period is indicated by the hydrographic data collected with the MSS (not shown). **Smearing from averaging over periods when there are large vertical displacements can often be removed by averaging along along or between isopycnals. In this case, the isopycnals would move vertically some due to mixing so it would complicate things ,but it might be worth attempting this. Also, the way the sentence is worded makes it sound as if the elevated vertical displacment would inhibit warming from turbulence, which in fact, it just makes it harder to see/quantify.**

Yes, we fully agree. During our last revision, we had also calculated the increase of heat content between the ML and an isopycnal deeper in the water column. The net warming below the ML was comparable to the net warming we obtained from subtracting the average temperature profile of the first 12 MSS profiles from the average temperature profile of the second 12 MSS profiles that was shown in the last response letter. We had included the later analysis in the response letter, because showing the temperature increase from the difference of the averaged profiles against depth nicely pointed out the depth region of the warming.

Thank you for pointing out the unclear phrase. We removed the sub-clause “despite elevated variability of vertical displacement in the upper thermocline,” from the manuscript.

Line 357-358, “Furthermore, the seasonal frequency distribution of elevated cooling events ($\partial T/\partial t < -0.2$ °C day⁻¹, Fig. 7d) is in agreement with the seasonal distribution of the monthly 358 prevalence of IA (Fig.6b,c). **Probably be good to note that this also in part due to the fact that the ML is shallowest then as well! If you removed this dependence, would it still show the same distribution?**

Thank you, we fully agree. Indeed, the distribution is less skewed when mixed layer variability is not accounted for. We added your suggestion to the sentence in question and now state, “The seasonal frequency distribution of elevated cooling events ($\partial T/\partial t < -0.2$ °C day⁻¹, Fig. 7d) peaks in July through September, which is in agreement with the seasonal maximum of the monthly prevalence of IA (Fig. 6b, c) and the seasonal minimum of MLD (Fig. 6d).”

Line 358-360, “On the contrary, the seasonal cycle of strong warming events is clearly reduced (red line in Fig. 7d).” **I don't think "on the contrary" is used correctly here. It usually is used following a statement which is *not* the case. Also, suggest ..."seasonal cycle of strong warming roughly follows that of cooling, but the changes in temperature are much smaller." I'm guessing this latter relationship holds in most parts of the world. Storms are short-lived and intense and usually cause cooling. Warming largely happens by integration in time of shortwave radiation.**

Thank you for your suggestion. The sentence of the previous version was replaced by “The seasonal cycle of strong warming roughly follows that of cooling, but the absolute changes in temperature are much smaller”.

Line 358-360, “Note that the currently available ePIRATA heat budget residual time series (2006-2017) exhibits a total of 30 elevated cooling events (< -0.2 °C day⁻¹) in August, 2.6 of these events on average. On the contrary, the seasonal cycle of strong warming events is clearly reduced (red line in Fig. 7d)”. **per year in August? I think a word or two missing here.**

We changed the sentence to “Note that the currently available 12-year ePIRATA heat budget residual time series (2006-2017) exhibits a total of 30 elevated cooling events (< -0.2 °C day⁻¹) in August, corresponding to 2.5 events in an average month.”

Line 371-373, “ 2b). In addition, strong near-inertial ML velocities are expected only when elevated monthly prevalence of NIEs coincides with a shallow MLD. “**I think this needs to be softened. You very well could get strong inertial motions at other times of the year, tho they wouldn't be a common as strong motions when h is shallow and monthly prevalence is high.**

Yes, we agree, thank you. “Only” was replaced by “to be more common” and the sentence now reads “In addition, strong near-inertial ML velocities are expected to be more common when elevated monthly prevalence of NIEs coincides with a shallow MLD.”

Line 403, “As the peak in monthly prevalence of NIEs during JAS in the ETNA coincides with a shallow MLD (Fig. 8c), vertical mixing intensity is most likely enhanced within this area. **And high latent heat fluxes! These are storms, or cause storms, after all...**

We added a sentence after the sentence above stating, “Likewise, elevated latent heat fluxes associated with AEWs might contribute to ML heat loss.”.

Annotations in the response letter:

(our previous text in the response letter is in blue, your remarks are bold and our reply is in green)

Page 12: We now think that an adequate term should be “vertical diffusive ML heat flux due to turbulent mixing across the base of the mixed layer.”

Yes, this is fine. Why not vertical turbulent ML heat flux? Why are you stuck on "diffusive"? Is this based on the equation form used to estimate these fluxes? (diffusivity times gradient).

We used vertical turbulent heat flux in the first submitted version. However, reviewer 3 remarked in the first review (quote) “**However, the term turbulent heat flux is typically used in the bibliography to refer to the sum of air-sea latent and sensible heat fluxes. Thus, I might suggest to find an alternative term**”. We are glad that everyone agrees to the term “vertical diffusive heat flux”. Indeed, as considered by the reviewer, the term “diffusive” additionally provide information on the underlying form of equation used to determine the fluxes.

Page 13: They [i.e. Moisan and Niiler 83] chose an isotherm for the vertical integration, but the derivation is the same if MLD is chosen instead.

Possibly, but note that the "entrainment" term now means something completely different than their entrainment term. I've included a write-up to maybe help clarify the problem I still have with this.

We thank the reviewer for his/her thoughts on the entrainment flux in the mixed layer heat balance equations by expanding the Moisan and Niiler (1998) derivation. Indeed, it nicely

shows that a vertical diffusive ML heat flux is required for entrainment to act in the case when the depth of the ML is defined by an isotherm. Furthermore, it highlights the importance of the exact form of the vertical profile of the vertical diffusive heat flux in the upper ocean for the exchange of heat at the base of the mixed layer. We agree with the reviewer that the definition of the ML depth (i.e. an isotherm or a density gradient criteria) alters the entrainment term in Moisan and Niiler (1998) approach.

However, we think that any attempt to quantify the heat flux contribution by the entrainment term in theory and in using our data, if at all possible, goes far beyond the scope of the manuscript. In the discussion section, we introduce the mixed layer balance solely to show that there is a large residual (now 150 W m^{-2}) during the period when the near-inertial wave is present. In the revised version, we refrain from attempting to quantify a possible contribution of the entrainment flux and state that the large parts of the residual could be explained by the enhanced vertical diffusive ML heat flux we are inferring. Nevertheless, we think that it would be very interesting to look more deeply into the problem you detailed in your write-up. This would, however, need to be published in a different paper.

Page 13: The entrainment term is due only to additional water being added to the mixed layer from below, either through thickening of the ML or horizontal divergence of mass within the ML.

How do you add water to the mixed layer from below except from vertical turbulent fluxes---e.g. $w'T'$? Pure vertical advection with no mixing would deepen or shoal the mixed layer $dh/dt=-w$, but add no water to it other than the needed lateral convergence/divergence within the ML to shoal or deepen h .

Yes, we agree. There is no entrainment without mixing and we cannot distinguish between the two processes from our data. This sentence was deleted from the current version of the manuscript.

Page 13: In reality there is turbulence when the ML thickens, but this shows up in the ML heat balance equation only in the residual.

this makes little sense to me.

We fully agree. We deleted this sentence from the manuscript as well.

Page 13: Based on these considerations, we conclude that the ML heat balance equation that we use is exact, though separating into entrainment and diffusive heat flux is not easily done

Then you agree that you cannot use the Kraus Turner model to separate "entrainment" from "diffusive" fluxes?

Yes, we fully agree and have adjusted the manuscript accordingly.

Page 13: There are two main reason why we avoid calculating heat fluxes in the mixed layer itself: ship induced turbulence and uncertainties of the mixing efficiency Γ in regions with low stratification. Ship induced turbulence typically reaches to water depth of 10 m to 15 m below the surface. These depths are close to the base of the mixed layer which has minimum values of 14 m in our study. Secondly, for regions of low N^2 , the canonical mixing efficiency of 0.2 is not valid, and we would need to make assumptions on how Γ behaves in our regime.

OK, I'm glad that you considered all these things (mixing efficiency, buoyancy Reynolds number and ship contamination).

Thank you.

Page 14: Concerning the second point, we agree that there are systematic errors in the estimate of epsilon due to the uncertainty in the sensitivities of the shear sensors as mentioned in Peters et al. 1988.

You need to mention this uncertainty in the text, not just sweep it under the rug.

As stated above, we included the statement in the text. The sentence now reads “Numbers in brackets indicate the upper and lower 95% confidence limits determined from statistical error propagation³⁸. Note that systematic errors, which may be as large as a factor of 2 (Peters et al., 1988) are not included in this estimate.”

(Reviewer statement last revision): 4. You need to clearly state why you use IIR filter for IA vs. just the PM model. It still isn't clear in the manuscript.

Page 15 (our response last version): Thank you for pointing this out. We have added another sentence before the definition of IA in section 2.5 stating: We reformulate the model equations into an equivalent linear filter to emphasize the fact that it is the wind stress alone that we base our metric on.

This does not address my question. Why did you not just run the PM model with a fixed mixed layer depth? What is the difference between the filter and a fixed-depth PM model?

The reviewer also expressed this concern in the second point to the authors above and in the annotations in the manuscript to Line 277. Please see our reply to the second point to the authors on the first page of the response letter.

References:

Peters H, Gregg MC, Toole JM. On the parameterization of equatorial turbulence, *J. Geophys. Res.* **93**, 1199-1218 (1988).

Pollard RT, Millard RC. Comparison between observed and simulated wind-generated inertial oscillations. *Deep Sea Research and Oceanographic Abstracts* **17**, 813-821 (1970).

REVIEWERS' COMMENTS fifth round:

Reviewer #2 (Remarks to the Author):

I thank the authors for appropriately addressing my previous concerns--in particular the ongoing and understandable confusion between entrainment fluxes and vertical turbulent fluxes at the ML base. I have no significant issues with the paper in its present form. This paper presents a coherent, clear analysis and discussion of the likely influence of NIWs/NIEs on the mixed layer heat budget in the tropical Atlantic and is an important contribution to understanding factors controlling tropical SST.

Paper NCOMMS-18-31254D „Surface cooling caused by rare but intense near-inertial wave induced mixing in the tropical Atlantic” by Hummels, Dengler, Rath, Foltz, Schütte, Fischer and Brandt

Response to Reviewer #2

Remarks to the authors:

I thank the authors for appropriately addressing my previous concerns--in particular the ongoing and understandable confusion between entrainment fluxes and vertical turbulent fluxes at the ML base. I have no significant issues with the paper in its present form. This paper presents a coherent, clear analysis and discussion of the likely influence of NIWs/NIEs on the mixed layer heat budget in the tropical Atlantic and is an important contribution to understanding factors controlling tropical SST.

We would like to thank the reviewer for his/her critical review of our manuscript again and are very happy that he/she does not have any more significant issues in the present form. Below we answer point by point the remaining annotations in the manuscript and the supplementary file.

Annotations in the manuscript:

Line 63, “tropical Atlantic”, **suggest remove "tropical Atlantic" as this is stated again shortly after as part of PIRATA.**

Thank you, we removed it.

Line 168, “ ... 3h) already pointing to a high probability of $Ri \leq 0.25$.”, **what determines this high probability? Maybe... "suggests critical Ri at smaller vertical scales." ?**

Thank you for this suggestion, we changed the formulation to: “... suggesting $Ri \leq Ri_c$ at smaller vertical scales.”

Line 173, “(Ozmidov length scale $L_{oz} = \varepsilon^{1/2} N^{-3/2}$). **usually written a bit more compactly as $\sqrt{\varepsilon/N^3}$.**”

Thank you, we changed the formulation also according to the Nature Communications format.

Line 180, “ If shear at the base of the ML were absent, direct wind stress forcing at the surface would result in turbulence within the ML being enhanced compared to the thermocline values (e.g.44), which is not observed (Fig. 3g).” **This is a bit confusing as direct wind forcing *will* cause shear at the ML base, but as you said, also will elevate turbulence within the ML. You could just end the previous sentence with, "and not due to direct wind forcing, which would elevated dissipation within the ML."**

Changed, thank you that makes it clearer and shorter.

Line 183, “... diapycnal eddy diffusivity (K_ρ) and vertical diffusive heat flux (J_h) ...”, **can you ref. the "methods" section here so the reader knows where to go to get the definitions of these terms?**

Yes, thank you. We inserted the reference to the Methods section.

Line 187, “...1 m and 10 m below the ML¹⁹. **I won't belabor this point because I've already mentioned it, but due to your very conservative definition of the MLD, as long**

as there is no contamination from ship wake your estimates of vertical turbulent heat fluxes AT the MLD and even somewhat above should be valid. In figure 3g it is clear that ϵ at the MLD is greatly enhanced for the latter part of the time series. Including these fluxes would likely only add to your story.

Thank you for the discussion on this point. We agree with the reviewer, that slightly modifying the definition of the MLD will maybe strengthen, however not significantly change the results of our study.

.Line 215, "...verify ...", "**support**" maybe a better word here.

Line 216, "...by comparing...", "**with comparisons**" if you choose to say "support" as previously recommended.

Thanks for your suggestions, we changed both.

Line 233, "The wind event that triggered the NIW covered approximately 5° in latitude and longitude (see Supplementary Figure 3)." **Note that this scale is influenced by the resolution of the CCMP dataset. A finer grid scale would undoubtedly show finer horizontal wind structure.**

Yes, you are right that a finer grid scale would show a finer wind structure. However, as we are interested in the large-scale setting triggering the NIW event, we believe it is correct to make this statement here.

Line 261, "As the observations in combination with the slab model show that the identified wind event is able to force an energetic NIW associated with enhanced vertical diffusive ML heat flux, ..." **I think it would be helpful to more clearly lay out the sequence of the logic here: e.g. 1) large vertical heat fluxes associated with elevated shear, low Ri at ML base, 2) measured vmADCP velocities suggest this elevated shear associated with a downward propagating wave, 3) Slab model results strongly suggest that these velocities were associated with a strong NIW event.**

Thank you. We have rephrased this sentence and now state: "The MSS observations showed elevated vertical diffusive ML heat fluxes at the ML base associated with low Ri numbers, while the concurrent velocity observations revealed that the observed elevated shear is related to the presence of a downward propagating wave. The combination of these observations with the results of the slab model strongly suggests that the velocities are associated with a strong NIW event. The question then arises how often the wind causes similar responses in and below the ML."

Line 277, "Note that q contains a slowly varying Ekman component that is eliminated by applying a 30-day high pass filter prior to calculating I_A . **Note that this can be explicitly expressed in the PM model and can be removed without need for filtering, so you may be able to explicitly remove it here. $Z_e = T / (\omega H)$, where $T = (t_x + it_y) / \rho$, $\omega = r + if$. Ah, but I see you addressed this in the supplement and that the difference in your approach to the "exact" approach is not significant.**

Exactly. We explicitly checked this and the difference is insignificant.

Line 280, "...CCMP wind...", **I know you are following Nature's format, but it is rather backwards not to define these abbreviations when they first appear, but instead at the end of the paper. Can you at least put some indication here where the reader can get more info on this? Or give a ref. the first time you use this abbreviation.**

We added the reference the first time CCMP winds are mentioned.

Line 282, "This approach is validated by comparing the *IA* from CCMP with the *IA* from PIRATA winds (Fig. 6a). ", **From the supplement plot it looks like CCMP may assimilate PIRATA data....so of course the plots will agree when there is PIRATA data.**

Yes, it is true that CCMP assimilates PIRATA wind data. However, this does not mean that the resulting *IA* time series is automatically identical. At first, changes can arise as other data like e.g. scatterometer winds are assimilated as well, which could alter the variability in the wind time series at this location in the CCMP product. And secondly, the wind data gets interpolated etc. when calculating *IA* and hence it is ad hoc not clear that these time series, which are available at different temporal resolution gave the same high frequency variability, which determines *IA* in the end. Hence, we feel that it is really important to check this.

Line 308, "Stirring", **This usually is used to refer to lateral processes that sharpen gradients, that mixing then acts upon. I'm not sure if you are using it in that way here? Also, many mixed layer models can mostly capture inertial shear at the MLD (e.g. PWP), so you may mention that the KT model accounts for only direct wind/buoyancy forcing and does not accumulate momentum in the ML.**

Thank you for pointing out the problems with this term. We have rephrased the two sentences and they now read: "Mixed layer deepening due to local wind stress and buoyancy fluxes cannot explain the elevated vertical diffusive ML heat loss. An evaluation of a one-dimensional ML model, which only accounts for direct wind⁽⁴⁴⁾, see Supplementary Methods (C) for details), initialized with microstructure temperature profiles and using wind speeds from the PIRATA buoy at 11.5°N, 23°W yields a heat flux across the base of the ML of only 18 – 22 W m⁻².

Line 312, "perception", **conclusion?**

Changed to conclusion.

Line 360, "... in August ...", **in the month of (just to help to clarify this more)**

Line 360, "... an average month." **August.**

Thank you for this suggestion, we changed it accordingly.

Line 398, "... 4.8-5.7 / 2.4-2.9 / 1.6-1.9 days at 5°N / 10°N / 15°N respectively." **Why do these have a range for a specific latitude? Is it due to the latitudinal scale of the AEW?**

No, we give the according range to our definition of the NIW frequency in section 2.2 defined as between 1-1.2 *f*, where we also give the range for 11.5°N as 2-2.5 days.

Line 437, "... 1-D ML Model KPP, ⁶⁰... ", **1-D KPP ML model?**

Thank you, changed.

Line 438, "This result is consistent with results on equatorial shear driven mixing⁶³." **Also note that you *averaged* values over a 10 m depth range. If you didn't do the same for KPP, then I'm guessing values at the base of the ML will be substantially higher than your depth-averages. Also, one major way that all these models fail (KPP, PWP) is that they can't properly account for NIW propagation. So they do get some inertial shear at the ML base due to turbulent momentum fluxes, BUT the phase and intensity of shear can be way off at times as the models don't have IW dynamics.**

Yes, that is true we would have to compare the averaged values. In our first average KPP profile calculated with the observed CTD and velocity data as input, there was a peak of 1000 W m² directly below the ML dropping to about 150 over about 5m and a secondary peak of about 600 W m² at 30m depth. As the MLD in our observations was at around 20m

for all profiles, we formulated the above statement. We completely agree that as KPP and PWP do not feature any IW dynamics they can be way off in terms of phase and intensity of shear, however we only wanted to point towards the fact here that as pointed out previously (by Zaron and Moum for the equatorial Pacific) KPP tends to overestimate the diffusive heat flux below the ML.

Reviewing the former results due to this comment we realized that our first average KPP profile was obtained using the full KPP formulation including the boundary layer part. If we only used the interior part of KPP, the results are much closer to the observations. Hence, we reformulated this part now stating that KPP is in general consistent with observations, however as mentioned above it will not get the shear right due to the missing IW dynamics and anyway 25 MSS profiles are not sufficient to properly test a parametrization scheme like KPP.

Line 463, “ The climatological monthly error of the residual term is then calculated as the square root of the sum of the squares of the monthly climatological heat storage rate, advection, and surface heat fluxes.” **You may want to note that this error estimate does not include bias estimates or systematic error estimates. For example, when the ML is shallow, the pen. rad scheme chosen can be very important to the estimate of sw rad leaving the base of the ML. It is not going to be on the order of your residual (100 W/m²) although... more like 10.**

Thank you, we added the following: “Note that this error of the residual does not include bias estimates or systematic error estimates. However, potential biases were estimated to be on the order of maximum 20 W m⁻², when the ML is thin (Foltz and McPhaden 2009).

Line 473, “ The velocity data were ...”, **Velocity data were.**

Changed.

Line 498, “ and provides wind estimates at 6 hour resolution on a 0.25 degree horizontal grid 68, 69. **suggest rewording this sentence for clarity. Also, I think important to mention that this product assimilates *buoy data*!**

Yes, we agree it is important to mention that the buoy data is included. The paragraph now reads:... “and provides wind estimates at 6 hour resolution on a 0.25 degree horizontal grid 68, 69. The CCMP winds have been shown to better capture synoptic events that are missing in, e.g., reanalysis products 68. Note that buoy wind data is also assimilated in CCMP.”

Line 931, “ ... mid of the month.” **mid-point or middle**

Changed to middle.

Line 982, “ Grey shading in all panels indicates the time defined for the NIW event also used in Fig. 7 for the instantaneous heat budget at the PIRATA buoy.” **This didn't show in my pdf for some reason. Just the vertical lines bounding it.**

Sorry about that. It did appear in the final word version, that must have happened when the pdf was created during the resubmission process. As we have to deliver .eps files now, this issue should hopefully be solved.

Line 1040, “ b-d Monthly mean chlorophyll a distribution for the respective months February, June and August. “ **Which year(s)?**

Thank you, we added “...for the years 2002-2012.”

Line 1062, Figure 10, **This figure is hard to interpret. It looks like MLD has a significantly higher correlation with Chl-a than IA. Again, can you combine the two to**

look at a proxy for Sh^2 at the base of the ML? (e.g. q^2/MLD). You maybe have already tried this. ...and I understand MLD is climatology.

Yes, as you point out MLD is a climatology and we have tried to display this figure in different ways. However, as we state in the text one should not look for an exact match of Chl-a with either IA and MLD, as – like you point out – one has to consider q^2/MLD , which we cannot display as MLD is a climatology and there is interannual variability for both IA and MLD. In addition, as also stated in the text, other processes e.g. spatial differences in the nutrient reservoir will also influence the Chl-a distribution. We are displaying this figure to point out that not only the ML heat balance is affected by the seasonal cycle of NIEs, but also biologically relevant parameters like Chl-a, and this needs to at least be considered when interpreting the seasonal cycle of parameters like Chl-a.

Annotations in the supplementary information:

Line 71, “...strong inertial events in the region that is based only on the wind-stress signal that avoids making assumptions about the state of the ocean.” **This isn't completely true. The linear decay parameter, r , is an ad-hoc way of accounting for energy loss due to NIW radiation and other forms of energy loss. This parameter *does* say something about the state of the ocean as it appears to be dependent upon both the ML depth and possibly the upper-thermocline stratification.**

Yes, of course that is true. The damping scale r does make some assumptions about the state of the ocean. We included the word “detailed” to account for the fact that r does at least make some assumptions about the ocean state at this place.

Line 131, “ ... time scales.” **I believe there's a Pleuddemann or Farrar ref for this?**

Yes, we actually also cite Pleuddemann and Farrar at this place (reference number 7), but the sentence read “As ⁷ stated, ...”. We removed this phrase, give only the statement as the sentence and put the ⁷ at the end. Thanks for pointing out that this was not clear to read.

Supplementary Figure 3 **I guessing CCMP uses data assimilation and specifically PIRATA buoy data? --- otherwise the comparison is uncanny!**

Yes, it is true that CCMP also assimilates the PIRATA buoy wind data, which we now also explicitly state in the data section (see comment above). However, this does not mean that the time series are really identical and fit together as also other data sets influence the assimilated product and therefore we think it is important to check this.